# Hyperparameter Transfer Enables Consistent Gains of Matrix-Preconditioned Optimizers Across Scales

**Shikai Qiu**[*]    **Zixi Chen**[*]    **Hoang Phan**    **Qi Lei**    **Andrew Gordon Wilson**
New York University

## Abstract

Several recently introduced deep learning optimizers utilizing matrix-level preconditioning have shown promising speedups relative to the current dominant optimizer AdamW, particularly in relatively small-scale experiments. However, efforts to validate and replicate their successes have reported mixed results. To better understand the effectiveness of these optimizers at scale, in this work we investigate *how to scale* preconditioned optimizers via hyperparameter transfer, building on prior works such as $\mu$P. We study how the optimal learning rate and weight decay should scale with model width and depth for a wide range of optimizers, including Shampoo, SOAP, and Muon, accounting for the impact of commonly used techniques such as blocking and grafting. We find that scaling the learning rate according to $\mu$P improves transfer, but can still suffer from significant finite-width deviations that cause drifting optimal learning rates, which we show can be mitigated by blocking and explicit spectral normalization. For compute-optimal scaling, we find scaling independent weight decay as $1/\text{width}$ is nearly optimal across optimizers. Applying these scaling rules, we show Muon, SOAP and Shampoo consistently achieve near $1.4\times$ speedup over AdamW for training Llama-architecture language models of sizes ranging from 190M to 1.4B, whereas the speedup vanishes rapidly with scale under incorrect scaling. Based on these results and further ablations, we argue that studying optimal hyperparameter transfer is essential for reliably comparing optimizers at scale given a realistic tuning budget.

## 1 Introduction

Developing better optimization techniques can fundamentally accelerate deep learning. While AdamW [31] continues to dominate deep learning workloads, including training frontier-scale language models [16, 51, 29], several recently introduced optimizers have demonstrated significant speedups relative to AdamW, including Shampoo [19, 25, 3], SOAP [47], and Muon [23]. These optimizers use matrix rather than elementwise preconditioners that lead to faster learning per step while incurring a mild overhead. Notably, Muon's recent success in training trillion-parameter language models [45] points to the potential in replacing AdamW as the default optimizer at scale.

Unfortunately, subsequent works have reported mixed results in replicating these claimed speedups, especially on larger-scale experiments. For example, while Liu et al. [30] demonstrates consistent $2\times$ speedup of Muon relative to AdamW from 400M to 1.5B-parameter language models, Wen et al. [49] show that Muon is at most $1.4\times$ more efficient than well-tuned AdamW, and this speedup quickly diminishes to $1.1\times$ for 1.2B models, with SOAP following a similar trend. Similarly, Semenov et al. [39] shows the speedups of matrix-preconditioned optimizers over AdamW decrease with scale. While the different experiment setups make it difficult to fully reconcile these studies, we hypothesize that the lack of a robust and consistent way of choosing hyperparameters, particularly for the larger models where careful tuning is impractical, is likely the main reason for these inconsistent findings.

---

[*]Equal contribution. Correspondence to `sq2129@nyu.edu`, `zc2157@nyu.edu`, `andrewgw@cims.nyu.edu`.

39th Conference on Neural Information Processing Systems (NeurIPS 2025).

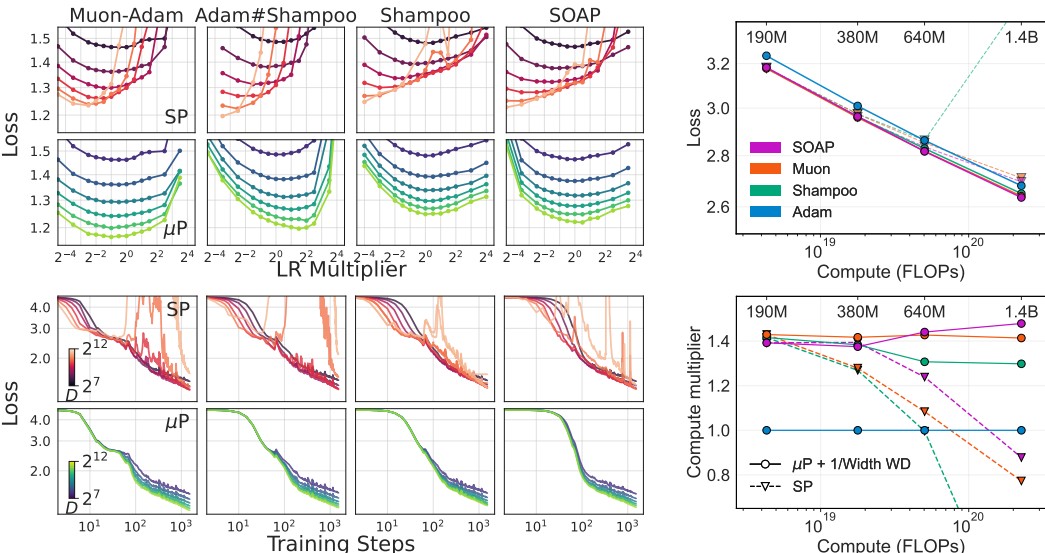

Figure 1: **Hyperparameter transfer is crucial for achieving good performance with matrix-preconditioned optimizers across scales.** (**Left**) We derive $\mu$P scaling for stabilizing the optimal learning rate across widths. Fixing the base learning rate, $\mu$P yields consistent training dynamics across widths for transformers on Open-WebText and that wider is always better, whereas the standard parameterization (SP) leads to instability and unstable optima (Section 3). Muon-Adam uses Muon in hidden layers and Adam in embedding and readout layers. Adam#Shampoo is Shampoo with Adam-grafting. Shampoo and SOAP use block size of 128. (**Right**) Combining $\mu$P with $1/$width independent weight decay, Muon, SOAP and Shampoo achieve consistent speedups around $1.4\times$ over well-tuned AdamW in training 190M to 1.4B-parameter models on FineWeb. By contrast, SP rapidly deteriorates the performance of all three optimizers as they scale (Section 4).

Indeed, a key lesson from scaling up deep learning over recent years is that hyperparameters such as initialization, learning rate, weight decay, and training horizon must be carefully co-varied as the compute budget increases to achieve efficient scaling [20, 24, 52, 17, 33, 48, 4]. Furthermore, the small loss-vs-compute scaling exponent in language modeling, estimated to be as low as $0.05$ [24, 30], means that a $2\%$ change in the loss can translate to a $40\%$ change in the estimated compute[2], making good hyperparameters critical for estimating compute efficiencies between approaches.

To accurately quantify and fully realize the advantage of more advanced optimizers at scale, it is essential to understand how to optimally choose their hyperparameters when tuning becomes infeasible. In this work, we study scaling rules for optimal hyperparameters for a range of matrix-preconditioned optimizers, and demonstrate their critical impact on how performance scales with compute. We summarize our main findings as follows:

1. We study hyperparameter transfer for matrix-preconditioned optimizers over both model width and depth. With a simple and general procedure, we show how to derive the Maximal Update Parameterization ($\mu$P) [52] for a wide range of optimizers, including Shampoo, SOAP, and Muon, accounting for the impact of commonly used techniques such as blocking and grafting [3, 43], and extend to depth scaling for residual networks, e.g., the transformer.

2. We show $\mu$P significantly improves learning rate transfer compared to unscaled learning rate (Figure 1 left). However, the optimal learning rate can be far from converged at realistic widths with certain matrix-preconditioners, particularly when used with RMS-based update normalization (Figure 2). Fortunately, we demonstrate such finite-width deviations are effectively mitigated by blocked preconditioning and spectral normalization [26] (Figure 3).

3. For compute-optimal training, we find 1) $\mu$P can achieve near-optimal transfer, 2) spectral normalization reduces learning rate sensitivity, 3) scaling the independent weight decay as $1/$width as suggested in Xiao [50] is near-optimal *across optimizers*, and 4) the compute-optimal model size is larger for matrix-preconditioned optimizers compared to Adam.

4. We show good hyperparameter transfer is crucial for realizing the benefit of matrix-preconditioning at scale. Using $\mu$P and $\Theta(1/D)$ weight decay, Muon, SOAP and Shampoo

---

[2]If $\mathcal{L} \sim C^{-\alpha}$, $dC/C = -\frac{1}{\alpha} d\mathcal{L}/\mathcal{L}$.

consistently achieve near $1.4\times$ speedups over AdamW in training 190M to 1.4B-parameter language models, and the speedups quickly vanish with incorrect scaling (Figure 1 right).

We make our code and wandb experiment logs available here.

## 2 Background

**Matrix-Preconditioned Optimizers.** While most deep learning optimizers apply either no preconditioning (SGD) or only elementwise preconditioning (Adam, Adafactor [41], Lion [13], etc.), matrix-preconditioned optimizers apply matrix-level preconditioning to the gradient of 2D parameter matrices. Let $W \in \mathbb{R}^{n \times m}$ denote the weight matrix of a given layer and $G \in \mathbb{R}^{n \times m}$ its gradient. The update rules typically take the form of $\Delta W \propto -P_L^{-1} G P_R^{-1}$, for positive definite matrices $P_L, P_R$ that depend on the past iterates. A representative example is Shampoo [19, 3, 43], where

$$\Delta W = -\eta \left(L + \epsilon_L I\right)^{-e_L} G (R + \epsilon_R I)^{-e_R}, \tag{1}$$

where $\eta$ is the learning rate, $\epsilon_{L,R}$ are regularization parameters, $e_L$ and $e_R$ are positive exponents, and $L, R$ are exponential moving averages of $GG^\top$ and $G^\top G$ respectively over the trajectory. The Shampoo preconditioner can be interpreted as a one-step Kronecker-factored approximation of the empirical Fisher using power iteration [35]. The original Shampoo uses $e_L = e_R = 1/4$ [19] while later works found $e_L = e_R = 1/2$ to perform better [35, 43]. A key weakness of the original Shampoo is that the matrix inversion scales cubically with the width of layer and can be unstable. To make Shampoo more practical, *blocking* [43, 3] partitions the parameter into sub-blocks, reducing the peak memory and compute overhead of the preconditioners. *Grafting* [3, 43, 1] normalizes the Frobenius norm of the Shampoo update by that of another optimizer, such as Adam, improving the stability. Besides Shampoo, K-FAC [18] and Preconditioned Stochastic Gradient Descent [28] also leverage structured-matrix approximations to the (empirical) Fisher as preconditioners.

Taking inspiration from Shampoo, several new optimizers have been recently introduced. Bernstein and Newhouse [7] showed that without the exponential moving average, Shampoo simplifies to $\Delta W = -\eta U V^\top$, where $G = U \Sigma V^\top$ is the SVD of the gradient matrix. The Muon optimizer [23] leverages this insight, using an iterative method to approximate $U V^\top$ directly, and can be viewed alternatively as steepest descent in the spectral norm. Muon has gained traction due to its algorithmic simplicity and robust performance. SOAP [47], another Shampoo-inspired algorithm, recognizes that Shampoo effectively applies Adafactor in the preconditioner's eigenbasis and builds on this insight by applying Adam in this eigenbasis instead. AdaMuon [44] applies Adam on top of Muon's orthogonalized gradient. We provide detailed update rules in Appendix A.

**Infinite-Limits and Hyperparameter Transfer.** The Maximal-Update Parameterization ($\mu$P) [52, 54, 53] pioneered a series of work on hyperparameter (HP) transfer: how HPs such as learning rate, initialization, and weight decay should scale as a function of width to allow stable and maximal feature learning in every layer as the width (e.g., the transformer's embedding dimension) approaches infinity. Empirically, under $\mu$P, HPs tuned on a small model can be transferred to models that are orders of magnitude larger without re-tuning [54]. Extending this idea to depth scaling, subsequent works have proposed to multiply the residual block output in transformers or other residual networks by $1/L^\alpha$ and scale the learning rate accordingly to ensure the magnitude of feature learning is stable across depth, where $L$ is the depth and $\alpha \in [0.5, 1]$ [57, 9, 10, 14]. Depth-$\mu$P [57] follows $\alpha = 0.5$ in order to maximize diversity of feature across the layers, while CompleteP [14] uses $\alpha = 1$ to ensure nontrivial feature learning in *each* layer, demonstrating better results in training deep transformers.

Prior works primarily explored hyperparameter transfer in the most widely used optimizers like SGD, Adam, and Adafactor [17], while analogous prescriptions for matrix-preconditioned methods remain underexplored, with the exception of Ishikawa and Karakida [21] who derived $\mu$P for vanilla Shampoo (without blocking or grafting) and K-FAC. Several recent large-scale investigations of Muon have resorted to heuristics for scaling its learning rate with width, such as directly using $\mu$P for Adam [40] or matching the update RMS of Adam [30], both are incorrect as we show in Appendix G.

**Finite-Width Scaling via the Spectral Norm.** Yang et al. [56] showed that one can recover $\mu$P's initialization and learning rate scaling by constraining the spectral norm of the initialization and update of each layer to $\Theta(\sqrt{d_{\text{out}}/d_{\text{in}}})$, corresponding to $\Theta(1)$ RMS-RMS operator norm. Instead of appealing to infinite-width asymptotics, Large et al. [26] showed that explicitly normalizing the updates to have precisely $\sqrt{d_{\text{out}}/d_{\text{in}}}$ spectral norm (before being multiplied by the learning rate)

provides worst-case stability guarantees for finite-width networks and empirically enables good learning rate transfer for Adam and SGD with a modest overhead. As Muon already normalizes the update spectral norm, its learning rate scales as $\Theta(\sqrt{d_{\text{out}}/d_{\text{in}}})$ as identified in [36, 2]. Scion [36] generalizes the choice of the operator norm and demonstrates good hyperparameter transfer on transformer models.

Table 1: Scaling rules for learning rate as a function of the weight matrix size $(d_{\text{in}}, d_{\text{out}})$, depth $L$, block size $(b_{\text{in}}, b_{\text{out}})$ and number of blocks $n_{\text{blk}}$ if using blocking. A multiplier $1/L$ is assumed to apply to the output of every residual block. $e_{L,R}$ are positive exponents for Shampoo, and 0-1 indicators for preconditioning either side for SOAP. For parameters outside of the residual blocks (e.g. embedding and last layer), set $L = 1$. $\eta_{Q_1}$ denotes the correctly scaled learning rate for optimizer $Q_1$. We provide scaling of $\epsilon$ parameters in Appendix B.

| Shampoo $(e_L, e_R)$ | SOAP $(e_L, e_R)$ | Muon | AdaMuon | Grafting $Q_1 \# Q_2$ |
|---|---|---|---|---|
| $\dfrac{(d_{\text{out}}/d_{\text{in}})^{1-(e_L+e_R)}}{L^{2(e_L+e_R)-1} n_{\text{blk}}^{e_L+e_R}}$ | $\dfrac{b_{\text{out}}^{e_L/2} b_{\text{in}}^{e_R/2}}{d_{\text{in}}}$ | $\sqrt{\dfrac{d_{\text{out}}}{d_{\text{in}}}}$ | $\dfrac{1}{d_{\text{in}}}$ | $\eta_{Q_1}$ |

## 3 Hyperparameter Transfer for Matrix-Preconditioned Optimizers

In this section, we present our scaling rules for the per-layer learning rate (and regularization $\epsilon$ when applicable) as a function of width and depth to ensure consistent feature learning as the model scales. Table 1 summarizes the results for optimizers we experiment with, though generalization is straightforward. Bias parameters have $d_{\text{in}} = 1$. We provide a sketch of the derivations with high-level intuitions here and present the full derivations in Appendix B. $Q_1 \# Q_2$ denotes grafting where we take the update direction of $Q_2$ but scaled to the Frobenius norm of $Q_1$. For a vector $v \in \mathbb{R}^d$, we write $v = \Theta(g(d))$ or $v \sim g(d)$ to mean $\|v\|_{\text{RMS}} = \Theta(g(d))$, where $\|v\|_{\text{RMS}} \equiv \sqrt{\sum_i v_i^2/d}$. In Section 3.4, we study co-scaling model size and training steps and investigate weight decay scaling.

### 3.1 Width Scaling via $\mu$P

We start with $\mu$P for scaling learning rate and other optimizer hyperparameters with width. As we will show, a simple and general procedure suffices for a wide range of optimizers, each requiring only a few lines of derivation. For simplicity, we consider training an $L$-layer ($L$ assumed fixed for now) MLP that outputs $f(\xi)$ on an input $\xi$ as follows:

$$x_0(\xi) = \xi, \; h_\ell(\xi) = W_\ell x_{\ell-1}(\xi), \; x_\ell(\xi) = \phi(h_\ell(\xi)), \; \ell = 1, \ldots, L-1, \; f(\xi) = h_L(\xi) = w_L^\top x_{L-1}(\xi),$$

where weights are drawn from a Gaussian $\mathcal{N}(0, \sigma_\ell^2)$ with layerwise variances $\sigma_\ell^2$ and $\phi$ is an element-wise nonlinear function. We assume $\xi \in \mathbb{R}, f(\xi) \in \mathbb{R}, h_\ell(\xi) \in \mathbb{R}^d$ for $\ell = 1, \ldots, L-1$, and refer to $d$ as the width. As shown in prior works, the conclusions derived here will hold for more general architectures consisting of linear layers and element-wise non-linearities, modulo a minor modification for the $1/d_{\text{head}}$ instead of $1/\sqrt{d_{\text{head}}}$ scaling for attention logits [54, 53].

As $\mu$P cares only about the large-width asymptotics with training steps and batch size held at $\Theta(1)$, analyzing the dynamics up to the first gradient step with a batch size of one turns out to be sufficient for obtaining the correct scaling [52, 53, 56], while leading to great simplifications (e.g. gradient is rank 1 and momentum can be ignored). To carry out the analysis, we track the change in layer outputs on a generic input $\xi'$ induced by the first gradient update on a training point $\xi$. We denote the change in any quantity $X$ after this step as $\Delta X$. We discuss generalization to batch size greater than one in Appendices B.9 and B.10.

**Initialization and Gradient Scales are Optimizer-Independent.** We first make the observation that the initialization scale $\sigma_\ell$ of $W_\ell$ is independent of the optimizer (e.g., identical to established scalings for SGD and Adam), with $\sigma_\ell = \Theta(1/\sqrt{d_{\text{in},\ell}})$ for $\ell < L$ and $\sigma_L = \Theta(1/d)$. This result follows from combining the stability and feature learning conditions of $\mu$P, which state that $h_\ell(\xi') = W_\ell x_{\ell-1}(\xi') = \Theta(1), f(\xi') = O(1)$ and $\Delta h_L = \Theta(1)$, respectively [52]. We leave detailed justifications for why this holds beyond SGD and Adam to Appendix B, but the gist is that nontrivial feature learning in the last layer inevitably constrains $\sigma_L$ to be small in $d$ for $f$ to not blow up with $d$. As a direct consequence, entries of the gradient of the loss w.r.t. the pre-activations $\delta_\ell(\xi') \equiv \nabla_{h_\ell(\xi')} \mathcal{L}$ scale as $\Theta(1/d_{\text{out},\ell})$ for all $1 \leq \ell \leq L$, since 1) for $\ell = L$, $\delta_L(\xi') = d\mathcal{L}/df(\xi') = \Theta(1)$ since $f(\xi') = O(1)$; 2) for $\ell = L - 1$,

$\delta_{L-1}(\xi') = \delta_L(\xi')w_L = \Theta(\sigma_L) = \Theta(1/d)$; and 3) backpropagating through the hidden layers preserve the scale of the gradient given that $\sigma_\ell = \Theta(1/\sqrt{d_{\text{in},\ell}}) = \Theta(1/\sqrt{d_{\text{out},\ell}})$ for $1 < \ell < L$. We can now state the much simplified $\mu$P condition for general optimizers.

**Simplified $\mu$P Condition for General Optimizers.** To ensure each layer is subsequently updated as much as possible without diverging, $\mu$P requires that it produces a $\Theta(1)$ update to its output for every gradient step. Let $\Delta W_\ell$ denote the update to $W_\ell$ due to training on $\xi \in \mathbb{R}$, this condition requires

$$\Delta W_\ell x_{\ell-1}(\xi') = \Theta(1), \quad \ell = 1, \dots, L. \tag{2}$$

The gradient of $W_\ell$ computed on a datapoint $\xi \in \mathbb{R}$ is $G_\ell(\xi) \equiv \nabla_{W_\ell}\mathcal{L}(\xi) = \delta_\ell(\xi)x_{\ell-1}(\xi)^\top$, where $\delta_\ell(\xi) \in \mathbb{R}^{d_{\text{out},\ell}}$ and $x_{\ell-1}(\xi) \in \mathbb{R}^{d_{\text{in},\ell}}$. To simplify notation, we will omit the layer subscripts and abbreviate $x(\xi)$ as $x$, $x(\xi')$ as $x'$, $\delta(\xi)$ as $\delta$, and $G(\xi)$ as $G$. Let $Q(G)$ be the update returned by the optimizer, we therefore wish to choose per-layer learning rate $\eta$ such that $\eta Q(G)x' = \Theta(1)$. Finally, we will use the fact that $x^\top x'$ behaves as a sum of $d_{\text{in}}$ i.i.d. correlated random variables (correlation due to being produced from the same weights) and thus scaling as $\Theta(d_{\text{in}})$, a standard result on the feature kernels of wide neural networks [38, 27, 22, 52, 8]. We will refer to this property as the *alignment* between $x$ and $x'$, following [56, 17].

Putting it together, the $\mu$P condition reduces to choosing $\eta$ and other hyperparameters in $Q$ such that

$$\boxed{\eta Q(G)x' = \Theta(1), \quad \text{where } G = \delta x^\top, \ \delta = \Theta(1/d_{\text{out}}), \ x = \Theta(1), \ x^\top x' = \Theta(d_{\text{in}}).} \tag{3}$$

Note $x^\top x = \Theta(d_{\text{in}})$ and $\delta^\top \delta = \Theta(1/d_{\text{out}})$ are implied. Given these scaling relations, solving for the learning rate (and other hyperparameters) boils down to fairly straightforward algebra. We now illustrate with a few illustrative examples and delegate full derivations for the results Table 1 to Appendix B. In Appendix B.8, we also show that the spectral norm condition $\eta\|Q\|_2 = \Theta(\sqrt{d_{\text{out}}/d_{\text{in}}})$ [56] remains a valid alternative characterization of $\mu$P for matrix-preconditioned optimizers.

**Example: Shampoo.** The rank–1 preconditioners are $L = \delta\,(x^\top x)\,\delta^\top$, $R = x\,(\delta^\top \delta)\,x^\top$ with unique nonzero eigenvalues $\lambda_L = \lambda_R = (x^\top x)(\delta^\top \delta) = \Theta\left(\frac{d_{\text{in}}}{d_{\text{out}}}\right)$. The preconditioned update is

$$Q(G) = (L + \epsilon_L I)^{-e_L}\, G\, (R + \epsilon_R I)^{-e_R}. \tag{4}$$

For $\epsilon_L, \epsilon_R$ to have $\Theta(1)$ effects, they must balance the only scale here, namely $\epsilon_{L,R} = \lambda_{L,R} = \Theta(d_{\text{in}}/d_{\text{out}})$. Since $G = \delta x^\top$ aligns with the corresponding eigenvectors,

$$Q(G)\,x' = (\lambda_L + \epsilon_L)^{-e_L}(\lambda_R + \epsilon_R)^{-e_R}\,\delta\,(x^\top x') \sim \left(\frac{d_{\text{in}}}{d_{\text{out}}}\right)^{1-e_L-e_R}, \tag{5}$$

hence $\eta \sim (d_{\text{out}}/d_{\text{in}})^{1-e_L-e_R}$. These results reproduce those in Ishikawa and Karakida [21]. If defined relative to $\lambda_{L,R}$, as done in [3], $\epsilon_{L,R}$ should be $\Theta(1)$ instead.

**As a Tensor Program.** Observe that Equation (5) expresses the Shampoo update purely in terms of vectors with i.i.d. entries, their inner products, and scalar nonlinearities, which shows that it can be expressed as a Tensor Program [53]. Thus, the full training process admits a well-defined, deterministic infinite-width limit by the Master Theorem [53], a much stronger statement than features updates being $\Theta(1)$ and a more fundamental reason why we should expect the learning rate to transfer. We show an analogous construction for batch size greater than one in Appendix B.10.

**Example: Shampoo with Blocking.** Blocking partitions the gradient into blocks $G_{ij} = \delta_i x_j^\top \in \mathbb{R}^{b_{\text{out}} \times b_{\text{in}}}$, where $\delta_i \in \mathbb{R}^{b_{\text{out}}}$ and $x_j \in \mathbb{R}^{b_{\text{in}}}$ are the $i$-th and $j$-th chunks of $\delta$ and $x$, each preconditioned independently. Within each block, the rank-1 preconditioners now have unique nonzero eigenvalue $\lambda_{ij} = (x_j^\top x_j)(\delta_i^\top \delta_i) = \Theta\left(\frac{b_{\text{in}} b_{\text{out}}}{d_{\text{out}}^2}\right)$, which sets the scale of $\epsilon_{L,R}$. Thus $Q(G_{ij})\,x_j' = \lambda_{ij}^{-(e_L+e_R)}\,\delta_i\,(x_j^\top x_j') \sim \left(\frac{b_{\text{in}} b_{\text{out}}}{d_{\text{out}}^2}\right)^{-(e_L+e_R)}\frac{b_{\text{in}}}{d_{\text{out}}}$. Inverting this quantity and further dividing number of blocks along the input dimension $n_{\text{in}} = d_{\text{in}}/b_{\text{in}}$ to account for the correlated contributions of $\{Q(G_{ij})x_j\}_{j=1}^{n_{\text{in}}}$, we get $\eta \sim \left(\frac{d_{\text{out}}}{d_{\text{in}}}\right)^{1-e_L-e_R}(n_{\text{in}} n_{\text{out}})^{-(e_L+e_R)}$ after some algebra, where $n_{\text{out}} = d_{\text{out}}/b_{\text{out}}$. Note when $e_L + e_R = \frac{1}{2}$ and $b_{\text{in}} = b_{\text{out}} = 1$, Shampoo degenerates to Adam and we recover the Adam scaling $\eta \sim 1/d_{\text{in}}$.

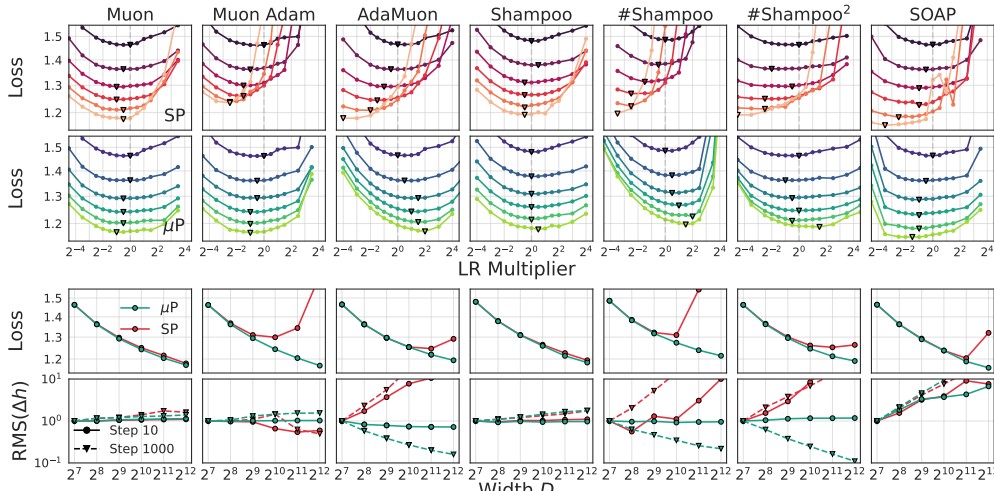

Figure 2: $\mu$**P leads to better but imperfect learning rate transfer for matrix-preconditioned optimizers.**
(**Top**) The optimal learning rate is more consistent across widths $D$ under $\mu$P for transformers trained on OpenWebText. We show the learning rate as the multiplier $\eta_{\text{base}}/\eta_0$, where $\eta_0$ is the optimal learning rate for the base model for each optimizer. (**Bottom**) $\mu$P achieves lower loss in zero-shot transferring the optimal learning rate found in the base model ($D = 128$) to larger models (up to $D = 4096$) and passes the "coordinate check": RMS of the one-step feature update in the last layer is invariant to width in early training (step 10), except for SOAP. We explain the imperfect transfer of $\mu$P and why it fails for SOAP in Section 3.2 . # stands for Adam-grafting, Muon-Adam uses Adam for the embedding and readout, and Shampoo$^2$ uses $e_L = e_R = 1/2$.

**Example: Shampoo with Adam Grafting.** While an explicit calculation in the style above suffices, the spectral norm condition [56] provides an easy alternative to reason about grafting, which normalizes the Shampoo update's Frobenius norm by that of Adam's. Since the preconditioner leaves Shampoo's update $\Theta(1)$-rank, its Frobenius norm scales identically with its spectral norm, a relation that holds also for Adam due to Adam update being low *stable* rank [56, 53]. Consequently, $\|Q_{\text{Adam\#Shampoo}}(G)\|_2 \sim \|Q_{\text{Adam}}(G)\|_2$, so with Adam grafting the learning rate should scale as if directly using Adam. More generally, $Q_1\#Q_2$ should scale the learning rate as if directly using $Q_1$, as long as the stable rank of both optimizers have ratio $\Theta(1)$.

## 3.2 Empirical Validation of $\mu$P and Improving Finite-Width Transfer

We train transformers on the OpenWebText dataset for 100M tokens. We use a small vocabulary of size 96 so it is practical to apply the full preconditioners on the embedding and readout layers in order to verify whether learning rate is scaled correctly for those layers where only one of the dimensions grow. We compare $\mu$P, where the per-layer learning rate is parameterized by $\eta_\ell(D) = \eta_{\text{base}}(D/D_{\text{base}})^{-\alpha_\ell}$ with $\eta_{\text{base}}$ the base learning rate, $D_{\text{base}}$ the base width, and $\alpha_\ell$ derived from Table 1 for each layer $\ell$ and optimizer, against the Standard Parameterization (SP)[3], where $\eta_\ell(D) = \eta_{\text{base}}$. For example, when using Muon, $\alpha_\ell = 0$ for hidden layers, $1/2$ for embedding, and $-1/2$ for readout. We set $D_{\text{base}} = 128$, the width at which SP matches $\mu$P. See further experiment details in Appendix D.

$\mu$**P Leads to Better Transfer and Consistent Early Dynamics.** In Figure 2, we find that $\mu$P leads to a more stable learning rate landscape (loss vs $\eta_{\text{base}}$) than SP for all optimizers as $D$ scales from 128 to 4096. Though the optimal learning rate is not always exactly stable, as expected from experiment noise and finite-width effects, $\mu$P consistently outperforms SP when zero-shot transferring the optimal learning rate found on the base model (Figure 2 3rd row). In early training (step 10), the RMS of the one-step feature update is invariant to width in $\mu$P as desired (with one exception for SOAP which we will soon discuss), passing the "coordinate check" [55], but can quickly diverge in SP (Figure 2 4th row). Furthermore, Figure 1 (bottom left) shows for a fixed $\eta_{\text{base}}$ in $\mu$P, the loss curves are *consistent* across widths and *wider is always better*, revealing convergence towards the infinite-width maximum update limit. That is, the network output for any fixed input $\xi$ at any fixed time step $t$ converges to a deterministic value [52, 8, 46].

---

[3]We zero-initialize the readout layer for both $\mu$P and SP, making LR and $\epsilon$ scaling their only difference.

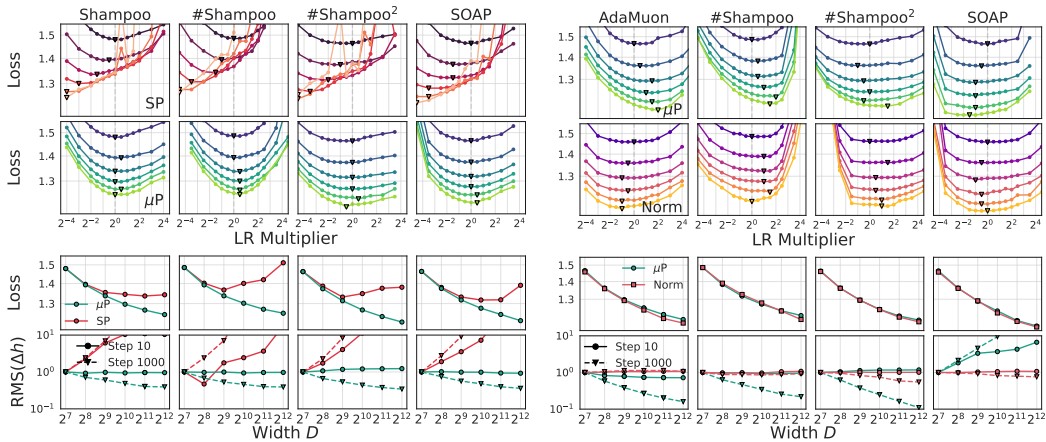

Figure 3: **Blocking and explicit normalization reduce finite-width deviations and improve transfer.** (**Left**) With a fixed block size of 128, $\mu$P consistently achieves good learning rate transfer, including for Shampoo with grafting and SOAP which otherwise have unstable optimal learning rates (Figure 2). (**Right**) Explicit spectral normalization (Norm) improves learning rate transfer where $\mu$P alone fails (no blocking used here).

**RMS Update Normalization Leads to Inconsistent Late Dynamics.** Figure 2 shows the optimal learning rate consistently increases with width in $\mu$P for AdaMuon, Shampoo and Shampoo$^2$ with grafting, accompanied by a decreasing trend in the feature updates in late training (step 1000). We attribute these phenomena to similar update normalizations performed by these optimizers, where the RMS of the optimizer update $U$ is normalized (via Adam-grafting or elementwise scaling) to $\Theta(1)$ *after applying matrix preconditioners*. Thus $U$ has a spectral norm $\Theta\left(\sqrt{d_{\text{in}}d_{\text{out}}/\text{srank}(U)}\right)$ where srank denotes the stable rank. At step $t$ with batch size $B$, the stable rank of the update is bounded by $\min(D, tB)$.[4] Whereas $\mu$P takes $D$ alone to infinity, reducing this quantity to $\Theta(1)$, for realistic finite widths and large $t, B$ ($B$ is typically millions of tokens for language models) it is $D$ that bottlenecks the stable rank, especially with matrix preconditioners designed to inflate the small singular values in $U$. As a result, with RMS-based normalization, at late times $\mu$P can undershoot the spectral norm of the update by $\sqrt{\text{srank}(U)} \gg 1$, which we show transitions from $\Theta(1)$ to growing with $D$ as training progresses for matrix-preconditioned optimizers (Figure 7).

**$\mu$P Can Fail to Model Expressive Preconditioners Even at Initialization.** Figure 2 shows $\mu$P fails the coordinate check for SOAP even at the start of training. In Appendix E.2, we trace this failure again to the infinite-width limit's inability to capture realistic finite-width behavior of expressive matrix preconditioners. Specifically, the eigenbasis-projected gradient is sparse in the infinite-width limit but dense for realistic finite widths and large batch sizes. See detailed findings in Appendix E.2.

**Blocking and Spectral Normalization Mitigate Finite-Width Deviations.** Fortunately, we find that both above failure modes of $\mu$P at finite-width transfer can be effectively mitigated by two simple approaches: blocking and explicit spectral normalization. When using a fixed block size, typically done to reduce the Shampoo optimizer overhead [43, 3], the preconditioner only operates on fixed-sized blocks, leaving less room for non-trivial finite-$D$ scaling, which only scales the number of independently preconditioned blocks. Figure 3 (left) shows that by applying blocking we restore good learning rate transfer in $\mu$P for SOAP, Shampoo and Shampoo$^2$ with Adam-grafting, and observe a milder decrease in feature update size in late training.

Spectral normalization, as proposed in [26], normalizes the update spectral norm to *exactly* (rather than asymptotically) $\sqrt{d_{\text{out}}/d_{\text{in}}}$, a strictly stronger constraint than $\mu$P [56] that can accommodate precisely the kind of finite-width deviations we have observed. Figure 3 (right) shows spectral normalization transfers learning rates equally well or better than $\mu$P and achieves markedly more consistent feature updates in late training. Implemented with online power iteration, this normalization requires only two matrix-vector multiplies per layer per step [26], a tiny overhead for matrix-preconditioned optimizers. We provide experiment and implementation details in Appendix F.

---

[4]Momentum results in at most linear scaling with $t$.

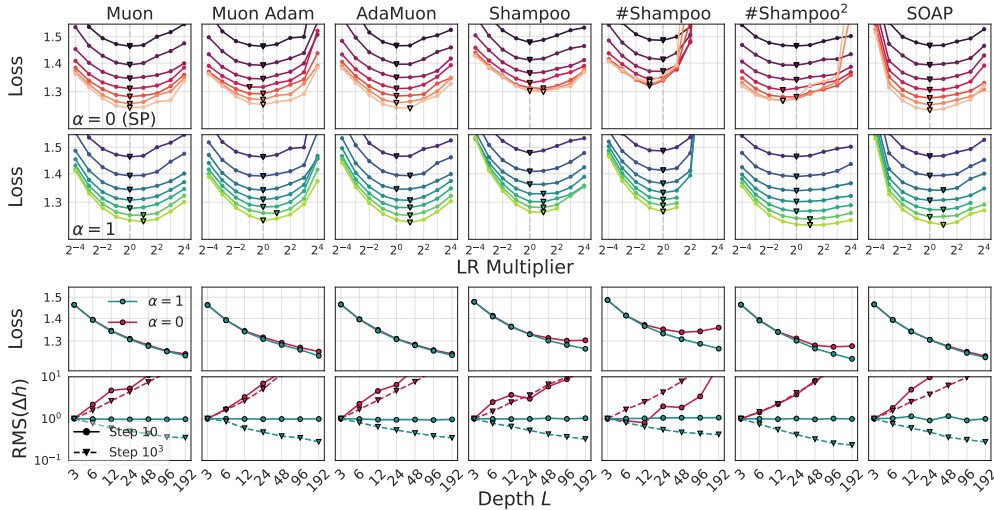

Figure 4: **Depthwise learning rate transfer is effective for all tested optimizers**. We apply a $1/L$ residual branch multiplier ($\alpha = 1$) and adjust the learning rate to ensure $\Theta(1)$ feature learning in each layer, following Dey et al. [14], outperforming SP ($\alpha = 0$) and stabilizing the size of early-time (step 10) feature update $\Delta h$ when transferring from 3 to 192-layer transformers on OpenWebText. We provide experiment details in Appendix D.2

### 3.3 Depthwise Hyperparameter Transfer

We now consider depth scaling for architectures with residual blocks (e.g. the transformer). For the MLP example, each block computes $h_\ell = W_\ell x_{\ell-1}$, $x_\ell = x_{\ell-1} + \phi(h_\ell)$. In general, the residual block can be more complex, such as involving layer normalization and multiple matrix multiplies as in the case of a transformer. Recent work has shown that by scaling down the output of each residual block by $\Theta(1/L^\alpha)$, i.e. $x_\ell = x_{\ell-1} + L^{-\alpha}\phi(h_\ell)$, with $\alpha \in [1/2, 1]$, and adjusting the learning rate to ensure stable feature learning leads to well-defined depth scaling limits [57, 9, 10]. Dey et al. [14] shows choosing $\alpha = 1$ and ensuring $\Theta(1)$ feature learning within each block leads to the best learning rate transfer for transformers trained with Adam [10, 14], referred to as the CompleteP. We derive the depth scaling rules for matrix-preconditioned optimizers based on the same criterion.

With our setup so far, the only change to the previous derivation in the width-only scaling case is that gradients in the residual blocks now scale as $\frac{1}{Ld_{\text{out}}}$ rather than $\frac{1}{d_{\text{out}}}$. Propagating these additional $L$-dependent factors to the solution of $\eta$ and $\epsilon$ leads to the final width-depth joint scaling rules in Table 1. See derivations in Appendix B. Figure 4 verifies the effectiveness of our depth scaling rules on models with $L = L_{\text{base}} = 3$ to $L = 192$ layers. Compared to width scaling, the optimal learning rates shift less for SP ($\alpha = 0$), but our scaling with $\alpha = 1$ achieves consistently lower loss and more stable feature update size when transferring optimal learning rate found on the base model.

> **Summary:** For width scaling, $\mu$P outperforms SP at learning rate transfer for all optimizers, though its infinite-width asymptotics can fail to model realistic finite-width scaling of matrix preconditioners, leading to shifting optima. Blocking and spectral normalization mitigate these finite-width deviations. For depth scaling, generalizing CompleteP, which scales residual branches as $1/L$ and adjusts LR to ensure $\Theta(1)$ feature learning per layer, is effective for all optimizers.

### 3.4 Hyperparameter Transfer Under Compute-Optimal Scaling

We now investigate how to scale the hyperparameters in the compute-optimal setup [20, 24]. We use the FineWeb dataset tokenized with the GPT-2 tokenizer, and train each model for 20 tokens per parameter. We use a fixed block size of 512 and detail the experiment setup in Appendix D.3.

$\mu$**P Approximately Stabilizes the Optimal Learning Rate.** Despite violating $\mu$P's assumption of $\Theta(1)$ training steps, we find $\mu$P still improves the stability of the learning rate landscape and the optimal learning rate (Figure 5 left), in contrast to SP where the optima shift significantly to the left. While the optimal learning rates aren't exactly invariant, transferring the optimal learning rate found on the base $D = 256$ model leads to near-optimal performance with $\mu$P up to $D = 2048$. Spectral normalization leads to slightly better performance than $\mu$P and reduces learning rate sensitivity.

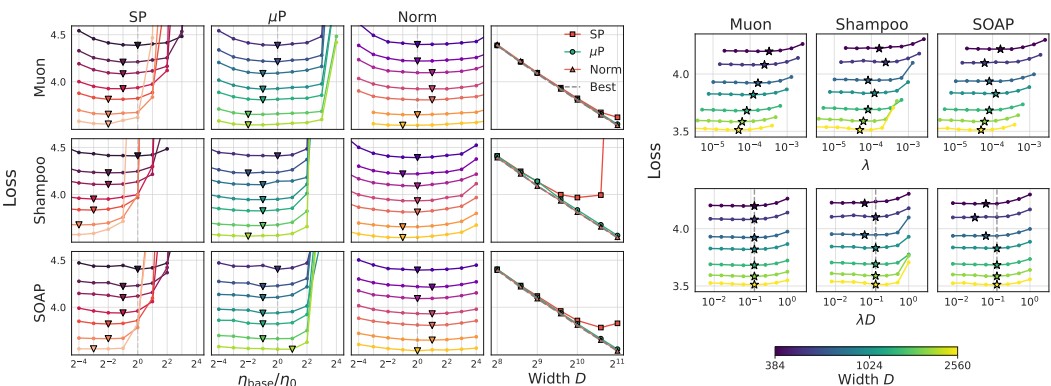

Figure 5: **Learning rate and weight decay transfer on FineWeb under compute-optimal training**. (**Left**) $\mu$P approximately stabilizes the optimal learning rate, while spectral normalization reduces learning rate sensitivity and achieves slightly better performance. Best indicates taking the minimum over all learning rates and parameterizations. (**Right**) Optimal independent weight decay scales like $1/D$. Muon uses Adam in the embedding layer. Shampoo and SOAP use a block size of 512. For Shampoo, we use Adam-grafting and Adam in the embedding and readout, which we found to perform better than applying one-sided Shampoo.

While our results align with findings in [14, 11, 34] on the effectiveness of $\mu$P in the compute-optimal regime, the larger-scale, higher-resolution sweeps done in Everett et al. [17] using Adam found that $\mu$P can considerably overshoot the learning rate. Therefore, while we observe a clear advantage of $\mu$P over SP, we caution that more careful scaling studies may be needed to determine how the optimal learning rate scales in the compute-optimal regime if width varies by more than a factor of $10$, such as by fitting a power-law correction *on top of* $\mu$P.

**Optimal Independent Weight Decay Scales as** $1/D$**.** While $\mu$P prescribes a $\Theta(1)$ independent weight decay $\lambda$ [53], Xiao [50] found $\lambda = \Theta(1/D)$ empirically leads to better scaling for AdamW. In Figure 5 (right), we find that $\lambda = \Theta(1/D)$ (bottom) is near-optimal for all three optimizers. At this scale, the models are still relatively small that parameters and tokens scale close to linearly with $D$, making $\lambda = \Theta(1/D)$ similar to keeping AdamW's EMA timescale constant [48, 42, 5]. We leave a finer comparison between the two approaches to future work.

> **Summary:** $\mu$P leads to good LR transfer under compute-optimal scaling, while the optimal independent weight decay scales as $1/D$. Thus, constant LR-coupled weight decay happens to work well for AdamW, but can be highly suboptimal for other optimizers (e.g., Muon).

## 4 Compute-Efficiency Gains of Matrix-Preconditioned Optimizers

We now evaluate the compute-efficiency gains of Muon, SOAP and Shampoo over AdamW in training up to $1.4$B-parameter language models and quantify the impact of good hyperparameter transfer.

**Experiment Setup.** We follow a similar setup to that in Wen et al. [49]. We use the Llama architecture, matching width and depth configurations in Wen et al. [49], covering four model sizes ranging from 190M to 1.4B. The model specifications are provided in Table 3. All models are trained on randomly shuffled FineWeb tokenized using the GPT-2 tokenizer. We use a context length of 1024 and batch size of 128 sequences. We extensively tune hyperparameters for each optimizer on the *base* model, which has 32 layers, embedding dimension 512, and 190M parameters. We detail the optimizer definition and hyperparameter tuning in Appendix H. As our batch size (0.12M tokens) is small compared to typical LLM training, we expect our findings to serve as a lower bound on matrix-preconditioned optimizers' speedup, considering they scale better with large batch sizes [49, 58, 32]. We use a block size of 512 for SOAP and Shampoo to achieve good learning rate transfer under $\mu$P

**Muon, SOAP and Shampoo Achieve Consistent Speedup.** Figure 6 (left) shows the scaling trend for the three optimizers, following either our proposed scaling with $\mu$P and $1/D$-scaled independent weight decay or constant learning rate and weight decay (SP). We quantify the speedup of each approach with its compute multiplier, which measures the compute-efficiency gain over AdamW (with $\mu$P and scaled weight decay) for achieving the same loss (details in Appendix H.2). Following Liu et al. [30], we count the FLOPs in the forward and backward passes, not the optimizer transform, which prior works [30, 47] argue can incur minimal runtime overhead with proper implementation.

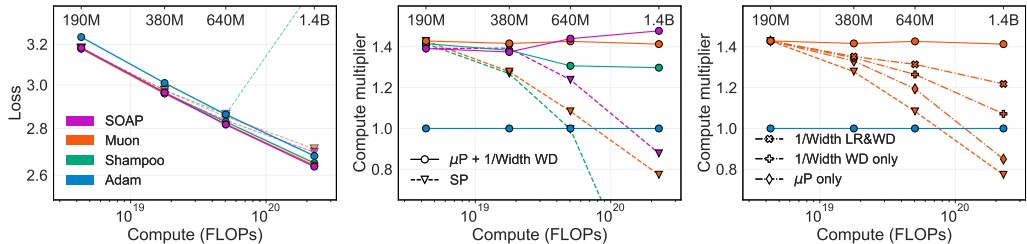

Figure 6: **With $\mu$P and $1/D$-scaled weight decay, Muon, SOAP and Shampoo achieve consistent speedups over AdamW across model sizes up to 1.4B**. Incorrect scaling rules (SP or ablating either $\mu$P or weight decay scaling) degrade performance significantly with scale. Compute multiplier is measured against AdamW with $\mu$P and scaled weight decay, which we show yields near-optimal hyperparameters for large models (Appendix H.4).

Figure 6 (right) shows Muon, SOAP and Shampoo achieve a stable speedup factor over AdamW of around $1.4\times$, using our proposed scaling rule. By contrast, SP leads to rapidly diminishing speedup.

**Hyperparameter and Scaling Rule Ablations.** In Appendix H.3, we verify that our scaling rule leads to near-optimal learning rate and weight decay for the larger models for all four optimizers by showing that perturbing them by factors of 2 in either direction leads to equal or worse performance. For Muon, we further ablate $\mu$P or weight decay scaling one at a time, shown in Figure 6 (right) and Appendix H.4, and find that both contribute significantly to achieving good performance at scale. Scaling Muon's learning rate and weight decay both as $1/$width, an effective scaling for AdamW [50], improves over SP but still falls short of $\mu$P $+ 1/D$ weight decay, decreasing the speedup to below $1.1\times$ for the 1.4B model. These results indicate that both components of our scaling rule are necessary to achieve the best performance and consistent speedups over AdamW across scales.

**Optimal Tokens Per Parameter Depends on the Optimizer.** Optimizers that achieve faster convergence are likely to have a smaller compute-optimal tokens-per-parameter (TPP) ratio, since diminishing returns from training on more data kick in faster (if loss at convergence is unchanged). In Appendix H.5, we find the optimal TPP on FineWeb is 9.6 for Adam and 7.4 for Muon. We also find that the optimal learning rate follows $\mu$P while independent weight decay should be kept constant as we vary model sizes subject to the fixed FLOPs budget, consistent with Bergsma et al. [4].

**Summary:** Muon, SOAP and Shampoo achieve consistent speedups around $1.4\times$ over AdamW from 190M to 1.4B parameters, but only under good hyperparameter transfer. We verify that combining $\mu$P and $1/D$-scaled weight decay yields near-optimal hyperparameter transfer for all tested optimizers, and both components are critical for achieving the best performance.

## 5 Discussion

Robust optimizer comparisons at scale depend as much on the scaling rules as on the optimizers themselves. We demonstrate that even modest deviations from optimal scaling rules incur dramatic efficiency losses, sufficient to obscure meaningful differences between optimizers. Under our best-effort optimal scaling rules, matrix-preconditioned optimizers like Muon, SOAP and Shampoo prove significantly more compute-efficient than AdamW up to 1.4B models with relatively stable efficiency gains, in contrast to the findings in Wen et al. [49]. Among the tested optimizers, we find Muon achieves top performance while being the easiest to use, tune, and scale, due to its algorithmic simplicity, small number of hyperparameters, and low compute and memory overhead.

The Maximum Update Parameterization ($\mu$P) is a crucial component of our scaling rule. While $\mu$P has a reputation of being mathematically involved, we show it can be generalized to many matrix-preconditioned optimizers with a brief and simple calculation. However, we identify a few settings where $\mu$P fails to model expressive preconditioners at realistic widths and leads to suboptimal learning rate transfer. Developing more robust scaling rules compatible with expressive preconditioning likely requires a stronger focus on understanding finite-width dynamics, such as in Large et al. [26], or designing optimizers with finite-width guarantees built in, e.g., Muon and Scion [7, 36].

Lastly, our experiments show that scaling weight decay as $1/$width matters as much for the performance as scaling the learning rate. Understanding why this scaling is effective across optimizers, how weight decay shapes the training dynamics of matrix-preconditioned optimizers, and whether better scaling exists, are exciting and important open questions.

## Acknowledgments and Disclosure of Funding

We thank Martin Marek, Andres Potapczynski, and Sanyam Kapoor, for helpful discussions. This work was supported by Google's TPU Research Cloud (TRC) program: https://sites.research.google/trc/. SQ thanks the support of the Two Sigma PhD Fellowship.

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

# A Update Rules For Matrix-Preconditioned Optimizers

This section presents the update rules for the optimizers considered in our work. Let $W_t \in \mathbb{R}^{m \times n}$ denote the weight matrix of a layer at time $t$, and let $G_t \in \mathbb{R}^{m \times n}$ denote its gradient. We use $w_t \in \mathbb{R}^{mn}$ and $g_t \in \mathbb{R}^{mn}$ to represent the flattened versions of the weights and gradients, respectively. The hyperparameters include the learning rate ($\eta$), momentum coefficients ($\beta$), regularization factors ($\epsilon$), and inverse exponents ($e$). Elementwise multiplication is denoted by $\odot$, and vector division is applied elementwise.

**Adam** [15] adjusts the magnitude of the first moment by the second moment. The update rule is:

$$g_t \leftarrow \nabla_w \mathcal{L}(w_{t-1}) \tag{6}$$

$$m_t \leftarrow \beta_1 m_{t-1} + (1 - \beta_1) g_t \tag{7}$$

$$v_t \leftarrow \beta_2 v_{t-1} + (1 - \beta_2) g_t \odot g_t \tag{8}$$

$$\hat{m}_t \leftarrow \frac{m_t}{1 - \beta_1^t} \tag{9}$$

$$\hat{v}_t \leftarrow \frac{v_t}{1 - \beta_2^t} \tag{10}$$

$$w_t \leftarrow w_{t-1} - \eta \frac{\hat{m}_t}{\sqrt{\hat{v}_t} + \epsilon} \tag{11}$$

**Shampoo** [19] preconditions the gradient with a Kronecker-factored preconditioner. The update rule is:

$$M_t \leftarrow \beta_1 M_{t-1} + (1 - \beta_1) G_t \tag{12}$$

$$L_t \leftarrow \beta_2 L_{t-1} + (1 - \beta_2) G_t G_t^\top \tag{13}$$

$$R_t \leftarrow \beta_2 R_{t-1} + (1 - \beta_2) G_t^\top G_t \tag{14}$$

$$\hat{L}_t \leftarrow \frac{L_t}{(1 - \beta_2^t)} \tag{15}$$

$$\hat{R}_t \leftarrow \frac{R_t}{(1 - \beta_2^t)} \tag{16}$$

$$W_{t+1} \leftarrow W_t - \eta (\hat{L}_t + \epsilon I)^{-e_L} M_t (\hat{R}_t + \epsilon I)^{-e_R} \tag{17}$$

$$\tag{18}$$

**SOAP** [47] runs Adam in a rotated space. The update rule is:

$$M_t \leftarrow \beta_1 M_{t-1} + (1 - \beta_1) G_t \tag{19}$$

$$L_t \leftarrow \beta_2 L_{t-1} + (1 - \beta_2) G_t G_t^\top \tag{20}$$

$$R_t \leftarrow \beta_2 R_{t-1} + (1 - \beta_2) G_t^\top G_t \tag{21}$$

$$Q_t^L \leftarrow \text{Eigenvectors}(L_t) \tag{22}$$

$$Q_t^R \leftarrow \text{Eigenvectors}(R_t) \tag{23}$$

$$G_t' \leftarrow \left(Q_t^L\right)^\top G_t Q_t^R \tag{24}$$

$$M_t' \leftarrow \left(Q_t^L\right)^\top M_t Q_t^R \tag{25}$$

$$V_t \leftarrow \beta_2 V_{t-1} + (1 - \beta_2) G_t' \odot G_t' \tag{26}$$

$$W_{t+1} \leftarrow W_t - \eta Q_t^L \frac{M_t'}{\sqrt{V_t} + \epsilon} \left(Q_t^R\right)^\top \tag{27}$$

$$\tag{28}$$

**Muon** [23] utilizes Newton-Schulz to compute the matrix sign [12] of the gradient. The update rule is:

$$M_t \leftarrow \beta_1 M_{t-1} + (1 - \beta_1) G_t \tag{29}$$

$$W_{t+1} \leftarrow W_t - \eta \, \text{Newton-Schulz} \left( \frac{M_t}{\|M_t\|_F + \epsilon} \right) \tag{30}$$

$$\tag{31}$$

**AdaMuon** [44] simply applies Adam on top of Muon's orthogonalized gradient.

**Grafting** [3, 43] is a technique that adjusts the direction of a primary optimizer's update to match the step size of a reference optimizer with more stable magnitude. Let $Q_2(G)$ denote the update from the original optimizer, and $Q_1(G)$ the update from the reference (grafted) optimizer. The grafted update, denoted as $Q_1 \# Q_2$, is given by:

$$W_{t+1} \leftarrow W_t - \eta \frac{\|Q_1(G)\|_F}{\|Q_2(G)\|_F + \epsilon} Q_2(G) \tag{32}$$

**Blocking** [43, 47] is a technique that partitions the weight matrix into sub-blocks, and updates are computed independently for each block.

## B  Derivations of Maximum Update Parameterization

Table 2: Scaling rules for the per-layer learning rate $\eta$ and damping parameter(s) $\epsilon$ as a function of the weight matrix size $(d_{\text{in}}, d_{\text{out}})$, depth $L$, block size $(b_{\text{in}}, b_{\text{out}})$ and number of blocks $n_{\text{blk}}$ if using blocking. A multiplier $1/L$ is assumed to apply to the output of every residual block. $e_{L,R}$ are positive exponents for Shampoo, and 0–1 indicators for preconditioning either side for SOAP. For parameters outside of the residual blocks (e.g. embedding and last layer), set $L = 1$. $\eta_{Q_1}$ denotes the correctly scaled learning rate for optimizer $Q_1$. The $\epsilon$ for AdaMuon refers to the one used in the denominator when applying Adam on top of the orthogonalized gradient.

| | **Shampoo** $(e_L, e_R)$ | **SOAP** $(e_L, e_R)$ | **Muon** | **AdaMuon** | **Grafting** $Q_1 \# Q_2$ |
|---|---|---|---|---|---|
| $\eta$ | $\dfrac{(d_{\text{out}}/d_{\text{in}})^{1-(e_L+e_R)}}{L^{2(e_L+e_R)-1} n_{\text{blk}}^{e_L+e_R}}$ | $\dfrac{b_{\text{out}}^{e_L/2} b_{\text{in}}^{e_R/2}}{d_{\text{in}}}$ | $\sqrt{\dfrac{d_{\text{out}}}{d_{\text{in}}}}$ | $\dfrac{1}{d_{\text{in}}}$ | $\eta_{Q_1}$ |
| $\epsilon$ | $\dfrac{d_{\text{in}}}{L^2 d_{\text{out}} n_{\text{blk}}}$ | $\dfrac{b_{\text{out}}^{e_L/2} b_{\text{in}}^{e_R/2}}{L d_{\text{out}}}$ | $\dfrac{1}{L}\sqrt{\dfrac{d_{\text{in}}}{d_{\text{out}}}}$ | $\sqrt{\dfrac{1}{d_{\text{in}} d_{\text{out}}}}$ | $\dfrac{1}{\eta_{Q_2}}\sqrt{\dfrac{d_{\text{out}}}{d_{\text{in}}}}$ |

### B.1  Initialization

Before discussing specialization to any particular optimizer, we make the observation that the initialization scale $\sigma_\ell$ of $W_\ell$ is independent of the optimizer, with $\sigma_\ell = \Theta(1/\sqrt{d_{\text{in},\ell}})$ for $\ell < L$ and $\sigma_L = \Theta(1/d)$. This result follows from the combination of stability and feature learning condition of $\mu$P. Specifically, to ensure stability at initialization, $\mu$P requires all activations in the hidden layers have $\Theta(1)$ entries and that the function output is $O(1)$:

$$h_\ell(\xi') = W_\ell x_{\ell-1}(\xi') = \Theta(1), \, \ell = 1, \ldots, L-1, \quad f(\xi') = O(1) \tag{33}$$

The first condition implies that $W_\ell$ have entries of scale $\sigma_\ell = \Theta(1/\sqrt{d_{\text{in},\ell}})$ for all but the last layer, where $d_{\text{in},\ell}$ is the input dimension of layer $\ell$ (in our scalar-input setup, $d_{\text{in},1} = 1$ and $d_{\text{in},\ell} = d$ for $1 < \ell < L$), while the second condition implies that $\sigma_L = O(1/\sqrt{d})$. To then ensure non-negligible feature learning, i.e., the change in the last layer feature $\Delta x_{L-1}(\xi')$ is $\Theta(1)$ per step, while ensuring the resulting change in the output $\Delta f(\xi')$ is $\Theta(1)$ (predictions are updated without diverging), the last layer weights $w_L$ must be $\Theta(1/d)$. To see that, note $\Delta f(\xi') = w_L^\top \Delta x_{L-1}$ is a sum of $d$ i.i.d. random variables, each with a generically non-zero mean due to the correlation between elements in $w_L$ and elements in $\Delta x_{L-1}$. The correlation is induced by backpropagation, causing $\Delta x_{L-1}$ to be strongly aligned with $w_L$. This will become clear in Appendix B.8. Therefore, given $\Delta x_{L-1}$ is $\Theta(1)$, $w_L$ and thus $\sigma_L$ must be $\Theta(1/d)$ by the Law of Large Numbers (LLN), if maximally initialized[5].

---

[5] $\sigma_L = 0$ is also allowed and in fact common, but requires carrying out the analysis to the 2nd gradient step.

Moving forward, we omit layer subscripts and abbreviate $x(\xi)$ as $x$, $x(\xi')$ as $x'$, $\delta(\xi)$ as $\delta$, and $G(\xi)$ as $G$. Recall from Section 3.1 that at initialization we have

$$\delta = \Theta\left(\tfrac{1}{d_{\text{out}}}\right), \quad x = \Theta(1), \quad x^\top x' = \Theta(d_{\text{in}}), \quad x^\top x = \Theta(d_{\text{in}}), \quad \delta^\top \delta = \Theta\left(\tfrac{1}{d_{\text{out}}}\right). \tag{34}$$

The gradient for a single datapoint is

$$G = \nabla_W \mathcal{L}(\xi) = \delta x^\top \in \mathbb{R}^{d_{\text{out}} \times d_{\text{in}}}. \tag{35}$$

The $\mu$P condition from Section 3.1 can be restated as follows: for an optimizer that returns update $Q(G)$, we want the per-layer learning rate $\eta$ such that

> Choose learning rate $\eta$ such that
> $$\eta Q(G)x' = \Theta(1), \quad \text{where } G = \delta x^\top, \ \delta = \Theta\left(\tfrac{1}{d_{\text{out}}}\right), \ x = \Theta(1), \ x^\top x' = \Theta(d_{\text{in}}). \tag{36}$$

Solving for $\eta$ and any additional hyperparameters (e.g. damping) is now just algebra plus bookkeeping of how quantities scale with $d_{\text{in}}$ and $d_{\text{out}}$.

## B.2 Warmup: SGD and Adam

We start by rederiving $\mu$P for SGD and Adam in this notation.

**SGD.** For SGD, $Q(G) = G$, so

$$\eta Q(G)x' = \eta G x' \tag{37}$$
$$= \eta\, \delta\left(x^\top x'\right). \tag{38}$$

Since $x^\top x' = \Theta(d_{\text{in}})$ and each coordinate of $\delta$ is $\Theta\left(\tfrac{1}{d_{\text{out}}}\right)$, the vector $Gx'$ has entries of size $\Theta\left(\tfrac{d_{\text{in}}}{d_{\text{out}}}\right)$, i.e.

$$Gx' = \Theta\left(\tfrac{d_{\text{in}}}{d_{\text{out}}}\right). \tag{39}$$

The $\mu$P condition $\eta G x' = \Theta(1)$ therefore requires

$$\eta = \Theta\left(\tfrac{d_{\text{out}}}{d_{\text{in}}}\right), \tag{40}$$

recovering the result in Yang and Hu [52], Yang et al. [56]. The whole derivation is just one algebraic step once the scalings of $x$, $x'$, and $\delta$ are known.

**SignSGD and Adam.** For SignSGD, the elementwise preconditioner is

$$Q(G) = \left(\sqrt{G^{\odot 2}} + \epsilon\right)^{-1} \odot G, \tag{41}$$

where $\odot$ denotes elementwise multiplication and the square/square root are also taken elementwise. Since $G_{ij} = \delta_i x_j$ and

$$\delta_i = \Theta\left(\tfrac{1}{d_{\text{out}}}\right), \quad x_j = \Theta(1), \tag{42}$$

we have

$$|G_{ij}| = \Theta\left(\tfrac{1}{d_{\text{out}}}\right). \tag{43}$$

To keep the elementwise factor

$$\frac{G_{ij}}{|G_{ij}| + \epsilon} \tag{44}$$

nontrivial in the large-width limit, we need $\epsilon$ to be of the same order as $|G_{ij}|$, so we reparameterize

$$\epsilon = \frac{\epsilon'}{d_{\text{out}}}, \quad \epsilon' = \Theta(1). \tag{45}$$

With this choice, each entry of $Q(G)$ is an $\Theta(1)$ function of $(\delta_i, x_j)$ with no residual dependence on $d_{\text{in}}$ or $d_{\text{out}}$. As a result, each row of $Q(G)$ has an inner product with $x'$ of order

$$Q(G)x' = \Theta(d_{\text{in}}), \tag{46}$$

by an LLN argument. Hence

$$\eta Q(G)x' = \Theta(\eta d_{\text{in}}), \tag{47}$$

and we need

$$\eta = \Theta\!\left(\tfrac{1}{d_{\text{in}}}\right). \tag{48}$$

Yang and Littwin [53] formalize this using the `OuterNonlin` instruction and show that $Q(G)x'$ has an $\Theta(1)$ infinite-width limit after factoring out the explicit $d_{\text{in}}$.

The same scaling holds for Adam: accumulating the first and second moments of the gradient over steps does not introduce new powers of $d_{\text{in}}$ or $d_{\text{out}}$, so Adam also requires $\eta = \Theta\!\left(\tfrac{1}{d_{\text{in}}}\right)$.

## B.3 Shampoo and Muon

For matrix-preconditioned optimizers we only need to additionally track how the preconditioners scale with width. As in Section 3.1, we ignore preconditioner accumulation and gradient momentum, which do not affect width scaling [53] in the infinite-width limit.

For Shampoo, the single-sample left and right preconditioners are

$$L = \delta x^\top x \delta^\top = \left(x^\top x\right)\delta\delta^\top, \tag{49}$$

$$R = x\delta^\top \delta x^\top = \left(\delta^\top \delta\right) xx^\top, \tag{50}$$

and the update is

$$Q_{\text{Shampoo}}(G) = (L + \epsilon_L I)^{-e_L}\,\delta x^\top\,(R + \epsilon_R I)^{-e_R}. \tag{51}$$

Define

$$\lambda_x \equiv x^\top x = \Theta(d_{\text{in}}), \quad \lambda_\delta \equiv \delta^\top \delta = \Theta\!\left(\tfrac{1}{d_{\text{out}}}\right), \quad \lambda \equiv \lambda_x \lambda_\delta = \Theta\!\left(\tfrac{d_{\text{in}}}{d_{\text{out}}}\right). \tag{52}$$

The vector $\delta$ is an eigenvector of $L$ with eigenvalue $\lambda$, and $x$ is an eigenvector of $R$ with the same eigenvalue $\lambda$. Hence

$$(L + \epsilon_L I)^{-e_L}\delta = (\lambda + \epsilon_L)^{-e_L}\delta, \tag{53}$$

$$x^\top(R + \epsilon_R I)^{-e_R} = (\lambda + \epsilon_R)^{-e_R}x^\top, \tag{54}$$

and therefore

$$Q_{\text{Shampoo}}(G) = (\lambda + \epsilon_L)^{-e_L}(\lambda + \epsilon_R)^{-e_R}\,\delta x^\top. \tag{55}$$

As before, we parameterize damping in units of the nonzero eigenvalue:

$$\epsilon_R = \epsilon'_B\lambda, \quad \epsilon_L = \epsilon'_A\lambda, \quad \epsilon'_A, \epsilon'_B = \Theta(1), \tag{56}$$

so that

$$(\lambda + \epsilon_L)^{-e_L} = \lambda^{-e_L}(1 + \epsilon'_A)^{-e_L} = \Theta\!\left(\lambda^{-e_L}\right), \tag{57}$$

$$(\lambda + \epsilon_R)^{-e_R} = \lambda^{-e_R}(1 + \epsilon'_B)^{-e_R} = \Theta\!\left(\lambda^{-e_R}\right). \tag{58}$$

Thus

$$Q_{\text{Shampoo}}(G) = \Theta\!\left(\lambda^{-(e_L + e_R)}\right)\delta x^\top. \tag{59}$$

Applying $Q_{\text{Shampoo}}(G)$ to $x'$ gives

$$Q_{\text{Shampoo}}(G)x' = \Theta\!\left(\lambda^{-(e_L + e_R)}\right)\delta\left(x^\top x'\right) \tag{60}$$

$$= \Theta\!\left(\lambda^{-(e_L + e_R)}\right)\Theta\!\left(\tfrac{1}{d_{\text{out}}}\right)\Theta(d_{\text{in}}) \tag{61}$$

$$= \Theta\!\left(\left(\tfrac{d_{\text{in}}}{d_{\text{out}}}\right)^{1 - e_L - e_R}\right), \tag{62}$$

since $\lambda = \Theta\left(\frac{d_{\mathrm{in}}}{d_{\mathrm{out}}}\right)$. Therefore the learning rate should scale as

$$\eta = \Theta\left(\left(\frac{d_{\mathrm{out}}}{d_{\mathrm{in}}}\right)^{1-e_L-e_R}\right). \tag{63}$$

**As a Tensor Program.** Going one step further, we can expand the preconditioners to write

$$(L + \epsilon_L I)^{-e_L} = \epsilon_L^{-e_L}(I - P_\delta) + (\lambda + \epsilon_L)^{-e_L} P_\delta, \tag{64}$$

$$(R + \epsilon_R I)^{-e_R} = \epsilon_R^{-e_R}(I - P_x) + (\lambda + \epsilon_R)^{-e_R} P_x, \tag{65}$$

where $P_\delta = \delta\delta^\top/(\delta^\top\delta)$ and $P_x = xx^\top/(x^\top x)$ are rank–1 projectors. This expresses Shampoo purely in terms of vectors and scalars (deterministic numbers as $D \to \infty$). Therefore, Shampoo can be written in terms of instructions supported in the Tensor Program [53]. We show a similar construction is possible even if batch size is larger than 1 in Appendix B.10. This is a much stronger statement than features being updated at a $\Theta(1)$ rate in width, as it shows that the training dynamics of models trained with Shampoo with $\mu$P scaling admit well-defined, deterministic infinite-width limits, by the Master Theorem [53]. This is why we can expect the loss curves in Figure 1 (left) to be highly consistent across widths.

**Blocking.** Blocking reduces the cost of second-order preconditioners by partitioning $W \in \mathbb{R}^{d_{\mathrm{out}} \times d_{\mathrm{in}}}$ into $n_{\mathrm{out}} \times n_{\mathrm{in}}$ blocks of size $b_{\mathrm{out}} \times b_{\mathrm{in}}$, where

$$n_{\mathrm{in}} = \frac{d_{\mathrm{in}}}{b_{\mathrm{in}}}, \quad n_{\mathrm{out}} = \frac{d_{\mathrm{out}}}{b_{\mathrm{out}}}. \tag{66}$$

Let $G_{ij} \in \mathbb{R}^{b_{\mathrm{out}} \times b_{\mathrm{in}}}$ denote the $(i, j)$-th block of the gradient,

$$G_{ij} = \delta_i x_j^\top, \tag{67}$$

where $\delta_i \in \mathbb{R}^{b_{\mathrm{out}}}$ and $x_j \in \mathbb{R}^{b_{\mathrm{in}}}$ are the $i$-th and $j$-th chunks of $\delta$ and $x$, respectively, and similarly $x_j' \in \mathbb{R}^{b_{\mathrm{in}}}$ is the $j$-th chunk of $x'$.

Shampoo is applied independently to each block. The blockwise preconditioners are

$$L_{ij} = \delta_i x_j^\top x_j \delta_i^\top = \left(x_j^\top x_j\right)\delta_i\delta_i^\top, \tag{68}$$

$$R_{ij} = x_j\delta_i^\top \delta_i x_j^\top = \left(\delta_i^\top \delta_i\right)x_j x_j^\top. \tag{69}$$

Define the block eigenvalue

$$\lambda_{ij} \equiv \left(x_j^\top x_j\right)\left(\delta_i^\top \delta_i\right). \tag{70}$$

As before, $\delta_i$ is an eigenvector of $L_{ij}$ and $x_j$ is an eigenvector of $R_{ij}$ with eigenvalue $\lambda_{ij}$. The Shampoo update on block $G_{ij}$ is

$$Q(G_{ij}) = (L_{ij} + \epsilon_L I)^{-e_L} \delta_i x_j^\top (R_{ij} + \epsilon_R I)^{-e_R}, \tag{71}$$

so

$$Q(G_{ij})x_j' = (\lambda_{ij} + \epsilon_L)^{-e_L}(\lambda_{ij} + \epsilon_R)^{-e_R} \delta_i\left(x_j^\top x_j'\right). \tag{72}$$

The inner products now have scales,

$$x_j^\top x_j = \Theta(b_{\mathrm{in}}), \quad \delta_i^\top \delta_i = \Theta\left(\frac{b_{\mathrm{out}}}{d_{\mathrm{out}}^2}\right), \tag{73}$$

so we get

$$\lambda_{ij} = \Theta\left(\frac{b_{\mathrm{in}} b_{\mathrm{out}}}{d_{\mathrm{out}}^2}\right). \tag{74}$$

We again parameterize damping as $\epsilon_L = \epsilon_A' \lambda_{ij}$ and $\epsilon_R = \epsilon_B' \lambda_{ij}$ with $\epsilon_A', \epsilon_B' = \Theta(1)$, which yields

$$Q(G_{ij})x_j' = \Theta\left(\lambda_{ij}^{-(e_L+e_R)}\right)\delta_i\left(x_j^\top x_j'\right). \tag{75}$$

The $i$-th chunk of the feature update is then

$$\Delta h_i = \eta \sum_{j=1}^{n_{\text{in}}} Q(G_{ij}) x_j' \tag{76}$$

$$= \eta\, \delta_i \sum_{j=1}^{n_{\text{in}}} \Theta\!\left( \lambda_{ij}^{-(e_L+e_R)} \right) (x_j^\top x_j') \tag{77}$$

$$= \eta\, \delta_i \sum_{j=1}^{n_{\text{in}}} \underbrace{\Theta\!\left( \lambda_{ij}^{-(e_L+e_R)} b_{\text{in}} \right)}_{c_{ij}}. \tag{78}$$

where $\{c_{ij}\}_j$ are i.i.d. (due to permutation symmetry over $j$) with a generally nonzero mean per entry, and we used $x_j^\top x_j' = \Theta(b_{\text{in}})$. Therefore, recalling $\delta_i = \Theta(1/d_{\text{out}})$ and the scale of $\lambda_{ij}$, we have

$$\Delta h_i = \Theta\!\left( \eta \left( \frac{b_{\text{in}} b_{\text{out}}}{d_{\text{out}}^2} \right)^{-(e_L+e_R)} \frac{b_{\text{in}}}{d_{\text{out}}} n_{\text{in}} \right) \tag{79}$$

$$= \Theta\!\left( \eta \left( \frac{b_{\text{in}} b_{\text{out}}}{d_{\text{out}}^2} \right)^{-(e_L+e_R)} \frac{d_{\text{in}}}{d_{\text{out}}} \right), \tag{80}$$

since $n_{\text{in}} = d_{\text{in}}/b_{\text{in}}$. Enforcing $\Delta h_i = \Theta(1)$ leads to

$$\eta = \Theta\!\left( \frac{d_{\text{out}}}{d_{\text{in}}} \left( \frac{b_{\text{in}} b_{\text{out}}}{d_{\text{out}}^2} \right)^{e_L+e_R} \right). \tag{81}$$

Equivalently,

$$\eta = \Theta\!\left( \left( \frac{d_{\text{out}}}{d_{\text{in}}} \right)^{1-(e_L+e_R)} (n_{\text{in}} n_{\text{out}})^{-(e_L+e_R)} \right), \tag{82}$$

since $n_{\text{in}} n_{\text{out}} = \frac{d_{\text{in}}}{b_{\text{in}}} \frac{d_{\text{out}}}{b_{\text{out}}}$. When $b_{\text{in}} = d_{\text{in}}$ and $b_{\text{out}} = d_{\text{out}}$, we recover the full Shampoo scaling $\eta = \Theta\!\left( \frac{d_{\text{out}}}{d_{\text{in}}} \right)^{1-e_L-e_R}$. When $e_L + e_R = \frac{1}{2}$ and $b_{\text{in}} = b_{\text{out}} = 1$, Shampoo degenerates to Adam and we get the correct Adam scaling $\eta = \Theta\!\left( \frac{1}{d_{\text{in}}} \right)$.

## B.4   Muon

Muon [23] applies Newton–Schulz to a normalized gradient to approximate the matrix sign. In the idealized limit of zero damping, full Shampoo with $e_L = e_R = 1/4$ yields the same matrix-sign direction; with nonzero damping, Shampoo and Muon differ only by a scalar nonlinearity applied to the singular values, so the Shampoo learning-rate scaling carries over. We now verify this directly for a rank–1 gradient.

Write $G = \delta x^\top = u\sigma v^\top$, where $u = \delta/\|\delta\|_2$, $v = x/\|x\|_2$, and

$$\sigma = \|\delta\|_2 \|x\|_2 = \Theta\!\left( \sqrt{\frac{d_{\text{in}}}{d_{\text{out}}}} \right). \tag{83}$$

Muon forms

$$\widetilde{G} = \frac{G}{\|G\|_F + \epsilon} = u\widetilde{\sigma} v^\top, \quad \widetilde{\sigma} = \frac{\sigma}{\sigma + \epsilon}. \tag{84}$$

Choosing $\epsilon = \Theta\!\left( \sqrt{\frac{d_{\text{in}}}{d_{\text{out}}}} \right)$ keeps $\widetilde{\sigma} = \Theta(1)$.

The Newton–Schulz iteration used in Muon has the form

$$Y_{k+1} = \frac{1}{2} Y_k \left( 3I - Y_k^\top Y_k \right), \quad Y_0 = \widetilde{G}, \tag{85}$$

so for rank–1 $\widetilde{G}$ we can check inductively that $Y_k = u s_k v^\top$ with the scalar recursion

$$s_{k+1} = \frac{1}{2} s_k (3 - s_k^2), \quad s_0 = \widetilde{\sigma}. \tag{86}$$

For any fixed $k = \Theta(1)$, this gives $s_k = \Theta(1)$, and therefore

$$\text{Newton-Schulz}(\widetilde{G}) = \Theta(1)\, uv^\top. \tag{87}$$

Indeed, this holds regardless of the coefficients used in the Newton–Schulz. Thus

$$Q_{\text{Muon}}(G)x' = \Theta(1)\, u\big(v^\top x'\big) = \Theta(1)\, \frac{\delta}{\|\delta\|_2}\, \frac{x^\top x'}{\|x\|_2} = \Theta\left(\sqrt{\frac{d_{\text{in}}}{d_{\text{out}}}}\right), \tag{88}$$

and the $\mu$P condition $\eta Q_{\text{Muon}}(G)x' = \Theta(1)$ yields

$$\eta = \Theta\left(\sqrt{\frac{d_{\text{out}}}{d_{\text{in}}}}\right). \tag{89}$$

Finally, the normalization requires

$$\epsilon = \Theta(\|G\|_F) = \Theta\left(\sqrt{\frac{d_{\text{in}}}{d_{\text{out}}}}\right). \tag{90}$$

## B.5 AdaMuon

AdaMuon [44] applies Adam on top of Muon's Newton–Schulz output $O \equiv \text{Newton-Schulz}(\widetilde{G})$. From the previous subsection, $O$ is rank–1 and has entries of size

$$|O_{ij}| = \Theta\left(\frac{1}{\sqrt{d_{\text{in}}d_{\text{out}}}}\right). \tag{91}$$

AdaMuon then applies an Adam-like elementwise rescaling to $O$. To keep the map $t \mapsto t/(|t| + \epsilon)$ nontrivial as width grows, we take

$$\epsilon = \frac{\epsilon'}{\sqrt{d_{\text{in}}d_{\text{out}}}}, \quad \epsilon' = \Theta(1), \tag{92}$$

so each entry of $Q_{\text{AdaMuon}}(G)$ is an $\Theta(1)$ function of the corresponding entry of $O$, with no residual scaling in $(d_{\text{in}}, d_{\text{out}})$. By the same LLN argument as in the Adam derivation, this implies

$$Q_{\text{AdaMuon}}(G)x' = \Theta(d_{\text{in}}), \tag{93}$$

and hence the $\mu$P condition $\eta Q_{\text{AdaMuon}}(G)x' = \Theta(1)$ gives

$$\eta = \Theta\left(\frac{1}{d_{\text{in}}}\right). \tag{94}$$

(Here $\epsilon$ is the one used in the Adam-on-top denominator; Muon's normalization $\epsilon$ is as in the previous subsection.)

## B.6 Grafting

When using $Q_2$ with $Q_1$ learning rate grafting, the update is

$$Q(G) = \frac{\|Q_1(G)\|_F}{\|Q_2(G)\|_F + \epsilon} Q_2(G). \tag{95}$$

Assuming both $Q_1(G)$ and $Q_2(G)$ have $\Theta(1)$ rank, which we have shown for Shampoo but generalization to other optimizers in this work is straightforward, $\|Q_1(G)\|_F \sim \|Q_1(G)\|_2$ and $\|Q_2(G)\|_F \sim \|Q_2(G)\|_2$. Let $\eta_{Q_2}$ denote the $\mu$P learning rate for $Q_2$, by construction $\eta_{Q_2}\|Q_2(G)\|_2\|x'\|_2 \sim \sqrt{d_{\text{out}}}$, due to the alignment between $Q_2(G)$ and $x'$, which shows $\|Q_2(G)\|_F \sim \frac{1}{\eta_{Q_2}}\sqrt{\frac{d_{\text{out}}}{d_{\text{in}}}}$. $\epsilon$ should match the scale of $\|Q_2(G)\|_F$, so

$$\epsilon = \Theta\left(\frac{1}{\eta_{Q_2}}\sqrt{\frac{d_{\text{out}}}{d_{\text{in}}}}\right). \tag{96}$$

after which scaling by $\|Q_1(G)\|_F$ sets $Q(G)x' \sim Q_1(G)x'$, so learning rate should now follow $\eta_{Q_1}$.

## B.7 SOAP

SOAP applies an Adam-like elementwise preconditioner in the eigenbasis of Shampoo's left and right preconditioners. Let $U \in \mathbb{R}^{d_{\text{out}} \times d_{\text{out}}}$ and $V \in \mathbb{R}^{d_{\text{in}} \times d_{\text{in}}}$ be orthogonal matrices whose first columns are the normalized eigenvectors of $L$ and $R$:

$$u_1 = \frac{\delta}{\|\delta\|}, \quad v_1 = \frac{x}{\|x\|}, \tag{97}$$

with $\|\delta\| = \sqrt{\delta^\top \delta}$ and $\|x\| = \sqrt{x^\top x}$. Then

$$U^\top \delta = \|\delta\| \, e_1^U, \tag{98}$$

$$V^\top x = \|x\| \, e_1^V, \tag{99}$$

where $e_1^U$ and $e_1^V$ are the first standard basis vectors in $\mathbb{R}^{d_{\text{out}}}$ and $\mathbb{R}^{d_{\text{in}}}$.

Transforming the gradient into this eigenbasis,

$$G' = U^\top G V = U^\top \delta x^\top V = \|\delta\| \|x\| \, e_1^U e_1^{V\top}. \tag{100}$$

Thus $G'$ has a single nonzero entry at $(1,1)$ of magnitude

$$g \equiv \|\delta\| \|x\| = \sqrt{\delta^\top \delta} \sqrt{x^\top x} = \Theta\left(\sqrt{\frac{d_{\text{in}}}{d_{\text{out}}}}\right). \tag{101}$$

SOAP applies the Adam-like rule elementwise to $G'$:

$$\text{Adam}(G') = \left(\sqrt{G'^{\odot 2}} + \epsilon\right)^{-1} \odot G'. \tag{102}$$

Since $G'$ is zero everywhere except at $(1,1)$, we have

$$\text{Adam}(G') = \frac{g}{g + \epsilon} \, e_1^U e_1^{V\top}. \tag{103}$$

To keep this factor $\frac{g}{g+\epsilon}$ nontrivial as width grows, we choose

$$\epsilon = g \, \epsilon' = \Theta\left(\sqrt{\frac{d_{\text{in}}}{d_{\text{out}}}}\right), \quad \epsilon' = \Theta(1), \tag{104}$$

so that $\frac{g}{g+\epsilon} = \frac{1}{1+\epsilon'} = \Theta(1)$.

Transforming back to the original basis,

$$Q_{\text{SOAP}}(G) = U \, \text{Adam}(G') V^\top \tag{105}$$

$$= \frac{g}{g + \epsilon} U e_1^U e_1^{V\top} V^\top \tag{106}$$

$$= \frac{g}{g + \epsilon} \frac{\delta}{\|\delta\|} \frac{x^\top}{\|x\|} \tag{107}$$

$$= \frac{1}{1 + \epsilon'} \frac{\delta x^\top}{\|\delta\| \|x\|}. \tag{108}$$

Using $\|\delta\| = \Theta\left(\frac{1}{\sqrt{d_{\text{out}}}}\right)$ and $\|x\| = \Theta(\sqrt{d_{\text{in}}})$, we obtain

$$Q_{\text{SOAP}}(G) = \Theta\left(\sqrt{\frac{d_{\text{out}}}{d_{\text{in}}}}\right) \delta x^\top. \tag{109}$$

Applying this to $x'$,

$$Q_{\text{SOAP}}(G)x' = \Theta\left(\sqrt{\frac{d_{\text{out}}}{d_{\text{in}}}}\right) \delta \left(x^\top x'\right) \tag{110}$$

$$= \Theta\left(\sqrt{\frac{d_{\text{out}}}{d_{\text{in}}}}\right) \Theta\left(\frac{1}{d_{\text{out}}}\right) \Theta(d_{\text{in}}) \tag{111}$$

$$= \Theta\left(\sqrt{\frac{d_{\text{in}}}{d_{\text{out}}}}\right). \tag{112}$$

Therefore,

$$\eta Q_{\text{SOAP}}(G)x' = \Theta(1) \quad \implies \quad \eta = \Theta\left(\sqrt{\tfrac{d_{\text{out}}}{d_{\text{in}}}}\right), \tag{113}$$

and SOAP's damping should scale as $\epsilon = \Theta\left(\sqrt{\tfrac{d_{\text{in}}}{d_{\text{out}}}}\right)$.

**Blocking.** As in the Shampoo blocking derivation, partition $W$ (and hence $G$) into $n_{\text{out}} \times n_{\text{in}}$ blocks of size $b_{\text{out}} \times b_{\text{in}}$ so that

$$n_{\text{out}} = \frac{d_{\text{out}}}{b_{\text{out}}}, \quad n_{\text{in}} = \frac{d_{\text{in}}}{b_{\text{in}}}. \tag{114}$$

Let $\delta_i \in \mathbb{R}^{b_{\text{out}}}$ and $x_j \in \mathbb{R}^{b_{\text{in}}}$ denote the $i$-th row-chunk and $j$-th column-chunk respectively, so the $(i, j)$ block of the gradient is

$$G_{ij} = \delta_i x_j^\top \in \mathbb{R}^{b_{\text{out}} \times b_{\text{in}}}. \tag{115}$$

From the global scales above, at the block level we have

$$\|x_j\|_2 = \Theta\left(\sqrt{b_{\text{in}}}\right), \tag{116}$$

$$\|\delta_i\|_2 = \Theta\left(\sqrt{b_{\text{out}}}/d_{\text{out}}\right), \tag{117}$$

$$x_j^\top x_j' = \Theta(b_{\text{in}}), \tag{118}$$

where $x_j'$ is the $j$-th chunk of $x'$.

**Per-block Shampoo preconditioners for SOAP bases.** SOAP operates in the eigenbases of the per-block Shampoo preconditioners

$$L_{ij} = \delta_i x_j^\top x_j \delta_i^\top + \epsilon_B I_{b_{\text{out}}} = \|x_j\|_2^2 \delta_i \delta_i^\top + \epsilon_B I_{b_{\text{out}}}, \tag{119}$$

$$R_{ij} = x_j \delta_i^\top \delta_i x_j^\top + \epsilon_A I_{b_{\text{in}}} = \|\delta_i\|_2^2 x_j x_j^\top + \epsilon_A I_{b_{\text{in}}}. \tag{120}$$

Thus:

- $L_{ij}$ has one eigenvector

$$u_{i1} = \frac{\delta_i}{\|\delta_i\|_2} \tag{121}$$

  with eigenvalue $\|x_j\|_2^2 \|\delta_i\|_2^2 + \epsilon_B$, and $b_{\text{out}} - 1$ orthogonal eigenvectors with eigenvalue $\epsilon_B$.
- $R_{ij}$ has one eigenvector

$$v_{j1} = \frac{x_j}{\|x_j\|_2} \tag{122}$$

  with eigenvalue $\|\delta_i\|_2^2 \|x_j\|_2^2 + \epsilon_A$, and $b_{\text{in}} - 1$ orthogonal eigenvectors with eigenvalue $\epsilon_A$.

Let $U_i$ and $V_j$ be the orthogonal matrices whose first columns are $u_{i1}$ and $v_{j1}$, respectively. When a side is *not* tracked (one-sided SOAP), we simply take the corresponding basis to be the identity on that side.

**SOAP step inside the tracked eigenbasis.** In the per-block tracked basis, the gradient transforms as

$$G_{ij}' = U_i^\top G_{ij} V_j. \tag{123}$$

SOAP applies an elementwise Adam-like rescaling

$$H_{ij} = \left(\sqrt{G_{ij}' \odot G_{ij}'} + \epsilon \mathbf{1}\right)^{-1} \odot G_{ij}', \tag{124}$$

and then rotates back

$$Q(G_{ij}) = U_i H_{ij} V_j^\top. \tag{125}$$

The block's contribution to the hidden update on a second input $x'$ is

$$\Delta h_i^{(j)} = \eta\, Q(G_{ij}) x_j'. \tag{126}$$

We will compute the *scale* of $\Delta h_i^{(j)}$ for each tracking pattern and then sum over $j = 1, \ldots, n_{\mathrm{in}}$ to obtain $\Delta h_i = \sum_{j=1}^{n_{\mathrm{in}}} \Delta h_i^{(j)}$. The $\mu$P condition requires

$$\|\Delta h_i\|_2 = \Theta\!\left(\sqrt{b_{\mathrm{out}}}\right), \tag{127}$$

i.e. entries of $\Delta h_i$ are $\Theta(1)$.

**Cases: both-sided, right-only, left-only, neither tracked.** We use $e_L, e_R \in \{0,1\}$ as indicators for whether the left/right side is tracked. The "neither tracked" case ($e_L = e_R = 0$) recovers Adam-like scaling.

**Case 1: both sides tracked ($e_L = e_R = 1$).** Transforming the rank–1 block gradient,

$$G_{ij}' = U_i^\top (\delta_i x_j^\top) V_j = \|\delta_i\|_2 \, \|x_j\|_2 \, e_1^U e_1^{V\top}, \tag{128}$$

where $e_1^U$ and $e_1^V$ are the first standard basis vectors in the $U_i$ and $V_j$ coordinates. The unique nonzero entry has magnitude

$$s_{ij} = \|\delta_i\|_2 \, \|x_j\|_2 = \Theta\!\left(\frac{\sqrt{b_{\mathrm{out}}}}{d_{\mathrm{out}}} \sqrt{b_{\mathrm{in}}}\right) = \Theta\!\left(\frac{\sqrt{b_{\mathrm{out}} b_{\mathrm{in}}}}{d_{\mathrm{out}}}\right). \tag{129}$$

To keep the Adam nonlinearity nontrivial, we choose

$$\epsilon = \Theta\!\left(\frac{\sqrt{b_{\mathrm{out}} b_{\mathrm{in}}}}{d_{\mathrm{out}}}\right). \tag{130}$$

Then the Adam-like map sends $G_{ij}'$ to

$$H_{ij} = \Theta(1)\, e_1^U e_1^{V\top}. \tag{131}$$

Rotating back,

$$Q(G_{ij}) x_j' = U_i H_{ij} V_j^\top x_j' = U_i e_1^U \cdot \left(e_1^{V\top} V_j^\top x_j'\right) \tag{132}$$

$$= u_{i1} \cdot \left(v_{j1}^\top x_j'\right) \cdot \Theta(1) \tag{133}$$

$$= \frac{\delta_i}{\|\delta_i\|_2} \cdot \Theta\!\left(\frac{x_j^\top x_j'}{\|x_j\|_2}\right) \tag{134}$$

$$= \frac{\delta_i}{\|\delta_i\|_2} \cdot \Theta\!\left(\sqrt{b_{\mathrm{in}}}\right), \tag{135}$$

since $x_j^\top x_j' = \Theta(b_{\mathrm{in}})$ and $\|x_j\|_2 = \Theta(\sqrt{b_{\mathrm{in}}})$. Thus

$$\left\|\Delta h_i^{(j)}\right\|_2 = \eta\, \Theta\!\left(\sqrt{b_{\mathrm{in}}}\right). \tag{136}$$

Summing over $n_{\mathrm{in}} = d_{\mathrm{in}}/b_{\mathrm{in}}$ blocks along the input dimension,

$$\|\Delta h_i\|_2 = \eta\, n_{\mathrm{in}}\, \Theta\!\left(\sqrt{b_{\mathrm{in}}}\right) = \eta\, \Theta\!\left(\frac{d_{\mathrm{in}}}{\sqrt{b_{\mathrm{in}}}}\right). \tag{137}$$

Enforcing $\|\Delta h_i\|_2 = \Theta(\sqrt{b_{\mathrm{out}}})$ gives

$$\eta = \Theta\!\left(\frac{\sqrt{b_{\mathrm{out}} b_{\mathrm{in}}}}{d_{\mathrm{in}}}\right). \tag{138}$$

**Case 2: right-only tracked ($e_L = 0, e_R = 1$).** Here $U_i = I_{b_{\mathrm{out}}}$ and $V_j$ is as above. Then

$$G_{ij}' = G_{ij} V_j = \delta_i x_j^\top V_j = \delta_i \|x_j\|_2 \, e_1^{V\top}, \tag{139}$$

so the nonzero column has entries of magnitude $\Theta\!\left(\frac{\sqrt{b_{\mathrm{in}}}}{d_{\mathrm{out}}}\right)$. To keep the elementwise Adam operation nontrivial,

$$\epsilon = \Theta\!\left(\frac{\sqrt{b_{\mathrm{in}}}}{d_{\mathrm{out}}}\right). \tag{140}$$

Then the first column of $H_{ij}$ has $\Theta(1)$ entries. Multiplying by $V_j^\top x_j'$,

$$Q(G_{ij})x_j' = H_{ij}V_j^\top x_j' \tag{141}$$

$$= (\text{column with } \Theta(1) \text{ entries}) \cdot \left(v_{j1}^\top x_j'\right) \tag{142}$$

$$= \Theta(1) \cdot \Theta\left(\sqrt{b_{\text{in}}}\right), \tag{143}$$

so

$$\left\|\Delta h_i^{(j)}\right\|_2 = \eta\,\Theta\left(\sqrt{b_{\text{in}}b_{\text{out}}}\right). \tag{144}$$

Summing over $n_{\text{in}} = d_{\text{in}}/b_{\text{in}}$ blocks,

$$\|\Delta h_i\|_2 = \eta\,\Theta\left(n_{\text{in}}\sqrt{b_{\text{in}}b_{\text{out}}}\right) = \eta\,\Theta\left(\tfrac{d_{\text{in}}}{b_{\text{in}}}\sqrt{b_{\text{in}}b_{\text{out}}}\right) \tag{145}$$

$$= \eta\,\Theta\left(d_{\text{in}}\sqrt{\tfrac{b_{\text{out}}}{b_{\text{in}}}}\right). \tag{146}$$

Setting $\|\Delta h_i\|_2 = \Theta\left(\sqrt{b_{\text{out}}}\right)$ yields

$$\eta = \Theta\left(\tfrac{\sqrt{b_{\text{in}}}}{d_{\text{in}}}\right). \tag{147}$$

**Case 3: left-only tracked** ($e_L = 1, e_R = 0$). Now $U_i$ is as above and $V_j = I_{b_{\text{in}}}$. Then

$$G_{ij}' = U_i^\top G_{ij} = U_i^\top \delta_i x_j^\top = \|\delta_i\|_2\, e_1^U x_j^\top, \tag{148}$$

so the nonzero row has entries of magnitude $\Theta\left(\tfrac{\sqrt{b_{\text{out}}}}{d_{\text{out}}}\right)$. So we choose

$$\epsilon = \Theta\left(\tfrac{\sqrt{b_{\text{out}}}}{d_{\text{out}}}\right). \tag{149}$$

Then the first row of $H_{ij}$ has $\Theta(1)$ entries. Rotating back and multiplying $x_j'$,

$$Q(G_{ij})x_j' = U_i\left(e_1^U h_j^\top\right)x_j' \tag{150}$$

$$= U_i e_1^U \cdot \left(h_j^\top x_j'\right) \tag{151}$$

$$= \frac{\delta_i}{\|\delta_i\|_2} \cdot \Theta(b_{\text{in}}), \tag{152}$$

where $h_j$ is an elementwise transform of $x_j$ coming from Adam's nonlinearity and is aligned with $x_j$ by the Master Theorem [53], so $h_j^\top x_j' = \Theta(b_{\text{in}})$. Hence

$$\left\|\Delta h_i^{(j)}\right\|_2 = \eta\,\Theta(b_{\text{in}}). \tag{153}$$

Summing over $n_{\text{in}} = d_{\text{in}}/b_{\text{in}}$ blocks,

$$\|\Delta h_i\|_2 = \eta\,\Theta(n_{\text{in}}b_{\text{in}}) = \eta\,\Theta(d_{\text{in}}). \tag{154}$$

Enforcing $\|\Delta h_i\|_2 = \Theta\left(\sqrt{b_{\text{out}}}\right)$ gives

$$\eta = \Theta\left(\tfrac{\sqrt{b_{\text{out}}}}{d_{\text{in}}}\right). \tag{155}$$

**Case 4: neither side tracked** ($e_L = e_R = 0$; **Adam-like**). When neither side is tracked, no rotations are applied and we simply recover Adam:

$$\eta = \Theta\left(\tfrac{1}{d_{\text{in}}}\right),\ \epsilon = \Theta\left(\tfrac{1}{d_{\text{out}}}\right) \tag{156}$$

**Unified expressions.** Let $e_L, e_R \in \{0,1\}$ indicate whether the left/right sides are tracked (for two-sided SOAP take $e_L = e_R = 1$, for one-sided set exactly one of them to 1, and for Adam set both to 0). Define the *block concentration factor*

$$T_{\text{blk}}(e_L, e_R) \equiv b_{\text{out}}^{\ e_L/2}\, b_{\text{in}}^{\ e_R/2}. \tag{157}$$

Then the per-layer learning rate and the SOAP (Adam) $\epsilon$ *inside the eigenbasis* scale as

$$\eta = \Theta\left(\frac{T_{\text{blk}}(e_L, e_R)}{d_{\text{in}}}\right) = \Theta\left(\frac{b_{\text{out}}{}^{e_L/2} b_{\text{in}}{}^{e_R/2}}{d_{\text{in}}}\right), \tag{158}$$

$$\epsilon = \Theta\left(\frac{T_{\text{blk}}(e_L, e_R)}{d_{\text{out}}}\right) = \Theta\left(\frac{b_{\text{out}}{}^{e_L/2} b_{\text{in}}{}^{e_R/2}}{d_{\text{out}}}\right). \tag{159}$$

These specialize to:

- Adam ($e_L = e_R = 0$): $\eta = \Theta(1/d_{\text{in}})$, $\epsilon = \Theta(1/d_{\text{out}})$.
- Left-only SOAP ($e_L = 1, e_R = 0$): $\eta = \Theta(\sqrt{b_{\text{out}}}/d_{\text{in}})$, $\epsilon = \Theta(\sqrt{b_{\text{out}}}/d_{\text{out}})$.
- Right-only SOAP ($e_L = 0, e_R = 1$): $\eta = \Theta(\sqrt{b_{\text{in}}}/d_{\text{in}})$, $\epsilon = \Theta(\sqrt{b_{\text{in}}}/d_{\text{out}})$.
- Two-sided SOAP ($e_L = e_R = 1$): $\eta = \Theta(\sqrt{b_{\text{out}} b_{\text{in}}}/d_{\text{in}})$, $\epsilon = \Theta(\sqrt{b_{\text{out}} b_{\text{in}}}/d_{\text{out}})$.

## B.8 Alignment at Initialization and the Spectral Norm Condition for Matrix-Preconditioned Optimizers

Given our explicit expressions of the optimizer update purely in terms of vectors and scalars in Appendix B so far, it is easy to check that $Q(G)$ is always (a) aligned with $\delta$ on the left, and (b) aligned with $x'$ on the right. Property (a) explains why the last layer $w_L$ must have entrywise scale of $\Theta(1/\text{width})$ : explicit backpropagation shows $\delta$ is aligned with $w_L$, and the update to the last layer feature $\Delta h_{L-1}$ is $\Theta(1)$ and aligned with $\delta$ and thus $w_L$, which causes an update to the output of scale $\Delta f = \Theta(\text{width} \cdot 1 \cdot \sigma_L)$, requiring $\sigma_L = \Theta(1/\text{width})$ for stability. Property (b) shows that it is the spectral norm of $Q(G)$ that governs the scale of $Q(G)x'$. Thus, the spectral condition that $\eta\|Q(G)\|_2 = \Theta(\sqrt{d_{\text{out}}/d_{\text{in}}})$ [56] continues to hold for these matrix-preconditioned optimizers.

## B.9 Larger than One Batch Size and Number of Steps

So far we have analyzed the first gradient step with a batch size of one. Now we explain why increasing batch size $B$ and the number of steps $t$ does not change the asymptotic $\mu$P width scaling conclusions, as long as both $B$ and $t$ are $\Theta(1)$ with respect to width. We illustrate this with Shampoo and SOAP as concrete examples.

**Shampoo.** For a batch of size $B$, the (per-layer) gradient is

$$G = \frac{1}{B}\sum_{b=1}^{B} \delta^{(b)} x^{(b)\top}, \tag{160}$$

up to a $1/B$ factor that does not affect width exponents, where $x^{(b)} \in \mathbb{R}^{d_{\text{in}}}$ and $\delta^{(b)} \in \mathbb{R}^{d_{\text{out}}}$ are the activations and backpropagated signals for the $b$-th example in the batch. Under the same initialization and feature-kernel assumptions as in Section 3.1, each term

$$G^{(b)} \equiv \delta^{(b)} x^{(b)\top} \tag{161}$$

has the same width scaling as the single-example gradient. As a result, in particular, $G$ has $\Theta(1)$ non-zero singular values, all of size $\Theta(\sqrt{d_{\text{in}}/d_{\text{out}}})$ as before, and the corresponding left and right singular vectors are precisely the eigenvectors corresponding to the non-zero eigenvalues of $L = GG^\top$ and $R = G^\top G$. Therefore, the nonzero spectrum of $L$ and $R$ remains $\Theta(d_{\text{in}}/d_{\text{out}})$ and the number of nonzero eigenvalues and associated eigenvectors is $\Theta(1)$, independent of width, and aligned to non-zero singular vectors of $G$. As a result, all width-dependent scalings remain unchanged. For a completely rigorous treatment, we provide an explicit construction in Appendix B.10 for expressing Shampoo with batch size greater than one as a Tensor Program and show the learning-rate and $\epsilon$ scalings carry over.

**SOAP.** SOAP applies an Adam-like elementwise rescaling in the eigenbasis of the Shampoo preconditioner. In the single-sample analysis, we saw that in this basis, the transformed gradient $G'$ has only one nonzero entry with magnitude

$$\Theta\left(\sqrt{\frac{d_{\text{in}}}{d_{\text{out}}}}\right), \tag{162}$$

and that choosing

$$\epsilon = \Theta\left(\sqrt{\frac{d_{\text{in}}}{d_{\text{out}}}}\right) \tag{163}$$

keeps the elementwise mapping $t \mapsto t/(|t| + \epsilon)$ nontrivial. With batch size $B = \Theta(1)$, the number of nonzero entries in $G'$ is still $\Theta(1)$ since it only has support in the non-zero eigen-directions of the preconditioners $U, V$, and each entry remains at the same $\Theta\left(\sqrt{d_{\text{in}}/d_{\text{out}}}\right)$.

Therefore, the SOAP learning-rate and $\epsilon$ scalings derived in the single-sample case,

$$\eta = \Theta\left(\sqrt{\frac{d_{\text{out}}}{d_{\text{in}}}}\right), \quad \epsilon = \Theta\left(\sqrt{\frac{d_{\text{in}}}{d_{\text{out}}}}\right), \tag{164}$$

remain valid for $B = \Theta(1)$.

$t > 1$ **steps.** Similar reasoning shows why analyzing beyond $t = 1$ steps leads to the same asymptotic width-scaling conclusions, so long as $t = \Theta(1)$. In particular, accumulation in the preconditioners and momentum can be viewed as effectively increasing the batch size.

## B.10 Expressing Shampoo as a Tensor Program When Batch Size Exceeds One

We now show Shampoo can be expressed as a Tensor Program even when the batch size is larger than one. Ignoring accumulation, full Shampoo with $e_L = e_R = 1/4$ recovers the matrix-sign direction when $\epsilon \to 0$, and Muon applies Newton–Schulz to approximate the same direction; thus a similar Tensor Program construction applies to Muon.

For batch size $B$, the per-layer gradient is a sum of $B$ rank–1 terms,

$$G = \frac{1}{B} \sum_{b=1}^{B} \delta^{(b)} x^{(b)\top}. \tag{165}$$

Collect the batch vectors into matrices

$$\Delta \equiv \left[\delta^{(1)}, \ldots, \delta^{(B)}\right] \in \mathbb{R}^{d_{\text{out}} \times B}, \quad X \equiv \left[x^{(1)}, \ldots, x^{(B)}\right] \in \mathbb{R}^{d_{\text{in}} \times B}, \tag{166}$$

so that

$$G = \frac{1}{B} \Delta X^\top. \tag{167}$$

The only scalars we will ever need are inner products between these vectors. In particular, define the $B \times B$ Gram matrices

$$K_x \equiv X^\top X, \quad K_\delta \equiv \Delta^\top \Delta. \tag{168}$$

Ignoring EMA accumulation for the moment (for which a similar analysis applies), Shampoo uses

$$L = GG^\top, \quad R = G^\top G, \tag{169}$$

which become

$$L = \frac{1}{B^2} \Delta K_x \Delta^\top, \quad R = \frac{1}{B^2} X K_\delta X^\top. \tag{170}$$

This immediately implies $\text{rank}(L) \leq B$ and $\text{rank}(R) \leq B$, so when $B = \Theta(1)$ the nontrivial eigenspaces of $L$ and $R$ are $\Theta(1)$-dimensional.

To make the dependence on inner products explicit, assume for simplicity that $K_\delta$ and $K_x$ are invertible, and define the whitened matrices

$$\widetilde{\Delta} \equiv \Delta K_\delta^{-1/2}, \quad \widetilde{X} \equiv X K_x^{-1/2}, \tag{171}$$

then

$$\widetilde{\Delta}^\top \widetilde{\Delta} = I, \quad \widetilde{X}^\top \widetilde{X} = I. \tag{172}$$

If either Gram matrix is singular, one can instead interpret $K_\delta^{-1/2}$ and $K_x^{-1/2}$ as Moore–Penrose pseudoinverses; this only affects null directions and does not change any of the statements below about nonzero eigenvalues. We can rewrite the preconditioners as

$$L = \widetilde{\Delta} S_L \widetilde{\Delta}^\top, \quad S_L \equiv \frac{1}{B^2} K_\delta^{1/2} K_x K_\delta^{1/2}, \tag{173}$$

and

$$R = \widetilde{X} S_R \widetilde{X}^\top, \quad S_R \equiv \frac{1}{B^2} K_x^{1/2} K_\delta K_x^{1/2}. \tag{174}$$

Under the invertibility assumption, $\widetilde{\Delta}$ and $\widetilde{X}$ have orthonormal columns, so the nonzero eigenvalues of $L$ are exactly those of $S_L$ (and similarly for $R$ and $S_R$). Therefore the matrix functions appearing in Shampoo satisfy

$$(L + \epsilon_L I)^{-e_L} = \epsilon_L^{-e_L}\left(I - \widetilde{\Delta}\widetilde{\Delta}^\top\right) + \widetilde{\Delta}(S_L + \epsilon_L I)^{-e_L}\widetilde{\Delta}^\top \tag{175}$$

$$= \epsilon_L^{-e_L} I + \Delta A_L \Delta^\top, \tag{176}$$

where

$$A_L = K_\delta^{-1/2}\left((S_L + \epsilon_L I)^{-e_L} - \epsilon_L^{-e_L} I\right) K_\delta^{-1/2}. \tag{177}$$

Similarly,

$$(R + \epsilon_R I)^{-e_R} = \epsilon_R^{-e_R} I + X A_R X^\top, \tag{178}$$

with

$$A_R = K_x^{-1/2}\left((S_R + \epsilon_R I)^{-e_R} - \epsilon_R^{-e_R} I\right) K_x^{-1/2}. \tag{179}$$

These formulas use only the vectors in $\Delta$ and $X$ and the scalars in $K_x$ and $K_\delta$ (through constant-size operations on $B \times B$ matrices). In particular, this shows that Shampoo can be implemented using Tensor Program instructions [53], since it reduces to forming inner products, applying scalar nonlinearities to the resulting scalars, and taking linear combinations of a $\Theta(1)$ number of vectors. When $B = 1$, $K_x$ and $K_\delta$ are scalars, and the expressions reduce to the rank-1 projector form in the main text.

Plugging these expansions into the Shampoo update,

$$Q_{\text{Shampoo}}(G) = (L + \epsilon_L I)^{-e_L} G (R + \epsilon_R I)^{-e_R}, \tag{180}$$

and using $G = \frac{1}{B}\Delta X^\top$, we obtain

$$Q_{\text{Shampoo}}(G) = \frac{1}{B}\Delta M X^\top \tag{181}$$

for some $B \times B$ coefficient matrix $M$ constructed from $K_x, K_\delta, \epsilon_L, \epsilon_R, e_L, e_R$ via constant-size matrix operations. Thus the update is always a linear combination of the $B^2$ outer products $\{\delta^{(b)} x^{(b')\top}\}_{b,b'=1}^B$ with coefficients that are scalar functions of Gram matrices, which is exactly the Tensor Program form.

Finally, we check that the width scalings of $\epsilon$ and $\eta$ are unchanged when $B = \Theta(1)$. Under $\mu$P initialization, $\|x^{(b)}\|^2 = \Theta(d_{\text{in}})$ and $\|\delta^{(b)}\|^2 = \Theta\left(\frac{1}{d_{\text{out}}}\right)$, so each rank–1 term $\delta^{(b)} x^{(b)\top}$ has a nonzero singular value of size $\Theta\left(\sqrt{d_{\text{in}}/d_{\text{out}}}\right)$. Since $B = \Theta(1)$, $G$ has $\Theta(1)$ nonzero singular values of this size, so the nonzero eigenvalues of $S_L$ and $S_R$ satisfy

$$\lambda(S_L) = \Theta\left(\frac{d_{\text{in}}}{d_{\text{out}}}\right), \quad \lambda(S_R) = \Theta\left(\frac{d_{\text{in}}}{d_{\text{out}}}\right). \tag{182}$$

Therefore, choosing

$$\epsilon_L = \Theta\left(\frac{d_{\text{in}}}{d_{\text{out}}}\right), \quad \epsilon_R = \Theta\left(\frac{d_{\text{in}}}{d_{\text{out}}}\right) \tag{183}$$

keeps $(S_L + \epsilon_L I)^{-e_L}$ and $(S_R + \epsilon_R I)^{-e_R}$ nontrivial without trivializing to scalar multiplication, exactly as in the $B = 1$ case.

With these choices, applying Shampoo to a test vector $x'$ proceeds as in Section 3.1: each inner product $x^{(b)\top} x'$ is $\Theta(d_{\text{in}})$ by alignment, the right preconditioner contributes a factor $\Theta\left((d_{\text{in}}/d_{\text{out}})^{-e_R}\right)$ on the $\Theta(1)$-dimensional span of $x^{(b)}$, the gradient contributes the usual $\Theta(d_{\text{in}}/d_{\text{out}})$ factor through $\delta^{(b)} x^{(b)\top} x'$, and the left preconditioner contributes $\Theta\left((d_{\text{in}}/d_{\text{out}})^{-e_L}\right)$ on the $\Theta(1)$-dimensional span of $\delta^{(b)}$. Since $B = \Theta(1)$, summing over $b = 1, \ldots, B$ does not change width exponents, and we obtain

$$Q_{\text{Shampoo}}(G)x' = \Theta\left(\left(\frac{d_{\text{in}}}{d_{\text{out}}}\right)^{1 - e_L - e_R}\right). \tag{184}$$

Hence the $\mu$P condition $\eta Q_{\text{Shampoo}}(G)x' = \Theta(1)$ yields the same learning-rate scaling as before,

$$\eta = \Theta\left(\left(\frac{d_{\text{out}}}{d_{\text{in}}}\right)^{1 - e_L - e_R}\right). \tag{185}$$

Moreover, because $K_x/d_{\text{in}}$ and $d_{\text{out}} K_\delta$ have deterministic limits for fixed $B$, all entries of the constant-size coefficient matrices $A_L, A_R, M$ converge to deterministic limits as width grows. By the Master Theorem [53], this implies that Shampoo training dynamics under $\mu$P scaling admit well-defined deterministic infinite-width limits, explaining why the loss curves across widths can be highly consistent.

## C   Derivation for Depth-Scaling Rules

As discussed in the main text, for residual networks we scale the output of each block by $1/L$ while keeping $\Theta(1)$ feature learning inside each block, following Bordelon et al. [10], Dey et al. [14]. This implies that gradients in each residual block scale as

$$\delta = \Theta\left(\frac{1}{L d_{\text{out}}}\right), \tag{186}$$

instead of $\Theta\left(\frac{1}{d_{\text{out}}}\right)$. Thus

$$\delta^\top \delta = \Theta\left(\frac{1}{L^2 d_{\text{out}}}\right). \tag{187}$$

For Shampoo, the relevant eigenvalue becomes

$$\lambda = (x^\top x)(\delta^\top \delta) = \Theta\left(\frac{d_{\text{in}}}{L^2 d_{\text{out}}}\right), \tag{188}$$

and the update is still

$$Q_{\text{Shampoo}}(G) = (\lambda + \epsilon_L)^{-e_L}(\lambda + \epsilon_R)^{-e_R} \delta x^\top. \tag{189}$$

Parameterizing damping in units of the new $\lambda$ as

$$\epsilon_L = \epsilon'_A \lambda, \quad \epsilon_R = \epsilon'_B \lambda, \quad \epsilon'_A, \epsilon'_B = \Theta(1), \tag{190}$$

we have

$$Q_{\text{Shampoo}}(G)x' = \Theta\left(\lambda^{-(e_L + e_R)}\right)\delta(x^\top x') \tag{191}$$

$$= \Theta\left(\lambda^{-(e_L + e_R)}\right)\Theta\left(\frac{1}{L d_{\text{out}}}\right)\Theta(d_{\text{in}}). \tag{192}$$

Relative to the width-only case, we have replaced

$$\lambda \to \frac{\lambda}{L^2}, \quad \delta \to \frac{\delta}{L}, \tag{193}$$

so

$$Q_{\text{Shampoo}}(G)x' = \Theta\left(L^{2(e_L + e_R) - 1}\right)\Theta\left(\left(\frac{d_{\text{in}}}{d_{\text{out}}}\right)^{1 - e_L - e_R}\right). \tag{194}$$

To maintain $\eta Q_{\text{Shampoo}}(G)x' = \Theta(1)$ we therefore need

$$\eta = \Theta\left(L^{1-2(e_L+e_R)}\left(\frac{d_{\text{out}}}{d_{\text{in}}}\right)^{1-e_L-e_R}\right), \tag{195}$$

and the damping scales as

$$\epsilon_{L,R} = \Theta(\lambda) = \Theta\left(\frac{d_{\text{in}}}{L^2 d_{\text{out}}}\right). \tag{196}$$

For the remaining optimizers in Table 2, depth scaling can be read off from how the update behaves under an overall rescaling $G \mapsto cG$ with $c = 1/L$. In particular, for a rank–1 gradient $G = \delta x^\top$, we have

$$\|G\|_F = \|\delta\|_2 \|x\|_2 = \Theta\left(\frac{1}{L}\sqrt{\frac{d_{\text{in}}}{d_{\text{out}}}}\right), \tag{197}$$

and the RMS of entries of $G$ is $\Theta(1/(Ld_{\text{out}}))$.

**Muon.** Muon normalizes the (momentum) gradient by $\|G\|_F + \epsilon$ before applying Newton–Schulz. Choosing $\epsilon = \Theta(\|G\|_F)$ keeps this normalization nontrivial. With this choice, under $\delta \mapsto \delta/L$ both the numerator and denominator scale by $1/L$, so the normalized matrix and hence the Newton–Schulz output are unchanged. Therefore $\eta$ has no additional $L$ dependence, while

$$\epsilon = \Theta\left(\frac{1}{L}\sqrt{\frac{d_{\text{in}}}{d_{\text{out}}}}\right). \tag{198}$$

In other words, the Muon learning-rate scaling stays $\eta = \Theta\left(\sqrt{\frac{d_{\text{out}}}{d_{\text{in}}}}\right)$ and only $\epsilon$ depends on $L$.

**SOAP.** SOAP applies an Adam-like elementwise map $t \mapsto t/(|t| + \epsilon)$ in a rotated basis. Under $G \mapsto G/L$, all transformed entries scale as $t \mapsto t/L$, so this map is invariant provided $\epsilon$ scales the same way. Thus the SOAP learning-rate scaling is unchanged (no $L$ dependence), while its $\epsilon$ picks up an extra factor $1/L$ relative to the width-only scaling. In particular, in the blocked/one-sided setting with tracking indicators $e_L, e_R \in \{0, 1\}$, this gives

$$\epsilon = \Theta\left(\frac{b_{\text{out}}^{e_L/2} b_{\text{in}}^{e_R/2}}{L d_{\text{out}}}\right). \tag{199}$$

This pairs with the width scaling $\eta = \Theta\left(\frac{b_{\text{out}}^{e_L/2} b_{\text{in}}^{e_R/2}}{d_{\text{in}}}\right)$.

**AdaMuon.** AdaMuon applies Adam to Muon's orthogonalized gradient, which already removes the $L$-dependence in the gradient. Therefore neither the learning rate nor Adam's denominator $\epsilon$ acquires any additional $L$ dependence. In particular, $\eta = \Theta(1/d_{\text{in}})$ and $\epsilon = \Theta\left(\sqrt{\frac{1}{d_{\text{in}} d_{\text{out}}}}\right)$ (for the Adam denominator) remain unchanged.

## D  Scaling Rule Validation Experiment Details

### D.1  Width Scaling

In Section 3.2, we train GPT-2 architecture transformers on the OpenWebText dataset for 100M tokens using a linear decay learning rate schedule with no weight decay. We fix the depth (number of transformer blocks) to be $3$ and attention head dimension to $64$. We use a batch size of $512$ sequences of length $128$. We use a small vocabulary of size $96$ so it is practical to apply the full preconditioners on the embedding and readout layers in order to verify whether learning rate is scaled correctly for those layers where only one of the dimensions grow. We compare $\mu$P, where the per-layer learning rate is parameterized by $\eta_\ell(D) = \eta_{\text{base}}(D/D_{\text{base}})^{-\alpha_\ell}$ with $\eta_{\text{base}}$ the base learning rate, $D_{\text{base}}$ the base width, and $\alpha_\ell$ derived from Table 1 for each layer $\ell$ and optimizer, against the Standard Parameterization (SP) where $\eta_\ell(D) = \eta_{\text{base}}$. For example, when using Muon, $\alpha_\ell = 0$ for hidden layers, $1/2$ for embedding, and $-1/2$ for readout. We set $D_{\text{base}} = 128$, the width at which SP matches $\mu$P.

We zero-initialize the readout layer for both $\mu$P and SP, making learning rate scaling their only difference. We also zero-initialize MLP and attention output projections, as done in modded-nanogpt.

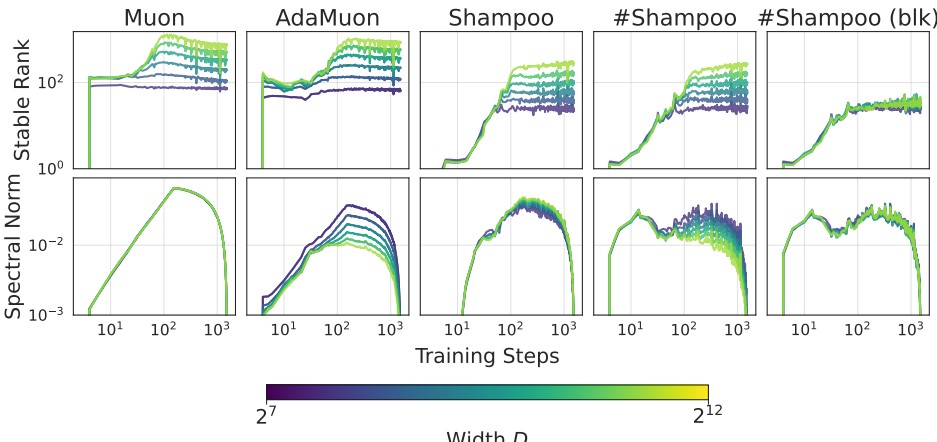

Figure 7: **Stable rank and spectral norm of the weight update at each step throughout training.**
In early training, the stable rank is well modeled as a $\Theta(1)$ quantity independent of width, but quickly
grows as training progresses unless a constant block size is used (blk). $\mu$P undershoots the spectral
norm for AdaMuon and Shampoo with grafting by a factor inversely related to their stable rank.

The embedding layer is initialized with a standard deviation 0.1. All other layers are initialized with
$1/D$ variance. We use $\beta_1 = 0.9$ (first moment) and $\beta_2 = 0.95$ (second moment and preconditioners)
for all optimizers. We set $\epsilon = 10^{-8}$ at the base model, except for Shampoo variants, which use
relative $\epsilon = 10^{-5}$. No weight decay is used for these experiments.

We use the GPT-2 architecture without bias and no learnable layernorm parameters. To increase the
maximum embedding dimension we can experiment with, we set the MLP expansion ratio to 1 rather
than 4, following [37].

### D.2 Depth Scaling

In Section 3.3, we use the same setup as in Appendix D.1, but keep width at 128 and only scale depth.
We use a base depth of 3, where $\alpha = 0$ and $\alpha = 1$ matches exactly. Due to zero-initialization of the
output projections, $\alpha = 0$ and $\alpha = 1$ have identical initializations.

### D.3 Compute-Optimal Scaling

We use the same setup as in Appendix D.1, but use the FineWeb dataset tokenized with the full GPT-2
tokenizer. We set $\beta_2 = 0.98$ for all experiments. Shampoo uses $e_L = e_R = 1/2$, i.e., Shampoo$^2$.

## E  Finite-Width Deviations

### E.1  Stable Rank is Not $\Theta(1)$ in Width in Late Training

Figure 7 shows the stable rank and spectral norm of the per-step parameter update throughout training
for a range of widths under a fixed base learning rate in $\mu$P. We show them for the up-projection
in the first MLP layer, but other hidden layers behave similarly. Unlike Adam, whose elementwise
preconditioner is known to be incapable of increasing the stable rank of the update beyond $\Theta(1)$
[56], matrix-preconditioned updates have low, $\Theta(1)$ stable rank only at the beginning of training, and
achieve high stable ranks that scale with width as training progresses. When using a fixed block size,
the preconditioner is less expressive and the stable rank remains nearly width-independent throughout
training.

As a result, for optimizers that always normalize the update RMS to $\Theta(1)$ after matrix-preconditioning
(without blocking), the spectral norm of the update decreases over time. Both Shampoo with
Adam-grafting and AdaMuon apply such normalization, through grafting and elementwise scaling
respectively. For these optimizers, no time-independent learning rate can ensure $\Theta(1)$ feature learning

(translating to $\Theta(1)$ spectral norm for hidden layers) throughout training, since early and late training require the learning rate to scale differently with width. As $\mu$P sets the learning rate based on the infinite-width limit, which effectively focuses on early training, the learning rate is too small for the bulk of training, explaining the increasing trend of optimal learning rates in Figure 2. Naturally, one might think it therefore makes sense to scale the learning rate instead based on the equilibrated stable rank, which appears to scale as $\Theta(\mathrm{width}^\gamma)$ for some $\gamma$ close to 1, but this would necessarily lead to instability in early training. By contrast, spectral normalization produces the right update scale throughout training by making time-dependent adjustments.

Finally, note that it is the combination of high stable rank with RMS or Frobenius-based normalization that breaks $\mu$P's learning rate transfer. For example, Muon and Shampoo without grafting show near-perfect learning rate transfer under $\mu$P (Figure 2). Adam also shows good learning rate transfer despite applying RMS-like normalization [55], because it does not produce a high stable-rank update due to an inexpressive preconditioner [56].

### E.2  SOAP

Our $\mu$P derivation for SOAP relies crucially on the observation that the gradient has $\Theta(1)$ rank, since $tB$ is $\Theta(1)$, and as a result the rotated gradient and its first and second moments in the preconditioner's eigenbasis are only non-zero in a $\Theta(1)$-dimensional subspace (non-zero only in a finite block in the top left corner). While true in the infinite-width limit, for realistic finite widths and large batch sizes this picture completely fails to model the structure of SOAP's update, which densely populates the full matrix, as illustrated in Figure 8 for an MLP up-projection layer. By applying Adam in the eigenbasis, SOAP can amplify tiny entries in the gradient to be $\sim \pm 1$, yielding a dense update even if the moments have small stable ranks, so long as they have high matrix ranks, which is expected from large batch sizes or training steps.

Given this observation, we experimented with alternative models for the spectral norm of SOAP's update, but none led to satisfactory transfer. For example, modeling the spectral norm as $\Theta(\sqrt{d_{\mathrm{in}}} + \sqrt{d_{\mathrm{out}}})$ as in a random Gaussian matrix with unit entry variance tends to decrease the maximum stable learning rate as width increases, and modeling it as $\Theta(\sqrt{d_{\mathrm{in}} d_{\mathrm{out}}})$ as if it were simply Adam undershoots the optimal learning rate as width increases.

## F  Spectral Normalization

We estimate the spectral norm of the optimizer update using online power iteration following Large et al. [26]. Specifically, we maintain an estimate $v_t$ for the top singular vector of the optimizer update $A_t$ at each step $t$. At each step, we compute $y_t = A_t v_t$, estimate the spectral norm of $A_t$ as $\hat{\sigma}_t = \|y_t\|$ and update $v_t$ to $v_{t+1} = \frac{A_t^\top y_t}{\|A_t^\top y_t\| + \epsilon}$ for a small $\epsilon > 0$. Therefore, only two matrix-vector multiplies are needed per step per layer. If $A_t$ varies slowly over time, as expected from the use of momentum, $v_t$ and $\hat{\sigma}_t$ should well approximate the instantaneous top singular vector and singular value. At initialization, $v_0$ is a random unit vector. Near initialization, $A_t$ may be zero for some

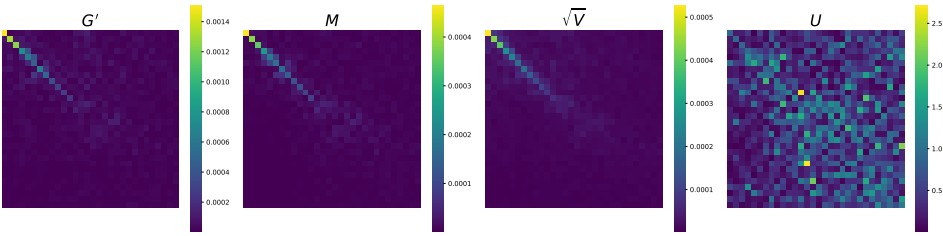

Figure 8: **Visualization of the absolute value of the SOAP update in the preconditioner's eigenbasis, cropped to the first $32 \times 32$ block.** In the eigenbasis, the gradient ($G'$), first moment ($M$), and second moment ($V$) concentrate along a few top entries along the diagonals, but the update ($U$) is dense.

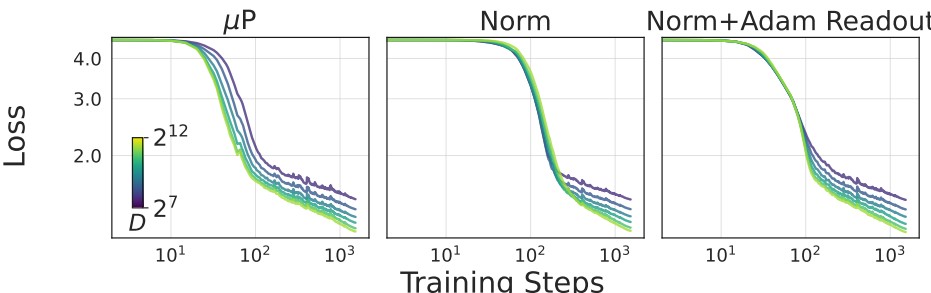

Figure 9: **SOAP (no blocking) training dynamics.** $\mu$P (left) does not give consistent training dynamics. Applying spectral normalization (middle) gives more consistent training dynamics but wider models learn slightly more slowly initially. Further replacing SOAP with Adam for the readout layer gives consistent dynamics.

layers, in which case performing $v_{t+1} = \frac{A_t^\top y_t}{\|A_t^\top y_t\| + \epsilon}$ would result in a zero vector. To prevent this, we skip this update if it would produce a vector of norm less than $0.5$.

Given the estimated spectral norm $\hat{\sigma}_t$, we rescale the update $A_t$ to $\sqrt{\frac{d_{\text{out}}}{d_{\text{in}}}} A_t / \hat{\sigma}_t$, setting its spectral norm to the desired $\sqrt{\frac{d_{\text{out}}}{d_{\text{in}}}}$ up to errors in the spectral norm estimate.

**RMS Normalization on the Embedding Layers.** For the token and positional embedding layers, we perform RMS normalization on the full embedding matrices instead, which is much cheaper with a large vocabulary. The rationale is as follows: since inputs to the embedding layer are one-hot, the embedding layer is better viewed as a fixed-size collection of vector parameters. Normalizing the spectral norm of each vector (viewed as a $d_{\text{in}} = 1$ and $d_{\text{out}} = d_{\text{embd}}$ matrix) reduces to RMS normalization. Based on this reasoning, Bernstein and Newhouse [6] proposed applying RMS normalization to each such vector. Here, we adopt the cheaper alternative of applying RMS normalization to the full embedding matrix, which should achieve similar results.

**Inconsistent SOAP Readout Dynamics Even with Spectral Normalization.** Figure 9 shows that even with spectral normalization, SOAP training dynamics are not fully consistent across widths, as wider models learn more slowly initially. We found this is due to the spectral norm overestimating the level of alignment in the readout layer for SOAP and can be fixed by replacing SOAP with Adam in the readout layer. We do so for the experiment in Figure 3 (right). However, we found this fix is not necessary for good learning rate transfer or performance given long training horizons (e.g., 20 tokens per parameter), where spectral normalization alone is sufficient. If blocking is used, then training dynamics are already consistent under $\mu$P without spectral normalization or using Adam for the readout layer (Figure 1 (left)).

## G   Muon $\mu$P Scaling

**Incorrect Alternative Scaling.** Using the experiment setup in Appendix D.1, Figure 10 (left) shows that the following learning rate scaling rules do not transfer the optimal learning rate, only the $\eta \sim \sqrt{d_{\text{out}}/d_{\text{in}}}$ scaling (first column) does:

1. Muon-Kimi: multiplying the learning rate by $\gamma = 0.2\sqrt{\max(d_{\text{in}}, d_{\text{out}})}$ by Liu et al. [30]. We show two versions of Muon-Kimi, one where no further width-dependent scaling is applied on top of the $\gamma$ factor (Muon-Kimi $\Theta(1)$), the other where one applies the correct Adam $\mu$P scaling $\Theta(d_{\text{in}}^{-1})$ on top of $\gamma$ (Muon-Kimi $\Theta(d_{\text{in}}^{-1})$). The latter reflects the purpose of matching Adam RMS, which is to make Adam learning rate directly transferable to Muon. However, both led to poor transfer. The latter approach fails because Liu et al. [30] assumes the Muon update is full stable rank, thus differing from $\mu$P.

2. Muon $\Theta(d_{\text{in}}^{-1})$: Shah et al. [40] directly uses Adam's $\mu$P scaling rule by Shah et al. [40], which undershoots the learning rate in all layers.

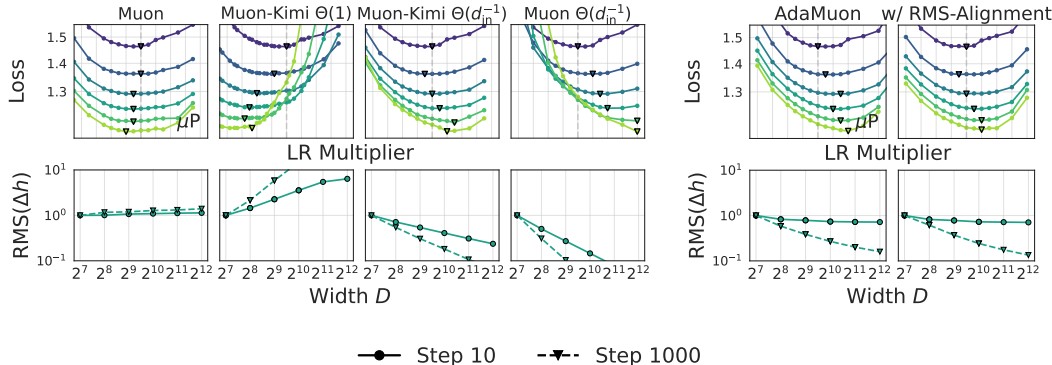

Figure 10: **Alternative learning rate scaling for Muon.** (**Left**) Learning rate scaling proposed in Liu et al. [30] (Muon-Kimi) and Shah et al. [40] (Muon $\Theta(d_{\text{in}}^{-1})$) do not transfer optimal learning rates. (**Right**) For AdaMuon, adding explicit RMS-alignment according to original implementation [44] does not impact learning rate transfer results and the optimal learning rates continue to shift right.

Note, however, Liu et al. [30] also scales batch size, training horizon, and number of layers together with width, in which case the optimal learning rate scaling may not follow $\mu$P. Furthermore, their Adam learning rate is determined from empirical power law fits, rather than scaling as $\mu$P's $\Theta(1/d_{\text{in}})$. These differences may explain why they still observe good learning rate transfer from Adam.

**RMS Alignment.** The original AdaMuon optimizer performs explicit RMS alignment, where the update is $-\eta \cdot 0.2\sqrt{d_{\text{in}}d_{\text{out}}}\frac{\hat{O}_t}{\|\hat{O}_t\|_F}$ where $\hat{O}_t$ is obtained from applying Adam on top of the orthogonalized gradient. Since Adam already normalizes the RMS of the update to $\Theta(1)$ at every step, doing so has little impact on how the learning rate should be scaled. In particular, $\mu$P scaling remains $\eta \sim 1/d_{\text{in}}$. Therefore, we did not use RMS alignment in our experiments in Figure 2. Figure 10 (right) shows adding this explicit RMS alignment does not address the shifting optima observed in Figure 2.

# H    Scaling Law Experiments

For the scaling experiments in Section 4, we use the FineWeb dataset with sequence length 1024 and batch size 128. We use the Llama-2 architecture without bias or learnable RMSNorm parameters, following modded-nanogpt. The models are trained with 20 tokens per parameter, a linear decay learning rate schedule, and gradient clipping of 1.

| Params | Seq Len | Embedding Dim | FFN Dim | # Layers | # Heads |
|--------|---------|---------------|---------|----------|---------|
| 190M | 1024 | 512 | 2048 | 32 | 8 |
| 380M | 1024 | 768 | 3072 | 32 | 12 |
| 640M | 1024 | 1024 | 4096 | 32 | 16 |
| 1.4B | 1024 | 1536 | 6144 | 32 | 24 |

Table 3: Architecture specifications for each model size.

## H.1    Base model tuning

**Optimizer choices.** We pick the optimizers that perform the best in the OpenWebText experiments:

- **Muon:** we find that using Adam for embedding and readout layers performs better than Muon, as observed by previous works [30, 23].
- **Shampoo:** we find that Shampoo$^2$ ($e_L = e_R = 1/2$) with Adam grafting and Adam applied to the embedding and readout layers performs well. We found one-sided Shampoo on

- embedding and readout layers [3] to perform significantly worse. We use blocking with block size 512.
  - **SOAP:** we use spectrally normalized SOAP since the optimal learning rates are more stable than regular SOAP. We use one-sided SOAP for embedding and readout layers. We use blocking with block size 512.

**Hyperparameter tuning.** We tune the main hyperparameters of each optimizer to ensure a fair comparison. We divide hyperparameters into subsets and tune each set of hyperparameters sequentially. At each stage of tuning, previously selected hyperparameters remain fixed, except learning rates, which we retune in every stage. We show the optimal hyperparameter values found at the end of each stage in Table 4, with links to the full tuning sweeps.

| Optimizer | LR | LR mult. | $\beta_1$ | $\beta_2$ | Warmup | WD | wandb | Best Loss |
|---|---|---|---|---|---|---|---|---|
| Adam | **4e-3** | - | 0.9 | **0.98** | **0.2** | 2e-4 | link | 3.23635 |
| | **4e-3** | - | 0.9 | 0.98 | 0.2 | **2e-4** | link | 3.23635 |
| Muon | **8e-3** | **1.6** | 0.95 | - | **3e-3** | 1.6e-4 | link | 3.18302 |
| | **8e-3** | 1.6 | 0.95 | - | 3e-3 | **3.2e-4** | link | 3.17795 |
| Shampoo | **2e-3** | **32** | 0.9 | 0.98 | **5e-2** | 2e-4 | link | 3.18254 |
| | **2e-3** | 32 | **0.95** | **0.98** | 5e-2 | 2e-4 | link | 3.17796 |
| | **2e-3** | 32 | 0.95 | 0.98 | 5e-2 | **2e-4** | link | 3.17796 |
| SOAP | **3.2e-2** | - | 0.9 | **0.999** | **6.25e-3** | 2e-4 | link | 3.1751 |
| | **3.2e-2** | - | 0.9 | 0.999 | 6.25e-3 | **2e-4** | link | 3.1751 |

Table 4: **Tuning sweeps for each optimizer considered in scaling experiments.** LR stands for learning rate. LR mult is the multiplier applied for embedding and readout layers when Adam is used, applicable only to hybrid optimizers. $\beta_1$ and $\beta_2$ are used differently for different optimizers, as specified in Appendix A. Warmup is the fraction of tokens used for warmup. WD is the *independent* weight decay. We share the Weight&Biases view for the sweep in wandb. Bolded values are optimal values found in the sweeps. We ensure the optima are not at the boundary of the sweep grids. We sweep LR, LR mult, WD over powers of two centered around reasonable base values, except $\beta_{1,2}$, which are swept over $0.9, 0.95, 0.98, 0.999$. Full wandb sweeps available through the links.

## H.2 Compute Multiplier Estimation

In Figure 6, we estimate the compute multiplier by dividing the estimated Adam compute to achieve the target loss over the actual compute spent by the optimizer. That is,

$$\text{Compute Multiplier}(C_{\text{opt}}) = \frac{C_{\text{Adam}}}{C_{\text{opt}}} \tag{200}$$

where $C_{\text{Adam}}$ is the estimated compute required by Adam to achieve the loss achieved by optimizer opt with $C_{\text{opt}}$ FLOPs. We estimate Adam's compute using linear interpolation between loss-compute pairs in log-log space. If the loss is smaller than Adam's smallest loss, we use the closest two points to extrapolate (`scipy.interpolate.interp1d` with `fillvalue="extrapolate"`).

The compute is estimated by counting the FLOPs for all operations in a transformer. We include the attention dot product FLOPs that scale quadratically with context length, which is slightly more accurate than the $6ND$ formula from [24]. We follow this estimation approach from Levanter.

## H.3 Near-Optimal Hyperparameter Transfer

We scale the learning rate using $\mu$P and the independent weight decay as $1/\text{width}$ as we scale up the model to transfer hyperparameters found on the base model. Here we verify that the scaled learning rates and weight decays are nearly optimal. In Figure 11, we demonstrate how the loss does not improve from our scaled hyperparameters if we change the learning rate, weight decay, or learning rate schedule. We estimate there is a $0.1\%$ noise floor in the final loss across runs with the exact same hyperparameters due to non-determinism, so differences below this floor should be ignored. Note,

though, there is a slowly decreasing trend of optimal learning rates, potentially due to the increasing training horizon [17]. This suggests at even larger scales our learning rate scaling may break down.

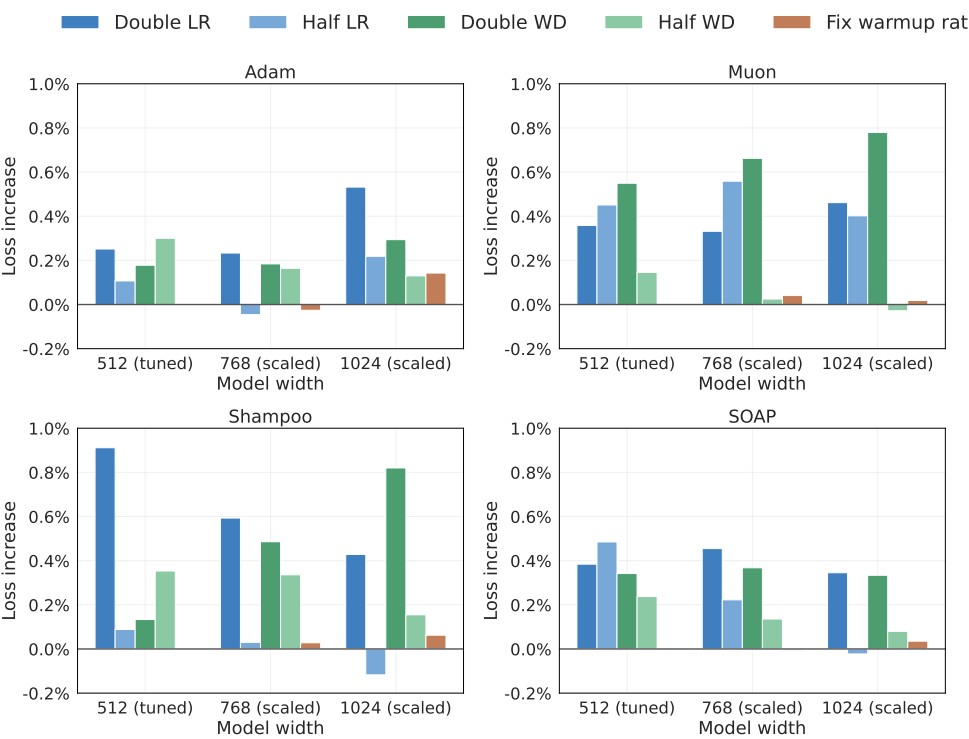

Figure 11: **Ablation of learning rate, weight decay and warmup schedule.** We have five perturbations to test the optimality of our scaled hyperparameters, namely halving or doubling the learning rate or weight decay and using a fixed warmup ratio instead of a fixed number of warmup tokens. Using fixed warmup ratio or tokens doesn't have a significant impact on the final loss. Perturbations on learning rate and weight decay lead to higher losses or differences below the $0.1\%$ noise floor.

## H.4 Alternative scaling approach

We expand the ablation experiments in Figure 6 to other optimizers to understand the effects of different components of the scaling rule. In Figure 12, we demonstrate that only scaling the learning rate with $\mu$P or only scaling the weight decay rapidly diminishes the efficiency gains, showing both are important.

## H.5 Optimal Token-Per-Parameters (TPP)

Matrix-based preconditioned optimizers are more data efficient, so we expect that in the compute-optimal regime, their optimal TPPs will be smaller than Adam. To verify, here we estimate the optimal TPP for Muon and Adam. We follow Approach 2 from Hoffmann et al. [20] to fix compute budgets and tune model sizes to find the optimal TPPs.

The tuning runs for Adam and Muon can be found in wandb sweeps (Adam and Muon). As we vary TPP while fixing the FLOPs budget, we found that using $\mu$P and fixing the independent weight decay yields optimal hyperparameters, consistent with Bergsma et al. [4] (one needs to convert their coupled weight decay to the equivalent independent weight decay to see this agreement). The results are shown in Figure 13 and Table 5. We observe that Adam has a 1.3 times larger optimal TPP than Muon, fairly consistent with the 1.4 times more compute efficiency Muon gets over Adam in Figure 6. The agreement between these two values is expected from a scaling law of the form $\mathcal{L} = E + (N/N_0)^{-\alpha} + (T/T_0)^{-\beta}$ where $N$ is the number of parameters and $T$ is the number of tokens, if we assume that only $T_0$ (data efficiency) can be altered by the optimizer, not the loss for a fixed model size given infinite data.

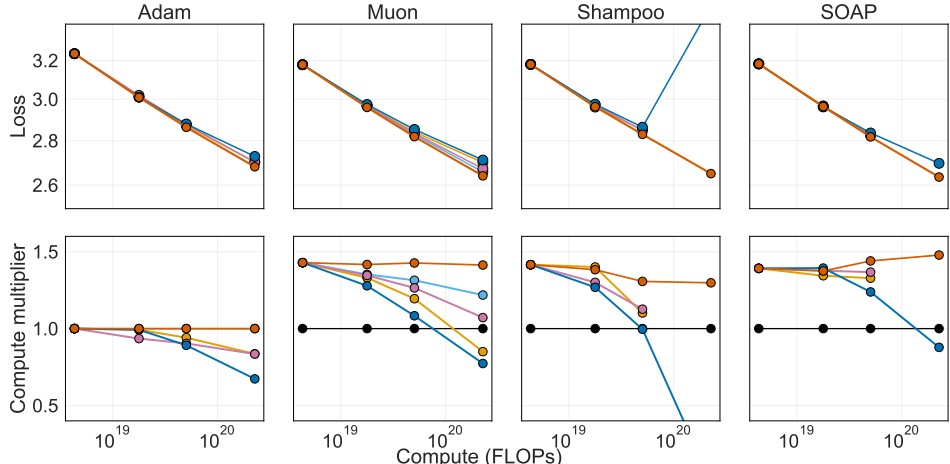

Figure 12: **Alternative scalings for various optimizers underperform.** Across various optimizers, $\mu$P with $1/D$-scaled weight decay yields the best performance as model size increases. Only scaling the learning rate or the weight decay, while better than scaling neither (SP), leads to worse performance, demonstrating both components are crucial.

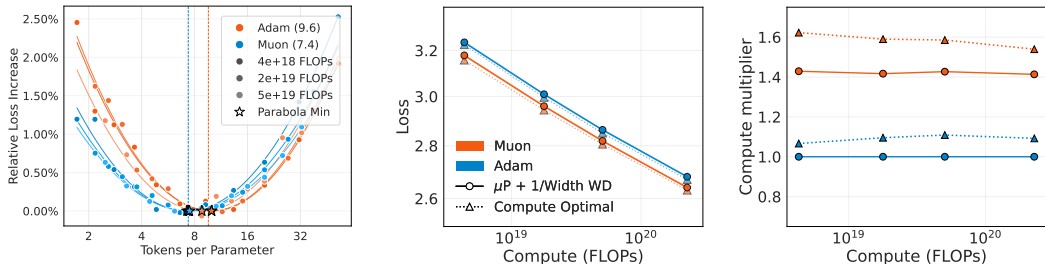

Figure 13: **Optimal TPP for Adam and Muon on FineWeb.** Optimal TPPs for Adam and Muon are estimated to be 9.6 and 7.4. Adam's optimal TPP is 1.3 times larger than Muon's. With the tuned TPP, both Muon and Adam achieve more speedup over the Adam baseline ($\mu$P, $1/D$ weight decay, 20 TPP) shown in solid blue.

Our result shows that the optimal TPP for Adam is 9.6, much smaller than the original estimate of 20 by Hoffmann et al. [20]. We believe the explanation is two-fold. First, the FineWeb dataset is known to be of higher quality than the C4 dataset used by Hoffmann et al. [20], which suggests it has higher information density (bits per token) and thus for a fixed number of parameters the number of tokens the model can fit is reduced. Second, by extensively tuning the (base) hyperparameters $(\eta, \beta_1, \beta_2, \lambda)$, the same model converges with fewer tokens, which also reduces the optimal TPP. For example, if training all models with a very small learning rate, the optimal TPP will be very high, scaling as $\Theta(1/\eta)$ as training approaches the gradient flow.

| Optimizer | Width | Loss | TPP |
|---|---|---|---|
| Adam | 576 | 3.226039 | 13.338798 |
| Adam | 896 | 2.996272 | 11.494327 |
| Adam | 1280 | 2.849975 | 8.783869 |
| Muon | 704 | 3.157175 | 6.607409 |
| Muon | 1024 | 2.943154 | 7.071705 |
| Muon | 1344 | 2.805800 | 7.325234 |

Table 5: **Optimal widths for the compute budgets of the** 190**M,** 380**M, and** 640**M models**.

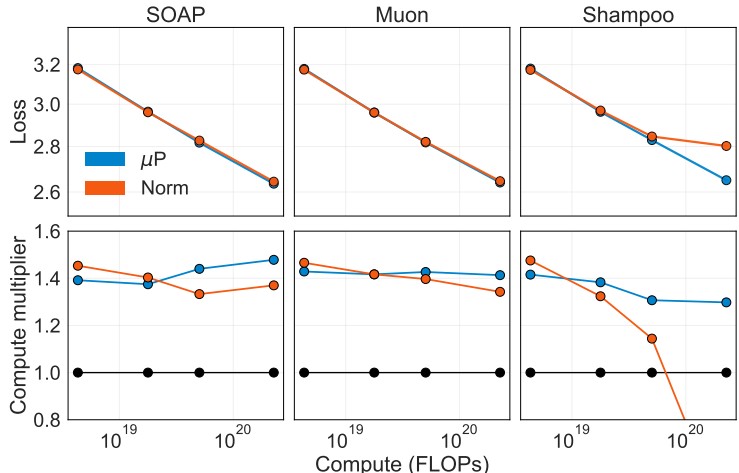

Figure 14: **Comparing spectrally normalized optimizers with $\mu$P.** The setting is identical to Section 4. We compare $\mu$P scaling and spectral normalization while keeping the same $1/$width weight decay scaling. In this setting, spectral normalization does not improve over $\mu$P.

### H.6 Spectral Normalization.

We study the scaling performance of the spectrally normalized variants of SOAP, Muon (where spectral normalization only affects the layers using Adam given Muon updates already have unit spectral norm), and Shampoo. In Figure 14, we find the spectrally normalized variants show no advantage over $\mu$P. This is not surprising given that we used a fixed block size (512) for all three optimizers, enabling good hyperparameter transfer already under $\mu$P (Section 3.2). For Shampoo, we find spectral normalization in fact performed worse than $\mu$P. Figure 15 shows that this is due to the optimal learning rate and weight decay shifting toward smaller values for larger models for spectrally normalized Shampoo. We hypothesize this is due to the base model being tuned too close to instability, which, combined with the finding that the optimal learning rate decreases with scale on top of $\mu$P in the compute-optimal regime, led to observed suboptimal performance for the larger models.

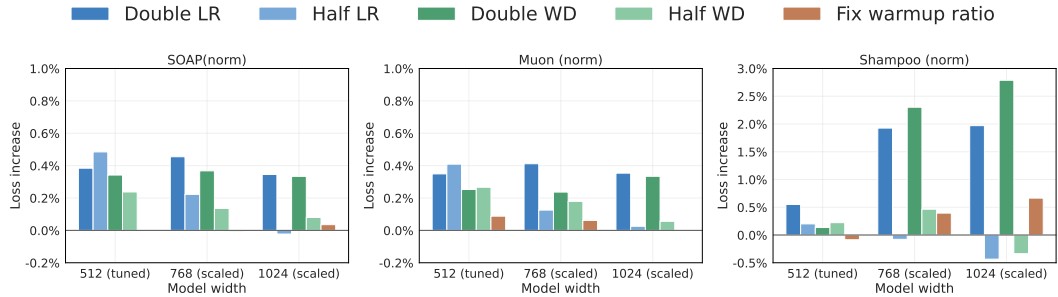

Figure 15: **Ablations for learning rates and weight decay for spectrally normalized optimizers.** SOAP and Muon have slightly shifting optimal learning rates and weight decays, partially due to the increasing horizon. Shampoo, however, shows a clear shifting toward smaller learning rates and weight decays.

## I   Hardware

We trained all of our models on TPU-v4 and TPU-v6e, supported by the Google TPU Research Cloud program. OpenWebText experiments are trained on TPU-v4-4 and FineWeb experiments on TPU-v6e-8, TPU-v6e-16, and TPU-v4-32.

# J   Broader Impact and Limitations

**Broader Impact.**   Our work improves the understanding of how to efficiently scale matrix-preconditioned optimization in deep learning, which has the potential to reduce the cost of training machine learning models and make machine learning research and workflows more accessible.

**Limitations.**   We perceive two main limitations of our work. First, due to limited computational budget, our experiments are relatively small in scale compared to typical training runs in industry. Verifying how well our results generalize to larger models trained with more compute is an exciting future direction. Second, while we have shown that optimizers like Shampoo and Muon can significantly improve the compute efficiency of training, we do not investigate *why* they improve the efficiency. Understanding this question holds potential for designing even more efficient future optimizers.

