# OpenReview forum: "Hyperparameter Transfer Enables Consistent Gains of Matrix-Preconditioned Optimizers Across Scales"
_NeurIPS.cc/2025/Conference — NeurIPS 2025 poster_

### Official Review · Reviewer_6YHN · 2025-06-30

**Clarity:** 2
**Significance:** 2
**Originality:** 2
**Rating:** 3
**Confidence:** 5

**Summary:**

This paper studies the scaling laws of loss, data, compute, when optimising neural networks with second-order optimisers, and it derives novel rules for scaling hyperparameters.
Previous work had already addressed these two topics, the novelty of this work is mainly the following: for a given compute budget, it shows that less data should be used to approach the Pareto frontier of loss vs compute when using second-order optimisers. Scaling laws for hyperparameters are derived theoretically and tested empirically for more optimisers and more hyperparameters than previously done.

**Questions:**

NA

**Ethical Concerns:**

["NO or VERY MINOR ethics concerns only"]

**Final Justification:**

The authors addressed most of my concerns, I increased the score, however I still do not give an accept rating because the quality of the first submission was very low, so I would have to trust that they improve the final paper by a substantial amount.

**Limitations:**

yes

**Quality:**

2

**Strengths And Weaknesses:**

Strengths:

- It considers a good number of optimisers: Shampoo, Shampoo2, Muon, Soap.
- It points out a possible mistake done in a previous study that did not achieve a Pareto frontier.

Weaknesses:
A few sections of the paper are not written clearly at all, and a section is missing. For example:
- Section 3.1 does not explain how important is the assumption of random weights in deriving the results.
- Section 3.4 refers to and describes results of a figure and a table that are not present in the main text. It seems that those were literally moved in the Appendix before submission because they did not fit in the page limit.
- All links between main text and Appendix are broken (there was no need to split them by the way).
- Appendix E is empty, it says “Blank”, as if the Appendix was not even checked before submission.
- There is no explanation about how the “target loss” is picked in Figure 3.
- Exponents and constants of the scaling laws in Figure 3 are not reported. So there is no answer to the important question given in the main text: “For any second-order optimizer that outperforms Adam… must improve, by either decreasing the constant t0 or increasing the exponent.”
- KFAC is studied in the Appendix but never mentioned anywhere else.

Other major issues:
- The effect of width and depth scaling on second order optimisers on Transformer is very little, for some optimisers there is no effect at all.
- I disagree with the statement “the transformer architecture is overall highly robust suboptimal learning rates.” There may be other reasons, for example that theory is derived for MLP, not Transformer, and also that second-order optimisers are often more robust to hyperparameters than first order ones.
- Theoretical results for batch size scaling seem to be not very useful, in fact an empirical scaling law was eventually used in experiments.
- Very small models are used for studying scaling rules of loss, data, compute, with character-level tokenisation that is very non-standard in language modelling.
- Scaling rules usually follow asymptotic arguments, meaning that the variable that is supposed to scale tends to infinity. However, that does not seem to be the case for the aspect ratio A in Table 1.

---

> ### Author Rebuttal · Authors · 2025-07-31
>
> Thank you for your thoughtful feedback. Inspired by your comments, we provide additional results below, as well as several clarifications.
>
> **On the novelty of this work**
>
> We would like to clarify that the novelty of this work goes beyond identifying that second-order optimizers have different Chinchilla scaling laws. As you pointed out, we derived hyperparameter scaling rules for a number of second-order optimizers that prior works did not consider. The only existing work studying µP scaling for second-order optimizers is [1], which only addresses width scaling and only covers K-FAC and Shampoo. By contrast, we presented a different and more general approach, avoiding the push-through identity used in [1], which does not apply when $\xi’\neq\xi$ in Equation 8 of our paper. Our approach allows us to derive the scaling rules for a much wider range of optimizers, including Muon, Shampoo with blocking, Shampoo with grafting, and SOAP, and easily handles both width and depth scaling. Our results have high practical relevance as 1) these optimizers have been shown to significantly outperform K-FAC and vanilla Shampoo considered in [1], and 2) joint width and depth scaling is more efficient than width-only scaling at scale as demonstrated in [2,3].
>
> **Testing our scaling rules in larger-scale, full vocabulary experiments**
>
> While Figure 1 (left) shows that the scaling rules had a relatively small impact on transformers, the lack of consistent feature learning under SP in Figure (right) and the growing gap between SP and our scaling rules in Figure 2 (left) strongly suggest that the benefit of correct scaling rules will be more pronounced at larger scales. To demonstrate that, we ran experiments with even larger models using the full vocabulary version of higher quality Fineweb dataset, showing that the distinction between SP and our scaling rules indeed become more pronounced at larger scales.
>
> *Compute-optimal scaling (c.f. Figure 3)*
>
> Furthermore, we conduct full vocabulary experiments on the Fineweb dataset in the compute-optimal setting. Due to time constraints, we only compare Adam with Muon, but will include additional optimizers in the final paper. In the following tables, we find that our observations that second-order optimizers 1) reduce Chinchilla-optimal tokens per parameter by about 2x, and 2) lead to about 1.5x - 2x compute efficiency gain relative to Adam continue to hold in this more standard experiment setup. Here, we also extended the model size up to 200M parameters. The optimal token per parameter is estimated using the same fit procedure in Section 4.1.
>
> | Optimizer | Estimated optimal token per parameter |
> | --- | --- |
> | Adam | 11.2 |
> | Muon | 5.4 |
>
> | Adam loss | 5.477 | 5.121 | 4.935 | 4.744 | 4.544 | 4.342 | 4.160 | 4.020 | 3.908 | 3.804 |
> |------|------|------|------|------|------|------|------|------|------|------|
> | Adam compute | 1.128e+16 | 2.635e+16 | 4.931e+16 | 8.212e+16 | 1.275e+17 | 2.712e+17 | 5.226e+17 | 9.465e+17 | 1.639e+18 | 2.739e+18 |
>
> | Muon loss | 5.889 | 5.382 | 5.041 | 4.785 | 4.605 | 4.329 | 4.136 | 3.980 | 3.852 | 3.747 |
> |------|------|------|------|------|------|------|------|------|------|------|
> | Muon compute | 5.534e+15 | 1.293e+16 | 2.420e+16 | 4.030e+16 | 6.257e+16 | 1.331e+17 | 2.565e+17 | 4.646e+17 | 8.048e+17 | 1.345e+18 |
>
> | Target loss | Estimated Muon compute efficiency gain |
> |-------------|--------------------------------|
> | 4.660 | 1.81× |
> | 4.261 | 2.17× |
> | 3.966 | 2.55× |
>
> Please also refer to our response to Reviewer xRrA subtitled "Larger-scale, full vocabulary experiments" to see our experiments regarding stable optimal learning rate.
>
> **Transformer’s robustness to learning rate**
>
> Our statement that the transformer architecture is overall highly robust to suboptimal learning rates is directly based on empirical evidence in Figure 1 (left), where the impact of the correct scaling rules is relatively small, as you also pointed out. Furthermore, our experiments in Figure 6 Appendix G show that MLPs are much less robust to suboptimal learning rates, unlike transformers, even with second-order optimizers, directly showing that the transformer architecture indeed leads to improved robustness.
>
> Despite being derived from MLPs, our scaling rules apply to transformers as the underlying principles of µP do not depend on specific architecture details and have been shown to apply to transformers both in theory and in practice [3]. We note that it is standard practice in the µP literature to perform theoretical derivations with MLPs for simplicity [3,4,5]. We will clarify this in the next version. Finally, the consistent feature learning strength across scales in Figure 1 (right) provides a direct empirical verification that our scaling rules are correct.
>
> **Significance of Random Initialization**
>
> We follow prior works on µP in assuming the weights are randomly initialized. This choice both reflects practice and allows one to prove existence of well-defined infinite limits using the central limit theorem and the law of large numbers [4,5,6] which underlie our derivation for the scale of the gradients, activations, and their updates in Section 3.1. We note that recent work shows that the random initialization assumption is not necessary, and can be replaced by conditions on the spectral norms of the weights at initialization [6].
>
> **Why asymptotic arguments apply**
>
> While the aspect ratio $A$ does not grow to infinity, asymptotic arguments in µP only require the width (or depth in the case of depth-scaling) to grow to infinity in order to appeal to the central limit theorem and the law of large numbers [4,5].
>
> **Choice of target Loss in Figure 3**
> To highlight the efficiency multipliers across different target losses, we extend results in Figure 3 into a more detailed table. Naive refers to using the same data scaling as is optimal for Adam, while ours indicate using corrected scaling we suggested that reduce the number of tokens per parameter for Muon and Shampoo.
>
> Wall-clock speed ups relative to Adam on Openwebtext:
>
> | Target Loss | Adam| Muon (naive) | Shampoo(naive) | Muon (ours) | Shampoo (ours) |
> | --- | --- | --- | --- | --- | --- |
> | 1.38 | 1.00 | 3.06 | 2.69 | 3.61 | 3.22 |
> | 1.20 | 1.00 | 2.69 | 2.35 | 3.19 | 2.78 |
> | 1.03 | 1.00 | 2.29 | 1.99 | 2.73 | 2.31 |
> | 0.94 | 1.00 | 2.05 | 1.78 | 2.46 | 2.05 |
>
> Wall-clock speed ups relative to Adam on Chess:
>
> | Target Loss | Adam| Muon (naive) | Shampoo (naive) | Muon (scaling) |Shampoo (scaling) |
> | --- | --- | --- | --- | --- | --- |
> | 0.83 | 1.00 | 1.42 | 1.71 | 1.57 | 1.87 |
> | 0.78 | 1.00 | 1.45 | 1.67 | 1.60 | 1.81 |
> | 0.74 | 1.00 | 1.48 | 1.63 | 1.62 | 1.76 |
> | 0.70 | 1.00 | 1.52 | 1.58 | 1.66 | 1.68 |
>
> **Exponents and Constants of Figure 3**
>
> In Figure 3, we fit a power law for loss
>
> $$ L(C) = \left( \frac{C}{C_0} \right) ^{-b} + L_0,$$
>
> where $C$ is the amount of training compute. We fix $L_0$ across the three optimizers so that the values of the other parameters ($C_0$ and $b$) are meaningfully comparable. We then show the fitted coefficients:
>
> Openwebtext:
>
> | Optimizer | $L_0$ | $C_0$ | $b$ |
> | --- | --- | --- | --- |
> | Adam | 0.578 | 0.374 | 0.105 |
> | Muon | 0.578 | 0.092 | 0.100 |
> | Shampoo | 0.578 | 0.102 | 0.099 |
>
> Chess:
>
> | Optimizer | $L_0$ | $C_0$ | $b$ |
> | --- | --- | --- | --- |
> | Adam | 0.538 | 7.252e-04 | 0.149 |
> | Muon | 0.538 | 5.118e-04 | 0.151 |
> | Shampoo | 0.538 | 3.099e-04 | 0.145 |
>
> We see that only the constant factor improved significantly (by ~2x) while the exponent changed less than 10%. This suggests that second-order optimizers primarily change the constant but not the scaling exponent. We will discuss this important finding in the updated version.
>
> **Why our batch size scaling experiments are valuable**
>
> Our results on batch size scaling are valuable as they reveal that existing theoretical models for batch size scaling based on SDEs do not allow effective hyperparameter transfer in practice. In the paper, we fully acknowledge that a better theoretical understanding of batch size scaling is needed. We believe our thorough evaluation and honest presentation should be viewed as a strength rather than a weakness, as acknowledged by Reviewer K9rj.
>
> **Improving paper organization**
>
> Thank you for your suggestions in improving the organization of our paper. We separately submitted the main paper and the appendix following the NeurIPS submission guidelines, which prevented the link from working properly. Per your suggestion, we will include the figures referenced in Section 3.4 to the main text using the additional page in the camera-ready revision to improve readability. We deferred those figures to the appendix due to space constraints. We apologize for the formatting error that led to an extra section in the appendix (section E) and will correct it in the revision. No content is intended for section E.
>
> References:
> 1. Ishikawa and Karakida. "On the Parameterization of Second-Order Optimization Effective towards the Infinite Width." *International Conference on Learning Representations (ICLR)*, 2024
> 2. Kaplan, Jared, et al. "Scaling laws for neural language models." *arXiv preprint arXiv:2001.08361*, 2020.
> 3. Dey, Nolan, et al. "Don't be lazy: CompleteP enables compute-efficient deep transformers." *arXiv preprint arXiv:2505.01618*, 2025.
> 4. Yang et al. "Tensor programs v: Tuning large neural networks via zero-shot hyperparameter transfer." *Advances in Neural Information
> Processing Systems (NeurIPS)*, 2021.
> 5. Yang and Littwin. "Tensor programs ivb: Adaptive optimization in the infinite-width limit." *International Conference on Learning Representations*, 2023
> 6. Yang, Greg, James B. Simon, and Jeremy Bernstein. "A spectral condition for feature learning." *arXiv preprint arXiv:2310.17813*, 2023.
> 7. Hoffmann, Jordan, et al. "Training compute-optimal large language models." *Advances in Neural Information
> Processing Systems (NeurIPS)*, 2022.

---

> > ### Comment · Reviewer_6YHN · 2025-08-04
> >
> > **Novetly** I did not question the novelty of this work, I just said what in my opinion is the *main* novelty. My opinion is unchanged.
> >
> > **Larger scale experiments** I appreciate that will improve the quality of the paper.
> >
> > **Theory for MLP transfers to Transformers** I still do not believe that theoretical results on MLP trivially transfer to Transformers. Even if that has been shown for some optimizers in previous papers, that does not mean that it automatically holds for other optimizers.
> >
> > **Random initialization** Even if some previous papers show that the norm, rather than the distribution, is enough for deriving the results, I still find Section 3.1 written very poorly. In particular, there was no mention in that section about the spectral norm, and it is not clear at all if you are planning to improve that section and how.
> >
> > **Asymptotic arguments** If I understand correctly, somewhere in your paper there should be a statement similar to: "we take the limit of large d_{in} and d_{out} and keeping their ratio A constant". However there is no such statement and it does not look like that you intend to add it.
> >
> > **Target loss** I appreciate the new experiments, that will improve the quality of the paper. However, there is no acknowledgement for why the value of the target loss was not included in the submission. It still seems to me that the paper was submitted at the last minute without proper checks, that makes me worry about the quality of any eventual final version.
> >
> > **Exponents and constants** I appreciate that you are giving the values now, but there is no acknowledgement for why these numbers were not given in the first place. It still seems to me that the paper was submitted at the last minute without proper checks, that makes me worry about the quality of any eventual final version.
> >
> > **Batch size scaling experiment** My understanding of your comment is that your theory on batch size should be taken as a negative result, an example of theory that does not work, that is valuable on its own and a is a proof that you are being honest. I disagree, in the sense that honesty is a pre-requisite, therefore it should not affect the evaluation of a paper, and I do not believe that NeurIPS has space to publish theories that did not work.
> >
> > **Paper organization** I think it is unfair to move some figures in the Appendix at the last minute, because they did not fit the page limit, and then say: "we will put them back in the paper once is accepted and we have one more page". The one additional page given after acceptance should serve to include the outcome of discussion with reviewers, not to bring back pieces of appendix in the main text. Overall, I do not feel that the authors value reviewers comment that much, which makes me feel worried about the quality of any eventual final version.
> >
> > By the way, my comment on why the Appendix has a section on KFAC was ignored.

---

> > > ### Author Response · Authors · 2025-08-06
> > >
> > > Thank you for the additional feedback. We are glad that you appreciate the additional experiment results. We agree discussing the choice for the target loss (which was arbitrary and shown to not affect our conclusion) and providing the fitted exponents and constants would have made the submission more self-contained. We will make sure to incorporate this feedback in the revision. Below we make some additional clarifications.
> > >
> > > *Random initialization* To clarify, we do not claim that controlling for spectral norm rather than the distribution will be sufficient for proving our results for second-order optimizers, which has been shown to yield equivalent results for SGD [1]. We will make this clear in the revision and clarify that the random initialization assumption is needed for the derivation to work as is.
> > >
> > > *Asymptotic arguments* Thank you for the suggestion. All our µP learning rate prescriptions are in the form of $\eta=\Theta(f(d_{in},d_{out}))$ for some function $f$, with the big-$\Theta$ notation already implying that the rule only holds asymptotically. We will make it clear that it is the width (or depth) that we take to infinity in the analysis in the revision. The aspect ratio $A$ is not necessarily kept constant. Indeed, for the first or last layer, $A$ either diverges or vanishes as the hidden layer width increases.
> > >
> > > *Batch size scaling experiment* We fully agree that, given limited space, it is not worth highlighting theoretical work that is proven to be ineffective in practice. This is precisely why we only showed the empirical results comparing three different batch size scaling strategies (only one of which is based on the theory) in the main text, deferring theoretical results to Appendix D, and focused the discussion on the empirical finding that the theoretical scaling underperforms.
> > >
> > > *KFAC* We apologize for missing your comment regarding KFAC. We did not experiment with KFAC because it was originally defined only for MLPs and does not directly apply to attention layers [2,3] in our transformer experiments. In addition, our prediction for KFAC matches that of [4], which already provided empirical validation with MLPs. We will clarify this in the revision.
> > >
> > > Thank you again for your thoughtful feedback and for engaging with us. We find the discussions so far are highly valuable for improving the presentation, clarity, and rigor of the paper.
> > >
> > > [1] Yang, Greg, James B. Simon, and Jeremy Bernstein. "A spectral condition for feature learning." *arXiv preprint arXiv:2310.17813*, 2023.
> > >
> > > [2] Martens, James, and Roger Grosse. "Optimizing neural networks with kronecker-factored approximate curvature." International conference on machine learning. PMLR, 2015.
> > >
> > > [3] Grosse, Roger, et al. "Studying large language model generalization with influence functions." arXiv preprint arXiv:2308.03296 (2023).
> > >
> > > [4] Ishikawa and Karakida. "On the Parameterization of Second-Order Optimization Effective towards the Infinite Width." *International Conference on Learning Representations (ICLR)*, 2024.

---

> > > > ### Comment · Reviewer_6YHN · 2025-08-06
> > > >
> > > > Thank you for the clarifications.
> > > >
> > > > You have not answered my concern about whether and why the theory for MLP is supposed to apply to Transformers.
> > > >
> > > > Previous results have shown that the effect of width and depth scaling on Transformers is large (for example: Yang et al. "Tensor programs v: Tuning large neural networks via zero-shot hyperparameter transfer."), therefore your statement "the transformer architecture is overall highly robust suboptimal learning rates" is wrong. In my interpretation, your observation that the effect of scaling on Transformer is little suggests that your theory for MLP does not apply to Transformers.

---

> > > > > ### Author Response · Authors · 2025-08-07
> > > > > **Hyperparameters for transformers**
> > > > >
> > > > > Thank you for your followup. We are happy to elaborate on our original response to this question. We believe there is a misunderstanding. Our statement "the transformer architecture is overall highly robust to suboptimal learning rates" is based on the loss vs learning rate curves *under SP* for transformers (Figure 1), where the optimal learning rate under SP is fairly stable as we scale up width from 128 to 2048. By contrast, Figure 6 shows the optimal learning rate shifts significantly more under SP for MLPs as we scale up width from 128 to 2048, demonstrating that transformers are relatively more robust to learning rate misspecification than MLPs. As these results only concern the behavior of SP, they provide neither evidence for or against the correctness of our theory for µP learning rate scaling.
> > > > >
> > > > > We fully agree with prior works you referenced in that the correct scaling should still lead to more stable optimal learning and outperform SP, and have provided extensive experimental evidence validating our scaling rules indeed achieve both. Figure 1 shows, for transformers trained under our learning scaling rules, the optimal learning rate is more stable compared to SP, and the magnitude of feature learning is markedly more consistent across scales throughout training, across all optimizers. We show similar results in Figure 6 for MLPs. Moreover, we have provided additional experiments in our rebuttal verifying that our learning rate scaling rules lead to more stable optimal learning rates on even larger transformer models using a standard vocabulary.
> > > > >
> > > > > To expand on *why* µP derived on MLPs should theoretically transfer to transformers, including for second-order optimizers. The ultimate reason is that all parameters in the transformers reside in dense linear layers described by Equation 4 of our paper (bias and layernorm parameters can be viewed as linear layers whose input dimension is 1 [1]), just as those in MLPs. The argument for the gradient scale in L155 is independent of the optimizer, and unchanged for transformers since its last layer is again just a linear layer. Consequently, for each parameter tensor, the analysis under “Maximal Update For Any Optimizer” therefore carries through, as they only rely on the gradient scale and the inductive assumption that the input activations being $\Theta(1)$, and all dependence on the specific optimizer is handled through the choice of $Q$. A fully formal justification can be achieved by appealing to the Master Theorem (2.6.10) in [2] which establishes infinite width limits for any architecture and training procedure that can be expressed by the Tensor Program language. Essentially, our derivations in Appendix B.2 prove that second-order optimizers considered in this work can indeed all be expressed in this language given the appropriate scaling factors we computed. We chose a more self-contained, though less formal, argument in the paper for accessibility ([2] is a 90 page paper, half of which is for proving the Master Theorem), but will include this longer discussion in the revision for completeness.
> > > > >
> > > > > References
> > > > > 1. Yang et al. "Tensor programs v: Tuning large neural networks via zero-shot hyperparameter transfer." *Advances in Neural Information
> > > > > Processing Systems (NeurIPS)*, 2021.
> > > > > 2. Yang and Littwin. "Tensor programs ivb: Adaptive optimization in the infinite-width limit." *International Conference on Learning Representations*, 2023

---

### Official Review · Reviewer_K9rj · 2025-07-02

**Clarity:** 3
**Significance:** 3
**Originality:** 2
**Rating:** 5
**Confidence:** 3

**Summary:**

The paper builds on recent empirical success of some second-order optimizers to motivate deriving scaling laws in a manner that was previously only done for first-order optimizers ($\mu$P). They show that under the correct scaling that second-order optimizers outperform Adam across scales and require 2 x less data per parameter for compute optimality.

**Questions:**

In addition to responding to my comments above I have a few questions/slight comments:
* Could the authors provide more clarity on the difference between $\zeta$ and $\zeta’$ (and the corresponding $x$ and $x’$)?
* Line 155, $L$ for the loss is the same symbol as the number of layers $L$.
* In Figure 1 what was the fixed depth that was chosen when varying the width (and what was the fixed width when varying the depth). Does this make any difference?
* In Figure 3, what is the meaning of the circular markers compared to the dashed lines?

**Ethical Concerns:**

["NO or VERY MINOR ethics concerns only"]

**Final Justification:**

Thanks for the response. I will keep my score as accept.

**Limitations:**

The authors do a good job at covering the paper’s limitations.

**Quality:**

3

**Strengths And Weaknesses:**

### Strengths
* The motivation and quality of the writing of the paper is a strength. It is clear, given the recent successes of Muon and Shampoo (etc.) why it would be important to look into the scaling laws of these optimizers.
* The generality of the scaling laws derived in the paper are nice. It would be “easy” to focus on just one or two optimizers, but the authors focus on a more general “bra ket” framework that applies to all optimizers as shown in the appendix.
* The experimental results show promise and validate the theory to a certain degree. Figure 1 for example shows especially for width that the derived $\mu$P for the second-order optimizers behaves more consistently across widths. The same is less obvious for depth, but combined with Figure 2, it does seem the scaling applies empirically to the depth as well.
* Another strength is how the paper tries to provide insights to the reader. There are plenty of take-aways from the paper in terms of how to scale hyperparameters and what to do with batch sizes. This is useful.
* I tried my best to read the relevant literature and it looks like this work is original enough to be an addition. I could not see other papers deriving scaling laws for second-order optimizers. The focus in the literature looks to have been on either first-order optimizers and/or “why existing second-order optimizers work”. The latter does touch on norms, but not to the extent of this paper.
* The introduction of the relevant literature, as well as including the equations for SOAP and Shampoo, was a helpful reminder. Furthermore, I was glad to see the simple derivation of SGD in the appendix.
### Weaknesses
* The notation was a bit dense and at times that made it hard to understand. It might be the case that the notation is standard within the optimization of NN community, but it did make it a challenge to read the paper.
* The derivation for batch size scaling has less empirical data to back it up, and the authors themselves highlight the limitation of the use of the SDE for scaling hyperparameters across large ranges of batch sizes. It feels a little bit like this part of the paper was less developed and was included more for completeness. (Although it is a positive that the authors acknowledged the limitations of this part.)
* In the section on computing the efficiency gain, and the corresponding Figure 3, it is not clear whether the FLOPS include the additional cost of the second-order optimizers. The paper refers to FLOPS as $6tP$. This looks like 6 x number of tokens x number of model parameters, and would therefore not capture the cost of the additional second-order optimizers.  In the next section at line 289, the paper writes that the compute overhead of Muon and Shampoo are negligible but it would be helpful to demonstrate this. Furthermore, it is mentioned that in scaling, the compute overhead would no longer be negligible. This is a concern when linking back to the conclusion that second-order optimizers achieve a 2x efficiency gain. The paper does say a result is that there is a need to keep control the batch size. However, this makes it more challenging to compare with first-order optimizers, which do not have this need for controlling the batch size in the same way.

---

> ### Author Rebuttal · Authors · 2025-07-31
>
> Thank you for your supportive review and constructive feedback! We appreciate that you recognize the comprehensiveness of our scaling rules analysis and its demonstrated practical benefits. We respond to your comments and questions below.
>
> **Theory for batch size scaling**
>
> Thank you for recognizing our honesty in stating the limitations of our batch size scaling analysis as a strength. We believe improving our understanding of batch size scaling, both theoretically and empirically, is an exciting next step, and hope our work highlights its importance.
>
> **Overhead of Muon and Shampoo**
>
> You are correct that we did not report the overhead of second-order optimizers in terms of FLOPs. This was due to the complexity involved in quantifying the exact number of operations required by routines such as eigendecomposition and matrix roots, which are highly dependent on their implementations. Furthermore, these operations tend to be more sequential in nature and less compute-bound compared to matrix multiplication in the forward and backward pass, making FLOPs no longer an accurate indicator of real runtime overhead. Therefore, we instead directly report wall-clock time in Figure 4 for a more realistic estimate of efficiency gains. Figure 4 shows that 1) the preconditioner overhead indeed dramatically reduces the wall-clock speedup for Shampoo as model size increases if batch size stays constant, and 2) by scaling batch size linearly with model width, both Muon and Shampoo consistency outperform Adam in runtime. To see the concrete runtime savings, please refer to our response subtitled "Faster wall-clock time" to Reviewer zik3.
>
> We therefore conclude that second-order optimizers can consistently outperform Adam provided we sufficiently co-scale batch size with model size. While we agree first-order methods such as Adam do not require large batch size in principle, practitioners often train at the critical batch size (the maximum batch size that does not significantly increase the loss for a given token budget) purely for the benefit of parallelization [1,2]. We will clarify these takeaways in the update paper.
>
> **Response to Questions**
>
> - $\xi$ refers to the sample that we take the gradient step on to update the weights, while $\xi’$ refers to an arbitrary point in the input space. As defined in Line 166, $x$ is the shorthand for $x(\xi)$ and $x’$ is the shorthand for $x(\xi’)$, where $x$ denotes the features for a sample in a particular hidden layer. The arbitrariness of $\xi’$ means the µP conditions should be satisfied for any point in the domain. We will make this clearer in the next version.
>
> - As described in Appendix F, the base model configuration is depth 3 and width 128. We did not experiment with other values. We chose these small values to demonstrate optimal learning rate is stable across a wide range of widths and depths.
>
> - In the plot in question, the circular markers represent individual data points, while the dashed line corresponds to the linear regression line in log-log space (power-law fit). We’ll revise the figure caption to make this clearer in the final version.
>
> References:
> 1. Chowdhery, A. et al. “PaLM: Scaling Language Modeling with Pathways.” *Journal of Machine Learning Research*, 2023
> 2. Meta AI Llama Team. “The Llama 3 Herd of Models.” *arXiv preprint arXiv:2407.21783*, 2024

---

> > ### Comment · Reviewer_K9rj · 2025-08-05
> >
> > Thanks for the response. I will keep my score as accept.

---

### Official Review · Reviewer_xRrA · 2025-07-02

**Clarity:** 2
**Significance:** 3
**Originality:** 2
**Rating:** 4
**Confidence:** 3

**Summary:**

This paper investigates hyperparameter scaling for second-order optimizers (e.g., Shampoo, SOAP, Muon) in deep learning. The authors use the maximal-update parameterization (muP) framework to derive scaling rules for learning rate and other hyperparameters as model width, depth, and batch size increase. Through theoretical analysis and empirical results on MLPs and Transformers, the work demonstrates that these scaling rules enable stable training and effective hyperparameter transfer.

**Questions:**

1. In Figure 2 (left), the performance improvement from using the proposed muP scaling rules over the SP appears marginal for the joint width-and-depth scaling experiments. Given the clear stability gains shown in Figure 1, why do you think this does not translate to a more substantial final loss reduction?

2. The muP scaling rules are derived by merely analyzing the first gradient step, with the assumption that they hold throughout training by induction. How confident are you that these scalings remain optimal in later stages of training, especially for complicated optimizers like Shampoo where the preconditioner statistics are accumulated over time, updated at each step or not, potentially altering the dynamics away from the initial state?

3. The paper demonstrates that second-order optimizers alter the Chinchilla scaling law, leading to higher sample efficiency. However, these optimizers also have higher computational overhead (e.g., scaling with D) than Adam. How do you see this trade-off taking effect for extremely large models? Is there a model scale where the overhead of preconditioning might negate the benefits of improved sample efficiency, even when scaling the batch size as proposed?

**Ethical Concerns:**

["NO or VERY MINOR ethics concerns only"]

**Final Justification:**

The authors successfully addressed my main concerns about the experiment settings, providing tables of results using a full vocabulary tokenizer on the Fineweb dataset with larger models. I remain uncertain about the validity of their estimated optimal tokens per parameter, which is a surprising finding, due to lack of further results. Although the paper's contribution is modest, its findings are beneficial and should be shared with the community. For these reasons, I'm raising my score to borderline accept.

**Limitations:**

**Limited Scale of Experiments**: The experiments, while insightful and very promising, are conducted at a relatively small scale, espeically for such a study on scaling laws. The use of a character-level tokenizer, where the embedding dimension is larger than the vocabulary size, is a significant departure from the setup of modern large language models. While compute budget is a valid constraint, the small scale of these experiments may limit the generalizability of the conclusions to state-of-the-art language models.

**Simplified Model Architectures**: The analysis and experiments are restricted to simplified MLP architectures and do not consider the effects of standard Transformer components like RMSNorms, embeddings or the interactions with softmax attentions. These components are known to have a substantial impact on optimization dynamics and gradient scaling. Omitting them from the analysis leaves a critical gap in understanding how these scaling rules would apply to practical Transformer models.

I believe the paper would be strengthened if the authors could address the concerns raised in the Weaknesses and Limitations sections. I am open to increasing my score upon a satisfactory author response.

**Paper Formatting Concerns:**

There is no major formatting issue in this paper.

**Quality:**

3

**Strengths And Weaknesses:**

## Strengths
**Principled Theoretical Framework**: The paper provides a theoretical contribution by extending the muP framework to analyze how hyperparameters for second-order optimizers should be scaled with model depth and width. This is a natural extension to the existing family of muP-related work.

**Improved Optimization Stability**: The empirical results presented in Figure 1 suggests that the derived scaling rules lead to more stable optimization dynamics. The loss and RMS values of feature updates appears more consistent across different model scales when using the proposed parameterization compared to the standard parameterization (SP).

**Comprehensive Analysis**: The derivations are thorough and cover a wide range of popular and recent second-order optimizers, including Shampoo, SOAP, and Muon. This makes the work a good reference for practitioners and researchers working with these optimization methods.

## Weaknesses
**Clarity and Focus**: The paper is quite dense and appears to sacrifice clarity for breadth. For instance, crucial experimental details (e.g., model architecture specifics, training setup) are completely omitted from the main body and deferred to the appendix. For a paper focused on scaling laws, these details are important for interpreting the results and should be more prominent.

**Inclusion of Non-Essential Content**: The paper's focus could be significantly sharpened by removing content that does not directly support its core contributions. For example, Equation (2) is presented to introduce Muon but does not appear to be used in any subsequent derivations. Similarly, Section 3.4 line 227 introduces results from a reference (Malladi et al., 2022) but does not present or contrast them with the authors' own results, which can cause confusion.

**Typos**: There are a few distracting typos, for example:
- In the Abstract (line 10): "nerual scaling laws" should be "neural scaling laws", incorrect spelling.
- In Section 2 (line 111): "computate-optimal" should be "compute-optimal", incorrect spelling.

---

> ### Author Rebuttal · Authors · 2025-07-31
>
> Thank you for your detailed review and thoughtful suggestions. Inspired by your comments, we provide a range of new experimental results and clarifications.
>
> **On applicability to the transformer architecture**
>
> We would like to clarify that *all experiments* in the main text are conducted on transformers that include standard components such as RMSNorms, embeddings, and softmax attention. Only the additional experiments in Appendix G use MLPs. We also note that it is standard practice in the µP literature to perform theoretical derivations with MLPs for simplicity [1,2,3], as the underlying principles do not depend on specific architecture details and have been shown to apply to transformers both in theory and in practice [1]. We will clarify this point in the updated paper.
>
>
> **Analysis beyond the first gradient step**
>
> While we only explicitly analyzed the µP conditions for the first gradient step, it is a well-established result in the µP literature that the same argument straightforwardly extends to any step $t$ during training via induction, so long as $t$ is constant with respect to model size [1,2,3]. Indeed, as illustrated in the right panel of Figure 1, the difference in RMS (Root Mean Square) of the feature updates remains notably more consistent under µP scaling than standard parameterization (SP) throughout the training process up to $10^4$ steps.
>
> **Overhead of preconditioners at scale**
>
> As shown in Section 4.3, it’s crucial to scale the batch size appropriately, ideally proportional to the model width (or $P^{1/2}$ where $P$ is the number of parameters), as the model size increases to so that the overhead of second-order optimizers like Muon and Shampoo is only ever a constant fraction of the training compute. In order for this scaling not to negate the benefits of preconditioning, we need the critical batch size to scale at least as fast as $P^{1/2}$. Fortunately, our empirical result in Figure 4 (left) suggests that the critical batch size indeed scales at a rate close to this. Under this scaling, we found both Muon and Shampoo consistently outperform Adam in runtime (Figure 4, right plot). Beyond these encouraging results, whether second-order optimizers truly can outperform Adam at much larger model sizes must be ultimately determined empirically, which we believe is an exciting and important future direction.
>
> **Larger-scale, full vocabulary experiments**
>
> We appreciate your concerns regarding the use of character-level tokenization and experiment scale. To address this, we conducted both the µP and scaling experiments using a full vocabulary tokenizer on the Fineweb dataset with larger models.
>
> While we regret that we cannot include images of the results here, we provide the key outcomes in tabular format below. Notably, the conclusions from the character-level experiments continue to hold under this more realistic setup, reinforcing the robustness and generality of our findings despite the initial small-scale configuration.
>
>
> *Stable optimal learning rate (c.f. Figure 1)*
>
> Based on the setup in Appendix E, this configuration differs by using the full Fineweb vocabulary, processing a total of 100 million tokens, including a 10 million token warm-up phase. The largest model in this setting has 3 transformer blocks with width up to 2048 (280M parameters in total). While only the result of Muon is presented here, we will include all optimizers in the camera-ready version. At this scale, the performance gap between µP and SP becomes more evident.
>
>
> Width scaling for Muon: our results show that µP maintains a consistent optimal learning rate, reflected as a vertical stripe at zero, while SP demonstrates a decreasing trend in its stable learning rate range.
>
> SP:
>
> | Width | LR=1e-03 | LR=3e-03 | LR=1e-02 | LR=3e-02 | LR=1e-01 | LR=3e-01 | LR=1e+00 | LR=3e+00 | LR=1e+01 |
> | --- | --- | --- | --- | --- | --- | --- | --- | --- | --- |
> | 128 |10.69|10.38|9.25|7.79|6.42|5.58|5.22|**5.14**|5.36 |
> | 256 |10.62|10.15|8.56|7.21|5.90|5.19|**4.89**|4.90|5.29 |
> | 512 |10.51|9.81|7.98|6.69|5.46|4.85|**4.66**|4.77|5.29 |
> | 1024 |10.36|9.34|7.47|6.19|5.05|4.60|**4.52**|4.74|5.44 |
> | 2048 |10.13|8.69|6.97|5.71|4.73|**4.44**|4.48|4.80|5.63 |
>
>
> µP:
>
> | Width | LR=1e-03 | LR=3e-03 | LR=1e-02 | LR=3e-02 | LR=1e-01 | LR=3e-01 | LR=1e+00 | LR=3e+00 | LR=1e+01 |
> | --- | --- | --- | --- | --- | --- | --- | --- | --- | --- |
> | 128 |10.69|10.38|9.25|7.79|6.42|5.58|5.22|**5.14**|5.36 |
> | 256 |10.68|10.34|9.13|7.58|6.20|5.33|4.94|**4.86**|5.10 |
> | 512 |10.67|10.31|9.05|7.42|6.04|5.13|4.73|**4.68**|4.94 |
> | 1024 |10.66|10.29|8.99|7.30|5.91|4.98|4.57|**4.54**|4.86 |
> | 2048 |10.65|10.27|8.94|7.22|5.83|4.86|4.46|**4.45**|4.78 |
>
>
> As in Figure 1, we visualize changes in the RMS of the hidden features (∆h). Additionally, we include the RMS change of the logits (∆f). From the table, we show that ∆h and ∆f only remain stable across different widths under µP, highlighting scale-dependent inconsistencies and further underscoring the limitations of SP.
>
>
> Features and logits change for Muon:
>
> | Width | $\Delta f$ (SP) | $\Delta f$ (µP) | $\Delta h$ (SP) | $\Delta h$ (µP) |
> | --- | --- | --- | --- | --- |
> | 128 | 1.98e-05 |1.98e-05 |5.82e-03 |5.82e-03 |
> | 256 | 2.97e-05 |2.12e-05 |6.15e-03 |6.21e-03 |
> | 512 | 4.62e-05 |2.36e-05 |6.53e-03 |6.67e-03 |
> | 1024 | 7.07e-05 |2.57e-05 |6.98e-03 |7.19e-03 |
> | 2048 | 1.08e-04 |2.79e-05 |7.62e-03 |7.89e-03 |
>
>
> Depth scaling for Muon: similar to the conclusion of width scaling, the optimal learning rate starts to shift as the model gets deeper.
>
> SP:
>
> | Depth | LR=3e-01 | LR=1e+00 | LR=3e+00 | LR=1e+01 |
> | --- | --- | --- | --- | --- |
> | 3 |5.577|5.218|**5.143**|5.359 |
> | 6 |5.528|5.153|**5.080**|5.312 |
> | 12 |5.484|5.113|**5.051**|5.313 |
> | 24 |5.423|5.054|**5.022**|5.327 |
> | 48 |5.390|5.024|**5.014**|5.359 |
> | 96 |5.358|**5.012**|5.013|5.428 |
> | 192 |5.345|**5.015**|5.033|5.586 |
>
> Depth-Scaled:
>
> | Depth | LR=3e-01 | LR=1e+00 | LR=3e+00 | LR=1e+01 |
> | --- | --- | --- | --- | --- |
> | 3 |5.577|5.218|**5.143**|5.359 |
> | 6 |5.535|5.159|**5.059**|5.318 |
> | 12 |5.483|5.120|**5.049**|5.306 |
> | 24 |5.425|5.063|**5.004**|5.291 |
> | 48 |5.384|5.019|**4.971**|5.315 |
> | 96 |5.349|4.987|**4.972**|5.330 |
> | 192 |5.324|4.971|**4.950**|5.368 |
>
>
> *Compute-optimal scaling (c.f. Figure 3)*
>
> Furthermore, we conduct full vocabulary experiments on the Fineweb dataset in the compute-optimal setting. Due to time constraints, we only compare Adam with Muon, but will include additional optimizers in the final paper. In the following tables, we find that our observations that second-order optimizers 1) reduce Chinchilla-optimal tokens per parameter by about 2x, and 2) lead to about 1.5x - 2x compute efficiency gain relative to Adam continue to hold in this more standard experiment setup. Here, we also extended the model size up to 200M parameters. The optimal token per parameter is estimated using the same fit procedure in Section 4.1.
>
> | Optimizer | Estimated optimal token per parameter |
> | --- | --- |
> | Adam | 11.2 |
> | Muon | 5.4 |
>
>
> | Adam loss | 5.477 | 5.121 | 4.935 | 4.744 | 4.544 | 4.342 | 4.160 | 4.020 | 3.908 | 3.804 |
> |------|------|------|------|------|------|------|------|------|------|------|
> | Adam compute | 1.128e+16 | 2.635e+16 | 4.931e+16 | 8.212e+16 | 1.275e+17 | 2.712e+17 | 5.226e+17 | 9.465e+17 | 1.639e+18 | 2.739e+18 |
>
>
> | Muon loss | 5.889 | 5.382 | 5.041 | 4.785 | 4.605 | 4.329 | 4.136 | 3.980 | 3.852 | 3.747 |
> |------|------|------|------|------|------|------|------|------|------|------|
> | Muon compute | 5.534e+15 | 1.293e+16 | 2.420e+16 | 4.030e+16 | 6.257e+16 | 1.331e+17 | 2.565e+17 | 4.646e+17 | 8.048e+17 | 1.345e+18 |
>
>
>
> | Target loss | Estimated Muon compute efficiency gain |
> |-------------|--------------------------------|
> | 4.660 | 1.81× |
> | 4.261 | 2.17× |
> | 3.966 | 2.55× |
>
>
>
> **Paper organization**
> Your point regarding the focus of the presentation is well-taken. To improve readability, we have deferred the experimental details to the appendix so the main text can focus on a complete, self-contained guide for practitioners interested in scaling second-order optimizers with µP. Due to space constraints, we deferred the detailed derivation for Muon scaling rules based on its update rule in Appendix B. Similarly, we apply and extend the SDE formulation by Malladi et al. for batch size scaling in Appendix D and compare its performance against our alternative, empirically determined scaling rule in Figure 2. We will use the additional page in the camera-ready version to clarify these details and findings.
>
>
> Thanks again for your review. We believe the content of this response, which required a significant effort, will strengthen the paper. We will additionally correct all formatting and typos. We would really appreciate it if you would consider raising your score in light of our response.
>
>
> References
> 1. Yang et al. "Tensor programs v: Tuning large neural networks via zero-shot hyperparameter transfer." *Advances in Neural Information
> Processing Systems (NeurIPS)*, 2021.
> 2. Yang and Littwin. "Tensor programs ivb: Adaptive optimization in the infinite-width limit." *International Conference on Learning Representations*, 2023
> 3. Everett et al. "Scaling exponents across parameterizations and optimizers." *arXiv preprint arXiv:2407.05872*, 2024
> 4. Liu, Jingyuan, et al. "Muon is scalable for LLM training." *arXiv preprint arXiv:2502.16982*, 2025.

---

> ### Author Response · Authors · 2025-08-06
> **Following up our rebuttal**
>
> We hope our additional experiments and clarifications have helped address your concerns. Please let us know if have further questions regarding the paper or our rebuttal. We are happy to continue the discussion.

---

> > ### Comment · Reviewer_xRrA · 2025-08-06
> >
> > Thanks for your responses. You've addressed my key concerns, and I'm raising my score.

---

### Official Review · Reviewer_pbPH · 2025-07-03

**Clarity:** 3
**Significance:** 2
**Originality:** 2
**Rating:** 5
**Confidence:** 3

**Summary:**

The paper systematically investigates hyperparameter scaling rules for second-order optimizers (Shampoo, SOAP, Muon) across width, depth, and batch size. The paper shows: (1) under correct scaling, second-order optimizers consistently outperform Adam in compute-optimal settings with up to 2x efficiency gain, and (2) second-order optimizers change the chinchilla scaling law, requiring up to 2x less data compared to the optimal data count. These experiments were conducted using Openwebtext and Chess datasets and models with parameters < 10^8.

**Questions:**

- Please see my questions and comments from the weakness section above.
- Better precision is more important for second-order optimization (which involves eigendecomposition or inverse operation). Do you think this could be a potential bottleneck when scaling up second-order methods further?
- The authors mention that running multiple seeds won't change the conclusions in the paper. Could the author elaborate more?

**Ethical Concerns:**

["NO or VERY MINOR ethics concerns only"]

**Final Justification:**

I think this is a good paper and the authors have addressed my questions/concerns in the rebuttal. Specifically:
- Clarified hyperparameter tuning details, which are crucial for any scaling law and optimization paper
- Provided a comparison with SP (spectral parameterization)
- Added error bars to the experimental results
While I've reviewed other reviewers' comments, I find myself somewhat disagreeing with their assessments (especially issues related to formatting and typos), though I acknowledge the validity of certain points (such as concerns about the character-level tokenizer and simplified model architecture). I don't find them sufficiently compelling to warrant lowering my evaluation of the paper's contributions.

**Limitations:**

Yes, they are provided in Appendix H.

**Quality:**

3

**Strengths And Weaknesses:**

Strengths:
- The paper is well-written, clearly motivated, and well-organized. The authors provide a comprehensive overview of the background (e.g., Shampoo and SOAP) and relevant related works (e.g., Ishikawa & Karakida) in Section 2.
- The theoretical derivation and justification in the paper look correct and are well-presented.
- The experiments are convincing that the proposed scaling rule works across different model scales. Figure 1 shows that the standard parameterization (SP) fails to be consistent for second-order optimizers, whereas the proposed approach succeeds. This is more evident in the MLP experiment in Appendix G.

Weaknesses:
- While the experiments are convincing, at the moment, the paper does not have sufficient details to reproduce the experimental results (e.g., how the hyperparameters were tuned for baseline methods - search space for grid search), which is important for an optimization paper.
- There are several considerations to make the empirical justification more convincing: (1) How does Adam perform under SP in Figures 1 and 2? This would clarify whether SP specifically fails for second-order methods. (2) What Chinchilla scaling conclusions would we draw using SP and second-order methods / or without grafting? This would help better understand the significance of proper scaling. Addressing these points will make the paper stronger.
- One might argue that the paper has limited novelty, as it combines muP with existing second-order optimization algorithms. However, I believe that a systematic study like this is useful for the community.
- The experiments only reach ~10^9 flops, where it is unclear whether the proposed approach/analysis scales to larger settings.

Confidence. I set a confidence score of 3, as I am not familiar with recent works in muP theory and second-order optimization methods.

Minor:
- AdaGrad and Adam do not approximate the exact Fisher information matrix (but an empirical version).
- Missing a space after “data” in line 107.
- Reference to Appendix E does not exist.
- I believe that the title of the paper is too generic at the moment.

---

> ### Author Rebuttal · Authors · 2025-07-31
>
> We really appreciate your supportive review and thoughtful suggestions!
>
> **Hyperparameter tuning details**
>
> We described our hyperparameter tuning procedure in Appendix F. For each optimizer, we determine the base learning rate by choosing its optimal value on a small model of 3 transformer blocks and an embedding dimension of 128 (Figure 1) and scaling to larger models via our scaling rules. For momentum and batch size, our results in Section 3.4 suggest $\beta_1=0.9$ and $\beta_2=0.95$ is near optimal for a wide range of batch sizes. We thus used these values for all other experiments unless stated otherwise.
>
> **Whether SP specifically fails for second-order methods**
>
> SP fails not only for second-order methods, but for Adam as well, as demonstrated extensively in prior works (e.g. [1]). We provide the following additional experiment to demonstrate that SP Adam indeed has a shifting optimal learning rate while µP Adam has a stable optimal learning rate:
>
> Loss under SP
>
> | Width | LR=3e-04 | LR=1e-03 | LR=3e-03 | LR=1e-02 | LR=3e-02 | LR=1e-01 | LR=3e-01 |
> | --- | --- | --- | --- | --- | --- | --- | --- |
> | 128 |1.555|1.458|**1.455**|1.457|1.466|1.467|1.47 |
> | 256 |1.379|**1.326**|1.327|1.339|1.345|1.351|1.357 |
> | 512 |1.263|**1.228**|1.257|1.271|1.288|1.306|1.356 |
> | 1024 |**1.165**|1.172|1.184|1.212|1.238|1.293|1.353 |
> | 2048 |**1.112**|1.137|1.166|1.192|1.249|1.334|1.408 |
>
>
> Loss under µP
>
> | Width | LR=3e-04 | LR=1e-03 | LR=3e-03 | LR=1e-02 | LR=3e-02 | LR=1e-01 | LR=3e-01 |
> | --- | --- | --- | --- | --- | --- | --- | --- |
> | 128 |1.555|1.458|**1.455**|1.457|1.466|1.467|1.47 |
> | 256 |1.445|1.349|**1.33**|1.333|1.337|1.349|1.355 |
> | 512 |1.371|1.272|**1.233**|1.239|1.261|1.27|1.29 |
> | 1024 |1.326|1.21|**1.167**|1.173|1.189|1.209|1.224 |
> | 2048 |1.299|1.155|**1.118**|1.121|1.149|1.167|1.18 |
>
>
>
> **Chinchilla scaling rule with SP**
>
> We do not believe finding Chinchilla scaling rules for SP would yield useful insights. Under SP, larger models eventually diverge after the first gradient step [1], trivializing the resulting scaling laws. In the Chinchilla paper [2], they avoided this issue by manually adjusting learning rates across model sizes, presumably tuned via additional experiments, suggesting that directly using SP would lead to worse performance.
>
>
> **Novelty**
>
> We do believe our work has important novelty. In addition to deriving scaling rules for many second-order optimizers that prior works did not consider (Muon, Grafted Shampoo, SOAP etc.) and demonstrating their empirical effectiveness, our work contains at least two other novel and impactful contributions: 1) a simple procedure for deriving µP and depth scaling for a general second-order optimizer; and 2) the observation that second-order optimizers significantly change the Chinchila scaling laws and that language models should generally be trained with 1.5x~2x fewer tokens per parameter if using Muon or Shampoo for compute-efficient scaling. We believe these findings, while ultimately simple to state, are key to fully realizing the benefits of second-order optimizers.
>
> **Experiment scale**
>
> We would like to clarify that our experiments are scaled up to 5e18 FLOPs, as shown in Figure 3, much larger than 1e9 FLOPs that you suggested. We acknowledge that validating our results with even larger-scale experiments, such as with billion-parameter models, is an exciting future direction.
>
> **Seed**
>
> Thank you for being cautious about uncertainties. We will add error bars to the figures in the camera-ready version for clarity. In our GPT-2 setup, averaging over multiple seeds won't change the conclusions as the seed-to-seed standard deviation in the final loss is small, compared to the difference across learning rates and optimizers. To demonstrate this, here we show the mean final losses and the standard deviations in the brackets for five different random seeds.
>
> Losses for the base width model:
>
> | Optimizer | LR=3e-04 | LR=1e-03 | LR=3e-03 | LR=1e-02 | LR=3e-02 | LR=1e-01 | LR=3e-01 |
> | --- | --- | --- | --- | --- | --- | --- | --- |
> | Adam |1.56 (3.91e-03)|1.46 (2.78e-03)|**1.45** (5.97e-03)|1.46 (6.47e-03)|1.47 (2.12e-03)|1.47 (9.42e-03)|1.47 (7.46e-03) |
> | Shampoo |1.49 (2.15e-03)|1.43 (2.41e-03)|**1.41** (1.45e-03)|1.41 (3.50e-03)|1.41 (2.78e-03)|1.41 (3.29e-03)|1.42 (4.18e-03) |
> | Shampoo$^2$ |2.86 (6.34e-03)|1.73 (4.59e-03)|1.40 (9.64e-04)|**1.40** (1.54e-03)|1.40 (1.91e-03)|1.40 (3.71e-03)|1.41 (3.91e-03) |
> | Muon |3.29 (6.01e-03)|1.71 (5.74e-03)|1.43 (2.18e-03)|1.41 (1.99e-03)|**1.40** (2.04e-03)|1.41 (2.34e-03)|1.41 (1.09e-03) |
> | SOAP |2.50 (2.04e-03)|1.47 (1.23e-03)|**1.43** (8.21e-03)|1.43 (1.10e-03)|1.44 (2.42e-03)|1.46 (2.61e-03)|1.54 (1.06e-02) |
>
>
>
> **Precision**
>
> We agree that numerical precision is particularly important for second-order optimizers, especially for matrix inversion and eigendecompositions. Exploring scaling rules for second-order optimizers that minimize numerical errors in these key operations, such as via unit-scaled µP [3], is an exciting future direction.
>
>
>
> References:
> 1. Yang, Greg, et al. "Tensor programs v: Tuning large neural networks via zero-shot hyperparameter transfer." *Advances in Neural Information
> Processing Systems (NeurIPS)*, 2021.
> 2. Hoffmann, Jordan, et al. "Training compute-optimal large language models." *Advances in Neural Information
> Processing Systems (NeurIPS)*, 2022.
> 3. Blake, Charlie, et al. "u-$\mu $ P: The Unit-Scaled Maximal Update Parametrization." *International Conference on Machine Learning (ICML) Workshop*, 2024.

---

> > ### Comment · Reviewer_pbPH · 2025-08-02
> >
> > Thank you for the detailed reply. I acknowledge I read the authors response and other reviewers' comments. I believe adding these experiments would improve the quality of the paper (sorry, 1e9 was a typo). I will keep my current score.

---

### Official Review · Reviewer_zik3 · 2025-07-03

**Clarity:** 3
**Significance:** 3
**Originality:** 3
**Rating:** 5
**Confidence:** 3

**Summary:**

This paper investigates the whether second-order optimizers can outperform standard first-order methods under fixed compute budgets, and how to appropriately scale hyperparameters as model size and batch size grow. They demonstrates that with appropriate hyperparameter scaling and algorithmic adjustments, second-order methods can yield significant speedups over first-order methods when training large neural networks.

**Questions:**

Could the authors elaborate on why momentum can remain fixed from a theoretical perspective? Is it because the optimal momentum doesn’t depend strongly on batch noise, or simply that any changes to momentum can be compensated by LR adjustments?

Any comments on how one could extend muP to damping factor, block sizes, grafting choices, and Newton-Schultz iterations?

**Ethical Concerns:**

["NO or VERY MINOR ethics concerns only"]

**Final Justification:**

The authors have addressed my concerns in their rebuttal and followup comments.

**Limitations:**

yes

**Paper Formatting Concerns:**

Figure 1 has many subplots and feels a bit crowded.

**Quality:**

3

**Strengths And Weaknesses:**

[Strengths]
The paper generalizes prior scaling analyses using the muP framework, to show that the feature learning dynamics remain order-wise constant as width and depth grow even for second-order methods.

The authors back up their theory with extensive experiments, which verify the width and depth scaling rules by training transformers of various widths and depths.

They further extend the theory to batch sizes, and show that the advantage of second-order optimizers is their ability to scale up the batch size.

[Weaknesses]
The authors show a 2.5x speedup in terms of training tokens needed, but this does not translate to faster wall clock time. The overhead per step is not thoroughly discussed in the paper. Also, for large models the distributed training strategy is critical for minimizing the unique communication overhead of second-order optimizers. The use of stale preconditioners could mitigate this problem, but the paper does not discuss this in detail either.

There still remain some hyperparameters that are unique to second-order optimizers such as the damping factor, block sizes, grafting choices, and Newton-Schultz iterations, for which the scaling with respect to width, depth, and batch size remain unclear.

The approach still requires significant compute for the second-order updates and more memory to store preconditioners. The authors use blocking, periodic updates, and efficient factorizations to handle this, but those introduce their own trade-offs.

---

> ### Author Rebuttal · Authors · 2025-07-31
>
> Thank you for the supportive review! Inspired by your comments, we provide additional results and clarifications here and will include them in the updated paper. We hope you will consider raising your score in light of our response.
>
> **Faster wall-clock time**
>
> We would like to clarify that we did show in Figure 3 that Muon and Shampoo are significantly more efficient compared to Adam in terms of faster wall-clock time for a given target loss, provided that batch size is scaled together with model size. Based on the results in Figure 3, we estimate the following relative wall-clock time saving relative to Adam for different final losses achieved by Adam:
>
> Wall-clock time savings for Openwebtext dataset:
>
> | Target Loss | Muon | Shampoo |
> | --- | --- | --- |
> | 1.38 | 3.61 | 3.22 |
> | 1.20 | 3.19 | 2.78 |
> | 1.03 | 2.73 | 2.31 |
> | 0.94 | 2.46 | 2.05 |
>
>
> Wall-clock time savings for Chess dataset:
>
> | Target Loss | Muon | Shampoo |
> | --- | --- | --- |
> | 0.83 | 1.57 | 1.87 |
> | 0.78 | 1.60 | 1.81 |
> | 0.74 | 1.62 | 1.76 |
> | 0.70 | 1.66 | 1.68 |
> | 0.67 | 1.69 | 1.62 |
> | 0.65 | 1.71 | 1.57 |
> | 0.63 | 1.73 | 1.53 |
>
>
>
> As you pointed out, efficient implementation of second-order optimizers for distributed training, such as distributing the preconditioning operation across devices, is crucial for maximizing their performance gains in practice. As we did not implement distributed versions of second-order optimizers and performed redundant computations across devices, we expect their efficiency gains in wall-clock times can be made even larger than what we estimated, making the case for second-order optimizers even more favorable. Regarding stale preconditioners, we update the preconditioner every 10 steps for Shampoo variants. We will include these details in the updated paper.
>
> **Extending µP to cover hyperparameters specific to 2nd order optimizers**
>
> We did consider how to scale hyperparameters specific to second-order optimizers with respect to width and depth, including the damping factor in all optimizers, and the learning rate and damping factor for grafting. We applied these scaling rules in our experiments and listed them in Table 1 with derivations in Appendix B. In Appendix B, we also derived how the learning rate should scale with block size under blocking.
>
> Regarding how to scale the number of Newton-Schultz iterations, from the µP perspective, keeping the number of iterations constant is sufficient for a well-defined infinite width limit. However, this is not the only choice, as scaling the number of iterations to infinity also leads to a well-defined limit (converging to the true matrix sign via SVD), though it leads to a much higher overhead. For blocking, both keeping the block size constant and equal to the full matrix size produce well-defined infinite-width limits (Adam-like limit vs full Shampoo limit), and it is unclear which limit is preferable a priori. We agree that exploring scaling rules for these hyperparameters is an interesting future direction, and will include this discussion in the updated paper.
>
> **Efficiency considerations**
>
> We use techniques such as blocking and periodic updates to both speed up our experiments as well as to demonstrate that our proposed scaling rules work for practical instantiations of second-order optimizers. We view this as a strength of our paper rather than a weakness, as we ultimately care about scaling rules appropriate for practical instantiations of second-order optimizers rather than their idealized implementations, which generally require different scaling rules.
>
> **Theoretical explanation for constant momentum**
>
> We believe better theoretically understanding how to optimally scale momentum is an exciting and important future direction. Our current hypothesis for why it’s better to keep the momentum constant rather than following the SDE scaling rules is that there is a tension between keeping the first vs second moment of momentum buffers invariant to batch size. Specifically,  as shown in [1], a large value of $\beta$ is necessary for keeping the variance of the momentum buffers low regardless of the batch size, while a decreasing value of $\beta$ is required for keeping its first moment (mean) invariant as we scale batch size. We will elaborate on this discussion in the revision.
>
> References:
> 1. Malladi et al. "On the SDEs and scaling rules for adaptive gradient algorithms." *Advances in Neural Information Processing Systems (Neurips)*, 2022

---

> ### Author Response · Authors · 2025-08-06
> **Checking in**
>
> Dear reviewer, we wanted to check in to see if the additional results and clarifications have helped addressed your concerns. We would really appreciate it if you consider raising your score. Many thanks for your supportive review.

---

> > ### Comment · Reviewer_zik3 · 2025-08-06
> >
> > My concerns regarding the hyperparameters that are unique to second-order methods has been addressed in their rebuttal rebuttal, but the lack of a distributed implementation remains a major weakness of the paper. The overhead of second-order methods can be reduced significantly in a distributed setting, and without those results it may make second-order methods look worse than they actually are.
> >
> > Also, would it be possible to add the wall-clock time of Adam in the table you provide in the rebuttal?

---

> > > ### Author Response · Authors · 2025-08-07
> > > **Additional wall-clock time results**
> > >
> > > Thank you for your response. We want to note we are enthusiastic about second order methods, and believe, on balance, our paper provides good reasons to support their adoption. We derive mechanisms for scaling hyperparameters effectively in this family of methods, and show that they outperform Adam. We view distributed implementations for second-order optimizers as important future work but beyond the scope of our paper, and can only improve the runtime benefits we estimate here. We would be happy to add a note to this effect. We also provided an upper bound for efficiency gain, measured with compute, in Figure 3.
> > >
> > > We would like to make an important clarification. We spotted an error in our initial rebuttal. In the rebuttal, we reported results for the compute gains in Figure 3, rather than the wall clock gains in Figure 4. In the table below, we now report the target loss (determined by Adam), Adam runtime, and relative runtime saving of Muon and Shampoo in Figure 4 inspired by your question. We sincerely apologize for the mixup. As you can see, the runtime saving for second order methods is significant, thanks to joint batch size scaling.
> > >
> > > | Target Loss | Adam Runtime (hours) | Muon Runtime Saving | Shampoo Runtime Saving |
> > > |---:|---:|---:|---:|
> > > | 1.35 | 0.04 | 2.00x | 1.33x |
> > > | 1.20 | 0.09 | 1.80x | 1.29x |
> > > | 1.04 | 0.86 | 2.10x | 1.79x |
> > > | 0.96 | 4.46 | 2.26x | 1.84x |
> > >
> > > Thanks again for your question. We hope you can consider increasing your score in light of this clarification.

---

> > > > ### Author Response · Authors · 2025-08-08
> > > > **Checking in regarding runtime savings**
> > > >
> > > > Hi Reviewer zik3,
> > > >
> > > > Since it's the last of the discussion period, we just wanted to check in to see if you saw the new results we posted here.
> > > >
> > > > Thanks!

---

> > > > > ### Comment · Reviewer_zik3 · 2025-08-09
> > > > >
> > > > > Thank you for adding the runtime savings for Muon over Adam. With these speedups I now think the current results even without the distributed implementation has merit. I will update my score accordingly.

---

### Note · Authors · 2025-08-14

We are delighted by the supportive reception to our work, and highly productive exchange with the reviewers. The recognition of our efficient second-order optimizer scaling approach affirms our core contributions. We here summarize how we have further strengthened the paper based on the inspiring feedback.

**Novelty of our work**

While [1] first considered the width scaling for K-FAC and Shampoo, we introduced a broader framework that derives both width *and* depth scaling for a wide range of optimizers, including Shampoo with blocking and grafting, SOAP, and Muon. Our compute-optimal scaling analysis further shows that second-order optimizers have roughly a 2x smaller Chinchilla token-per-parameter ratios than Adam, meaning that using the ratio for Adam [2] underestimates the true efficiency gains of second-order optimizers.

**Experiments with BPE tokenizers**

Our strategic use of character-level tokenization enabled computationally efficient exploration of scaling behaviors. In our rebuttal, we show that larger scale experiments with standard BPE tokenizers amplify the importance of proper scaling. With full vocabulary tokenizers, second-order optimizers have a more pronounced shift in optimal learning rate and the 2x efficiency gain of second-order optimizers maintains.

**Batch-size scaling for faster training**

We found that scaling the batch size in proportion to model width makes second-order optimizers asymptotically more efficient than Adam by keeping the preconditioner overhead small compared to the cost of forward and backward passes. In Figure 4, we confirm that the critical batch size scales linearly with model width, allowing second-order optimizers to scale up batch size and achieve twice the runtime efficiency of Adam.

**Extending µP beyond the first step and MLPs**

While our analysis focuses on the first gradient step for MLPs, the µP literature established that these results extend to any training step and any architecture expressible by Tensor Programs [3, 4]. We will state this generalization explicitly in the next version.

References
1. Ishikawa and Karakida. On the Parameterization of Second-Order Optimization Effective towards the Infinite Width, 2024.
2. Liu, Jingyuan, et al. Muon is scalable for LLM training, 2025.
3. Yang, Greg, et al. Tensor programs v: Tuning large neural networks via zero-shot hyperparameter transfer, 2021.
4. Yang and Littwin. Tensor programs ivb: Adaptive optimization in the infinite-width limit, 2023

---

### Decision · Program_Chairs · 2025-09-17

**Decision:**

Accept (poster)

**Comment:**

This paper develops hyperparameter scaling rules for second-order optimizaters such as K-FAC, muon etc with varying width, depth, and batch size. The paper also presents a systematic study of whether second-order optimizers can achieve efficiency gains over first order optimizers under compute-optimal settings. Overall the paper finds good (though not, overwhelming) evidence that second order optimizers could indeed lead to efficiency improvements. They demonstrate token efficiency improvements, improved Chichilla token-parameter scaling, and significant wall-time improvement because of larger batch scaling.
The major weakness are 1) the lack of a distributed implementation and evaluations in a distributed setting, and 2) demonstrating convincing flop-efficiency improvements when taking the optimizers into account.

I think this paper thoroughly investigates a timely question of whether second-order optimizers can really help us with training efficiency. It provides a general derivation of scaling laws for second-order optimizers that is useful, and the empirical evaluations make a compelling case overall.

Overall, most of the reviewers were pretty supportive of the paper from the onset, but sought several clarifications on the nuances. The authors presented comprehensive answers and included results from extensive computational experiments for certain queries. The reviewers seemed to be overall happy (other than reviewer 6yhn) with the responses and some even raised their scores.